# Mixture of Experts Provably Detect and Learn the Latent Cluster Structure in Gradient-Based Learning

**Ryotaro Kawata** [* 1 2]  **Kohsei Matsutani** [* 1 2]  **Yuri Kinoshita** [1]  **Naoki Nishikawa** [1 2]  **Taiji Suzuki** [1 2]

## Abstract

Mixture of Experts (MoE), an ensemble of specialized models equipped with a router that dynamically distributes each input to appropriate experts, has achieved successful results in the field of machine learning. However, theoretical understanding of this architecture is falling behind due to its inherent complexity. In this paper, we theoretically study the sample and runtime complexity of MoE following the stochastic gradient descent (SGD) when learning a regression task with an underlying cluster structure of single index models. On the one hand, we prove that a vanilla neural network fails in detecting such a latent organization as it can only process the problem as a whole. This is intrinsically related to the concept of *information exponent* which is low for each cluster, but increases when we consider the entire task. On the other hand, we show that a MoE succeeds in dividing this problem into easier subproblems by leveraging the ability of each expert to weakly recover the simpler function corresponding to an individual cluster. To the best of our knowledge, this work is among the first to explore the benefits of the MoE framework by examining its SGD dynamics in the context of nonlinear regression.

## 1. Introduction

Mixture of Experts (MoE) (Jacobs et al., 1991; Jordan & Jacobs, 1993), an ensemble of specialized models equipped with a router that dynamically distributes each input to appropriate experts, has been extensively studied and successfully deployed in a wide range of scenarios over the past few years. A key milestone was the development of sparsely-gated MoE (Shazeer et al., 2017), which was later integrated into transformer-based large language models (LLMs) and further refined in subsequent works (Fedus et al., 2022; Achiam et al., 2023; Georgiev et al., 2024; Jiang et al., 2024; Liu et al., 2024). This kind of MoE enables the activation of only a limited number of trained experts in one forward pass, drastically reducing the inference cost while maintaining performance competitive with other successful architectures of the same order of parameters.

However, theoretical understanding of this architecture is falling behind due to its inherent complexity. Especially, while the mechanism of the initialization, the optimization procedure and the behavior of the router are essentially the same for each expert, it has been repeatedly reported that each expert ultimately specializes in its own way, each contributing to different aspects of the learned task. It is still unclear why such phenomenon happens and why the router can learn to fairly distribute an input to appropriate experts without collapsing to a single expert.

To address these fundamental questions, prior work mathematically studied the mechanism of MoE from the perspectives of approximation theory on MoE for multi-level data (Fung & Tseung, 2022), statistical learning in Gaussian MoE models (Ho et al., 2022; Nguyen et al., 2023; 2024a) and nonlinear regression (Nguyen et al., 2024b;c), as well as optimization in both classification (Chen et al., 2022; Chowdhury et al., 2023) and linear regression, especially for continual learning (Li et al., 2024). However, a clear explanation of the success of the MoE is lacking in the context of *optimization* in *nonlinear regression* which is a more general problem than optimization in classification.

Therefore, in this paper, we focus on such a broader problem setting of optimization in nonlinear regression. We will theoretically study the sample and runtime complexity of MoE optimized with the stochastic gradient descent (SGD) when learning a regression task with an underlying cluster structure.

**Contributions**   Our contributions are summarized as follows. On the one hand, we prove that a vanilla neural network fails in detecting such a latent organization as it can only process the problem as a whole. This is intrinsically

---

[*]Equal contribution  [1]The University of Tokyo, Japan [2]Center for Advanced Intelligence Project, RIKEN, Japan. Correspondence to: Ryotaro Kawata <kawata-ryotaro725@g.ecc.u-tokyo.ac.jp>, Kohsei Matsutani <matsutani-kohsei140@g.ecc.u-tokyo.ac.jp>.

*Proceedings of the 42$^{nd}$ International Conference on Machine Learning*, Vancouver, Canada. PMLR 267, 2025. Copyright 2025 by the author(s).

related to the concept of *information exponent* which is low for each cluster, but increases when we consider the entire task. On the other hand, we show that a MoE succeeds in dividing this problem into easier subproblems by leveraging the ability of each expert to weakly recover the simpler function corresponding to an individual cluster. To the best of our knowledge, this work is among the first to explore the benefits of the MoE framework by examining its SGD dynamics in the context of nonlinear regression.

**Notation** $\mathbb{P}[\cdot]$ and $\mathbb{E}_x[\cdot]$ denote the probability of an event and the expectation over the randomness of a random variable $x$. $O(\cdot)$ and $o(\cdot)$ stand for the big-O and little-o notations with respect to $d$. $\Omega(\cdot)$ and $\Theta(\cdot)$ represent the lower and tight bounds. $\tilde{O}(\cdot)$, $\tilde{\Omega}(\cdot)$ and $\tilde{\Theta}(\cdot)$ denotes the upper, lower, and tight bound ignoring any poly-logarithmic constant. We call a probabilistic event $A$ happens *with high probability* (or w.h.p.) if $\mathbb{P}[A] \geq 1 - d^{-a}$ with a sufficiently large constant $a > 0$; the high probability events are closed under union bounds over sets of size $\text{poly}d$.

## 2. Related Works

**Theory of Mixture of Experts.** Various aspects of MoE have been theoretically studied in the context of deep learning so far. Ho et al. (2022) and Nguyen et al. (2023; 2024a) studied the convergence rate of expert estimation in Gaussian MoE models for classification, and Nguyen et al. (2024b;c) led similar investigation for MoE with a softmax gating for regression problems. Chen et al. (2022) pioneered studies of feature learning with MoE and analyzed the training of nonlinear MoE under a mixture of classification problems. Building on this, Chowdhury et al. (2023) extended the analysis to patch-level routing, addressing binary classification problems within nonlinear MoE settings. Li et al. (2024) focused on continual learning scenarios, but the analysis was limited to linear regression problems and linear MoE. In this work, we consider the broader and more practical problem setting of nonlinear regression problem with a nonlinear MoE model following a gradient-based optimization.

**Gradient-based Feature Learning** Gradient-based feature learning of low-dimensional functions using neural networks has garnered significant attention. Subjects of research encompasses functions such as single-index models (Dudeja & Hsu, 2018; Ba et al., 2022; Bietti et al., 2022; Abbe & Boix-Adserà, 2022; Mousavi-Hosseini et al., 2023b; Ba et al., 2023) and multi-index models (Damian et al., 2022; Ben Arous et al., 2022; Mousavi-Hosseini et al., 2023a; Bietti et al., 2023; Collins-Woodfin et al., 2023; Dandi et al., 2024a). The *information exponent* $k^*$, or *leap complexity* (Abbe et al., 2023), of the target is known to govern its difficulty of learning it, generally requiring a sample

complexity of $n = \tilde{O}(d^{k^*-1})$ (Arous et al., 2021), where $d$ is the input dimension. Damian et al. (2023) improved this rate to $\tilde{\Theta}(d^{\frac{k^*}{2}})$ by smoothing the landscape. Subsequently, techniques such as reusing batches (Dandi et al., 2024b; Lee et al., 2024; Arnaboldi et al., 2024) or altering loss function (Joshi et al., 2024) enabled to surpass the CSQ lower bound (Damian et al., 2022; Abbe et al., 2023). These approaches improve the sample complexity near information-theoretic limit $n \asymp d$, which is associated with the generative exponent (Damian et al., 2024). This approach based on the information exponent contributes to deepening our general understanding about the complexity of a task and has been applied to specific architectures or techniques, such as pruning (Vural & Erdogdu, 2024), pretrained transformer (Oko et al., 2024b), adversarially robust learning (Mousavi-Hosseini et al., 2024) and LoRA (Dayi & Chen, 2024). However, the application of this framework to MoE has not been explored yet, and it may hold promise for elucidating the intricate mechanism of MoE. Recently, Oko et al. (2024a) has conducted an extensive theoretical study on additive models, where several single-index models form a ridge combination. Our setting is analogous to this work, where the data exhibit an additive structure derived from diverse clusters.

## 3. Problem Setting and Preliminaries

In this section, we clarify the problem setting, including the data generation procedure, the formulation of the MoE, and the mathematical description of the training algorithm.

### 3.1. Data Generation

Let us first formally introduce the notion of information exponent.

**Definition 3.1** (Information Exponent). Let $\{\text{He}_j\}$ be the normalized Hermite polynomials. The Hermite expansion of a square-integrable function $f$ is given as $f(z) = \sum_j \frac{\alpha_j}{\sqrt{j!}} \text{He}_j(z)$. The information exponent is defined as $\text{IE}(f) := k^* = \inf_{j \geq 0} \{j \mid \alpha_j \neq 0\}$.

The information exponent is defined as the index of the first non-zero coefficient in the Hermite expansion of the nonlinear target function(Arous et al., 2021). The complexity of learning a nonlinear function via two-layer neural network optimized by SGD is closely associated with this value (Arous et al., 2021; Ge et al., 2018; Dudeja & Hsu, 2018; Bietti et al., 2022; Damian et al., 2022; Oko et al., 2024a).

The generation process is defined as follows.

**Assumption 3.2** (Teacher Models). Let $C = O_d(1)$ be the number of clusters. Let $f_c^*$ $(c = 1, \ldots, C)$ represent the local task specific to each cluster , and let $g^*$ denote the global task shared among clusters. A data pair $(x_c, y_c)$ in

the cluster $c$, where $c \sim \text{Unif}[1, \ldots, C]$, is generated as $x_c \sim \mathcal{N}(\rho v_c, I_d)$,

$$y_c = f_c^*(w_c^{*\top} x_c) + s_c g^*(w_g^{*\top} x_c) + \nu,$$

where $\nu \sim \mathcal{N}(0, \zeta^2)$ denotes an additive Gaussian noise that accounts for observation uncertainty. It is assumed to be sampled independently of the input $x_c$. The scalar $\rho \in \mathbb{R}$ represents a scaling factor that modulates the magnitude of the cluster mean vectors $v_c \in \mathbb{S}^{d-1}$, and is assumed to satisfy $\rho \simeq A_\rho$, where $A_\rho$ is a sufficiently large constant upper bounded by $\text{poly} \log d$. The coefficient $s_c \in \mathbb{R}$ encodes the influence of the global task within cluster $c$, and is constrained such that $\sum_c s_c = 0$ and $s_c = \Theta(1)$ for all $c$. $f_c^*$ and $g^*$ are univariate polynomials with information exponent $k^* > 2$ and degree $p^*$, and feature indices $w_c^*, w_g^* \in \mathbb{S}^{d-1}$. We write the Hermite expansion of $f_c^*$ and $g^*$ as $f_c^* = \sum_{i=k^*}^{p^*} \frac{\beta_{c,i}}{\sqrt{i!}} \text{He}_i$ and $g^* = \sum_{i=k^*}^{p^*} \frac{\gamma_i}{\sqrt{i!}} \text{He}_i$, respectively. The Hermite coefficients satisfy $|\beta_{c,k^*}| = |\gamma_{k^*}|$ for all $c$. The link functions and the index features are normalized as $\mathbb{E}_z[f_c^*(w_c^{*\top} z)^2] = 1$, $\mathbb{E}_z[g^*(w_g^{*\top} z)^2] = 1$, $\|w_c^*\| = 1$, and $\|w_g^*\| = 1$, where $z \sim \mathcal{N}(0, I_d)$.

This model is designed to introduce task interference by $\sum_c s_c = 0$, making the learning process more challenging as gradients from different clusters conflict, hindering effective learning. This scenario is closely related to a line of work on gradient interference in multi-task learning (Yu et al., 2020; Liu et al., 2021; Guangyuan et al., 2023; Zhang et al., 2024b) and recent work on MoE has also addressed this issue (Liu et al., 2024; Yang et al., 2025). Note that all $f_c^*$ and $g^*$ have the same information exponent $k^* > 2$ and their $k^*$th coefficients have the same absolute values, which implies that all $f_c^*$ and $g^*$ have the same difficulty, making it even more difficult to distinguish each component from the others. We will show that a vanilla neural network is incapable of handling such tasks, whereas the MoE can. We also suppose that $k^*$ is even, instead of assuming that the products of the Hermite coefficients of the teacher and the student models are positive in Simsek et al. (2024). We also impose a condition for each feature vectors $w_i^*$ and $v_c^*$ as follows.

**Assumption 3.3** (Task Correlation). For all $i, i' \in [C] \cup \{g\}$ such that $i \neq i'$, $w_i^{*\top} w_{i'}^* = \tilde{O}(d^{-\frac{1}{2}})$. Moreover, for all $i \in [C] \cup \{g\}$ and $c \in [C]$, $v_c^\top w_i^* = 0$.

This condition indicates that the tasks for each cluster are diverse. The correlation between two vectors $w_i^*$ can be satisfied, for instance, when the vectors are randomly drawn from $\text{Unif}(\mathbb{S}^{d-1})$. The cluster signal $v_c$ is assumed to be orthogonal to all feature indices for analytic tractability, and we believe this is not a necessary condition of our subsequent result. While Chen et al. (2022) assume mutual orthogonality among the feature indices, we relax this assumption in our analysis. Indeed, a randomized correlation

$v_c^{*\top} w_i^* \simeq \tilde{O}(d^{-1/2})$ should only introduce a negligible perturbation $\simeq d^{-1/2}$ to the Hermite coefficients $\beta_{c,i}$ and $\gamma_i$ of the teacher models.

## 3.2. Structure of the MoE

A MoE consists of its $M$ experts $f_1, \ldots, f_M$, a gating function $h(x; \Theta) = \Theta^\top x$ where $\Theta = (\theta_1, \ldots, \theta_M) \in \mathbb{R}^{d \times M}$ and a routing strategy that uses the output of the gating function $h = (h_1, \ldots, h_M)^\top$ to distribute the input to the appropriate experts. For example, for a top-1 routing, the index of the assigned expert is chosen as $m(x) := \text{argmax}_m h_m(x)$. We also define $\pi_m(x; \Theta) = \exp(h_m(x)) / \sum_{m'} \exp(h_{m'}(x))$ as the softmax gating functions.

In this paper, we will focus on two routing strategies. On the one hand, we define $F_1(x; W, \Theta)$, the output of MoE following a top-1 routing weighted by the corresponding softmax gating value, i.e., $F_1(x; W, \Theta) = \pi_{m(x)}(x) f_{m(x)}(x; W_m)$. This weighting enables to track the gradient of the gating function which is technically impossible with the simple top-1 routing. On the other hand, we introduce the (adaptive) top-$k$ routing. We adaptively choose $k$ experts for each input $x$ based on the value of each $h_m$ and a threshold. Here, we set the threshold to 0 and define the output as $\hat{F}_M(x; W, \Theta) := \sum_{m=1}^M \mathbb{1}[h_m(x) \geq 0] f_m(x; W_m)$ in Phase II; formally defined in Section 3.3. This choice of router is related to a recurrent problem that the router may fail to determine the appropriate expert when there are several models playing similar roles. After Phase II with top-1 routing, a data point $x_c$ is no longer routed to experts outside the set of professional experts (formally defined in Definition 4.7) for cluster $c$; however, competition among the professional experts may still occur. In general, this can result in issues related to load imbalance or the emergence of redundant experts, which may result from top-$k$ routing with a fixed $k \in \mathbb{N}$ (Zhou et al., 2022). To address this, expert choice routing (Zhou et al., 2022), soft MoE (Puigcerver et al., 2024), and several auxiliary losses, such as load balancing loss, importance loss (Shazeer et al., 2017), and z-loss (Zoph et al., 2022), were heuristically introduced to promote the even distribution of data and encourage diversity among experts. There have indeed been prior attempts to vary the number of activated experts depending on each token (Huang et al., 2024; Zeng et al., 2024). With our adaptive top-$k$ routing in Phase III and IV, we can also avoid this phenomenon without changing the loss as we will show that it prevents the data from being routed to non-corresponding experts and ensures that the experts that could be activated during inference are trained evenly.

Importantly, our adoption of adaptive top-$k$ routing is compelled by theoretical and technical considerations, and stands in contrast to the classification setting studied in Chen

et al. (2022); Chowdhury et al. (2023). This phenomenon arises specifically from the challenge of estimating a continuous function in the regression setting.

For each expert, we consider a two-layer neural network $f_m(x; W_m) = \frac{1}{J}\sum_{j=1}^{J} a_{m,j}\sigma_m(w_{m,j}^\top x + b_{m,j})$. The Hermite expansions are given as $a_{m,j}\sigma_m(z + w_{m,j}^\top v_c + b_{m,j}) = \sum_{i=0}^{\infty} \frac{\alpha_{m,j,i,c}}{\sqrt{i!}}\mathrm{He}_i(z)$.

Moreover, we assume that the activation function $\sigma_m$ satisfies the following assumption used in Oko et al. (2024a) to ensure that a fraction of neurons can align with the feature index even though the target functions are unknown.

**Assumption 3.4** (Student Activation Functions). The activation functions $\sigma_m$ of the student model and the link function $f_c^*$ and $g^*$ of the teacher models satisfy one of the following conditions: (A) $\sigma_m$ is a randomized polynomial activation of degree at most $O(1)$ as defined in Appendix A.3 and $f_c^*$ satisfies Assumption 3.2 for all $c = 1, \ldots, C$, or (B) $\sigma_m$ is the ReLU activation function, with the additional requirement that for each $f_i^*$, the absolute values of the all non-zero Hermite coefficients $|\alpha_{m,j,i}|$ are $\Theta(1)$. In addition to the conditions (A) and (B), we technically assume that the sign the Hermite coefficient $\alpha_{m,j,i,c}$ of $\sigma_m(\cdot + \rho w_{m,j}^t {}^\top v_c + b_{m,j})$ is invariant during the optimization since we cannot evaluate the contribution of higher-order Hermite coefficients.

### 3.3. Training Algorithm

To precisely track the model's evolution, we divide the training algorithm into layer-by-layer, similar to previous studies that have researched feature learning in neural networks (Damian et al., 2022; Ba et al., 2022; Bietti et al., 2023; Abbe et al., 2023; Mousavi-Hosseini et al., 2023b; Oko et al., 2024a; Lee et al., 2024). Specifically, we consider an algorithm separated into four phases. See Algorithm 1 for the outline.

Concretely, we start by initializing the weights of the experts as $w_{m,j}^0 \sim \mathrm{Unif}(\mathbb{S}^{d-1})$ and $a_{m,j} \sim \mathrm{Unif}\{-1, +1\}$ following Oko et al. (2024a) and Lee et al. (2024) and the weights of the router's gating network $\theta_m$ to zero. In Phase I, the first layer of the expert is optimized using a correlation loss, a technique supported by prior studies (Bietti et al., 2022; Damian et al., 2022; Abbe et al., 2023; Oko et al., 2024a; Lee et al., 2024). For optimization, the spherical gradient, defined as $\nabla_{w_{m,j}}^S \mathcal{L} := (I_d - w_{m,j}w_{m,j}^\top)\nabla_{w_{m,j}}\mathcal{L}$, is employed, as explored in Arous et al. (2021); Damian et al. (2023); Oko et al. (2024a); Lee et al. (2024). Phase II is devoted to the router's gating network which is trained via gradient descent. In this stage, we train the gating network in a single step with a batch size of $n = \tilde{\Theta}(d)$. Since the gradient of the gating network $h(x; \Theta)$ contains differences in the alignment $w_{m,j}^\top w_c^*$ between experts after the exploration stage, it becomes aligned with the cluster mean vector

---

**Algorithm 1** Gradient-based training of MoE

**Require:** Learning rates $\eta^t$, regularization parameter $\lambda$, sample sizes $T_1, T_2, T_3, T_4$, initialization scale $C_b$. Initialize $w_{m,j}^0 \sim \mathrm{Unif}(\mathbb{S}^{d-1}(1))$ and $a_{m,j} \sim \mathrm{Unif}\{\pm 1\}$.

**Phase I: Normalized SGD on first-layer of experts**
**for** $t = 0$ to $T_1 - 1$ **do**
    Draw new sample $(x_c^t, y_c^t)$.
    $w_{m,j}^{t+1} \leftarrow w_{m,j}^t + \eta^t y_c^t \nabla_{w_{m,j}}^S F_1(x_c^t)$.
    $w_{m,j}^{t+1} \leftarrow w_{m,j}^{t+1}/\|w_{m,j}^{t+1}\|$ for $j = 1, \ldots, J$.
**end for**

**Phase II: SGD on gating network of router**
**for** $t = T_1$ to $T_1 + T_2 - 1(= T_1)$ **do**
    Draw new samples $(x_c^i, y_c^i)_{i=1}^n$.
    $\theta_m^{t+1} \leftarrow \theta_m^t + \frac{\eta^t}{n}\sum_{i=1}^n y_c^i \nabla_{\theta_m} F_1(x_c^i)$.
**end for**

**Phase III: Normalized SGD on first-layer of experts**
Reinitialize $w_{m,j}^0 \sim \mathrm{Unif}(\mathbb{S}^{d-1}(1))$ and $a_{m,j} \sim \mathrm{Unif}\{\pm 1\}$
**for** $t = T_1 + T_2$ to $T_1 + T_2 + T_3 - 1$ **do**
    Draw new sample $(x_c^t, y_c^t)$.
    $w_{m,j}^{t+1} \leftarrow w_{m,j}^t + \eta^t y_c^t \nabla_{w_{m,j}}^S \hat{F}_M(x_c^t)$.
    $w_{m,j}^{t+1} \leftarrow w_{m,j}^{t+1}/\|w_{m,j}^{t+1}\|$ for $j = 1, \ldots, J$.
**end for**

**Phase IV: Convex optimization for second-layer of experts**
Initialize $b_{m,j} \sim \mathrm{Unif}([-C_b, C_b])$ and set $\hat{w}_j \leftarrow \delta_{m,j}w_{m,j}^{T_1+T_2+T_3}$, where $\delta_{m,j} \sim \mathrm{Unif}\{\pm 1\}$.
Draw new samples $(x_c^t, y_c^t)_{t=T_1+T_2+T_3}^{T_1+T_2+T_3+T_4-1}$.
Solve:

$$\{\hat{a}_m\}_m \leftarrow \underset{a_m \in \mathbb{R}^J}{\mathrm{argmin}}\frac{1}{T_4}\sum_{t=\tau_5}^{\tau_6-1}\left(\hat{F}_M(x^t) - y^t\right)^2 + \bar{\lambda}\sum_m \|a_m\|_2^2$$

where $\tau_5 = \sum_{i=1}^3 T_i$ and $\tau_6 = \sum_{i=1}^4 T_i$.

---

$v_c$. Before entering Phase III, the expert weights $w_{m,j}$ and $a_{m,j}$ are reinitialized. While not strictly necessary, this reinitialization helps ensure that the early learning of $w_c^*$ does not interfere with or disrupt the effective learning of $w_g^*$, particularly when $f^*$ and $g^*$ are similar functions. Note that if the activation function is ReLU, a random sign flip of $w_{m,j}$, as described in Oko et al. (2024a), becomes necessary. See Appendix A.3 for details. Finally, we conclude the training with Phase IV, where convex optimization with $L_2$-regularization is executed to the second layer using noise terms $b_{m,j}$ to facilitate the estimation of the polynomial of the ReLU activation function. Note that we use different routing strategy for Phases I and II ($F_1$) and Phases III and IV ($\hat{F}_M$). Refer to Remark 4.12 for details.

We will show that, at the end of each phase, a specific repre-

sentation of the complex task under consideration is learned, which is possible thanks to the idiosyncratic architecture of a MoE. More precisely, at the end of Phase I, some neurons within each expert *weakly recover* certain vectors and specialize to the corresponding cluster. In the next stage, the router learns to successfully dispatch data to the appropriate expert based on the weak recovery of clusters assigned to each expert. As for the second half of the algorithm, we prove that each expert successfully recovers both the local task and the global task associated with its assigned cluster.

## 4. Main Results

In this section, we provide our main results. We first prove that a vanilla neural network fails in detecting the latent structure of our task as it can only process the problem as a whole. Next, we show that the MoE, on the contrary, succeeds in dividing this problem into easier subproblems by leveraging the ability of each expert to weakly recover the simpler function corresponding to an individual cluster.

### 4.1. Limitations of the Vanilla Neural Network

#### 4.1.1. MAIN THEOREM

Here, we consider the vanilla neural network

$$f_m(x; W_m) = \frac{1}{J} \sum_{j=1}^{J} a_{m,j} \sigma_m(w_{m,j}^\top x + b_{m,j})$$

as the student model. This also corresponds to the special case of a MoE with only one expert. The size of the vanilla neural network is at most $J = O(\text{poly} d)$.

We will demonstrate that there are some scenarios of our teacher model that a single expert cannot solve. For example, consider the following. See Appendix B for further details.

**Example 4.1.** We construct a specific problem of our teach model 3.2 as follows. Assume all feature vectors $w_c^*$ and $w_g^*$ are completely orthogonal for simplicity. Moreover, functions are defined as $f_c^*(w_c^{*\top} x_c) = \beta_{c,k^*} \text{He}_{k^*}(w_c^{*\top} x_c)$, for all $c$, and $s_c g^*(x_c) = (-1)^{c+1} \text{He}_{k^*}(w_g^{*\top} x_c)$ for $c = 1, 2$ and otherwise 0. We assume that there exists at least one pair $(c, c')$ such that $\text{sgn } \beta_{c,k^*} \neq \text{sgn } \beta_{c',k^*}$. We additionally assume $k^* \geq 5$ to prove the result in Lemma B.15. $(s_c)_c$ are defined as $s_1 = 1$, $s_2 = -1$ and $s_c = 0$ (otherwise), to satisfy $\sum_c s_c = 0$. This means the signal $w_g^*$ is hard to recover. In short, the data $y_c$ are generated as

$$\begin{cases} \beta_{1,k^*} \text{He}_{k^*}(w_1^{*\top} x_c) + \text{He}_{k^*}(w_g^{*\top} x_c) + \nu & \text{if } c = 1, \\ \beta_{2,k^*} \text{He}_{k^*}(w_2^{*\top} x_c) - \text{He}_{k^*}(w_g^{*\top} x_c) + \nu & \text{if } c = 2, \\ \beta_{c,k^*} \text{He}_{k^*}(w_c^{*\top} x_c) + \nu & \text{if } c > 2. \end{cases}$$

*Remark* 4.2. It may be possible that $\text{sgn } \alpha_{m,j,k^*,c} \neq \text{sgn } \alpha_{m,j',k^*,c'}$ when $j \neq j'$ because we randomly initialize $a_{mj}$ and $b_{m,j}$.

The next theorem shows that $w_{m,j}^t$ never catches the signal $w_g^*$ for all $t$ because the gradient for $w_g^*$ is erased by those for $w_c^*$.

**Theorem 4.3** (Difficulty of finding the "hidden" signal $w_g^*$). *During the population spherical gradient flow of $w_{mj}^t$, for all $j = 1, \ldots, J$, we have*

$$\sup_{t \geq 0} |w_{m,j}^{t\top} w_g^*| \lesssim \tilde{O}(d^{-1/2})$$

*with high probability.*

Theorem 4.3 indicates that there is an insufficient number of neurons that can align the feature vector $w_g^*$. As a result, it becomes difficult to estimate the function $s_c g^*(w_g^{*\top} \cdot)$. This phenomenon is due to the condition $\sum_c s_c = 0$ and that the naive neural network (with polynomial width) cannot utilize the vectors $v_c$ to detect the cluster structure. Interestingly, such difficulty of SGD for the naive neural network has not been shown in prior works studying the optimization dynamics of the MoE (Chen et al., 2022; Chowdhury et al., 2023; Li et al., 2024). This was possible thanks to our theoretical analysis based on the information exponent, which appears in the context of nonlinear regression.

#### 4.1.2. PROOF SKETCH

We provide a sketch of the proof for the theorem. We demonstrate that a vanilla neural network predominantly aligns with simple tasks $w_c^*$, which prevents it from aligning with the more subtle one $w_g^*$. For a comprehensive explanation, please check Appendix B.

The spherical gradient flows of $|w_{m,j}^{t\top} w_c^*|$ are approximately evaluated as

$$\frac{d}{dt} |w_{m,j}^{t\top} w_c^*| \simeq \tilde{\Theta}(\eta J^{-1} |w_{m,j}^{t\top} w_c^*|^{k^*-1})$$

where $\eta$ is a learning rate, under the condition that $\beta_{c,k^*} \alpha_{m,j,k^*,c} > 0$. By integration, it takes $\tilde{\Theta}(\eta^{-1} J d^{(k^*-2)/2})$ time for the weak recovery.

Now, the most important observation is that the signals of $w_g^*$ are canceled out:

**Lemma 4.4.** *Recall that the inputs are generated as $x_c = \rho v_c + z$, where $z \sim \mathcal{N}(0, I_d)$. The Hermite coefficients $\alpha_{m,j,i,c}$ of $\sigma_m(\cdot + w_{m,j}^\top v_c + b_{mj})$ are close up to $\tilde{O}(d^{-1/2})$ at the initialization. Please refer to Lemma B.4 for the proof.*

Therefore, when initialized as $|w_{m,j}^{0\top} w_g^*| \simeq d^{-1/2}$,

$$\frac{d}{dt} |w_{m,j}^{t\top} w_g^*|$$
$$\lesssim \eta J^{-1} \left| \sum_c s_c \alpha_{m,j,k^*,c} \right| |w_{m,j}^{t\top} w_g^*|^{k^*-1}$$

$$\lesssim \eta J^{-1} d^{-1/2} |{w_{m,j}^t}^\top w_g^*|^{k^*-1}$$

$$\lesssim \eta J^{-1} |{w_{m,j}^t}^\top w_g^*|^{(k^*+1)-1},$$

which implies that the information exponent of $g^*$ increases. It takes at least $\tilde{\Omega}(\eta^{-1} J d^{((k^*+1)-2)/2})$ time for the weak recovery. Therefore, for all $j$, detecting one of the signals $\{w_c^*\}_c$ becomes easier than $w_g^*$ and each $w_{m,j}$ tends to align with $w_c^*$ rather than $w_g^*$.

Next, we identify the subset of tasks $\mathcal{C}_j$ from the entire set $[C]$ that neuron $w_{mj}$ can align with based on the necessary condition: for all $j$, there exists $c$ such that $\beta_{c,k^*} \alpha_{m,j,k^*,c} > 0$ with high probability. Let $\mathcal{C}_j$ be such a subset of $C$, then we can show that $w_{m,j}$ can only detect $w_c^*$ where $c \in \mathcal{C}_j$.

Based on the above argument, we obtain that for all $j$, $w_{m,j}$ aligns some feature vectors among $\{w_c^*\}_c$, not hidden $w_g^*$:

**Lemma 4.5.** *For all $j$, there exists $t_\Delta \lesssim \tilde{O}(J\eta^{-1} d^{(k^*-2)/2})$ such that for all $t \geq t_\Delta$,*

1. $|{w_{m,j}^t}^\top w_c^*| \gtrsim \tilde{\Omega}(d^{-1/4-1/(8k^*)})$ *for some $c \in \mathcal{C}_j$,*

2. $|{w_{m,j}^t}^\top w_c^*| \lesssim \tilde{O}(d^{-1/2})$ *for all $c \notin \mathcal{C}_j$,*

3. $|{w_{m,j}^t}^\top w_g^*| \lesssim \tilde{O}(d^{-1/2})$

*hold with high probability.*

See Lemma B.15 for formal proof. Intuitively, when the inequalities in Lemma 4.5 hold, then the alignment in the following inequality does not grow because the derivative continues to be negative: Let $\xi_{m,j,g}^t := {w_{m,j}^t}^\top w_g^*$ and $\kappa_{m,j,c}^t := {w_{m,j}^t}^\top w_c^*$. For all $t' \in [t_\Delta, t]$,

$$\frac{d}{dt}(|\xi_{m,j,g}^t| + |\kappa_{m,j,c}^t|)|_{t=t'}$$

$$\lesssim \frac{\eta}{CJ} \tilde{O}(|\xi_{m,j,g}^{t'}|^{k^*-1}$$

$$+ \sum_{c \notin \mathcal{C}_j} \underbrace{(-\beta_{c,k^*}\alpha_{m,j,k^*c})}_{\leq \tilde{O}(1)} (\kappa_{m,j,c}^{t'})^{k^*} |\xi_{m,j,g}^{t'}|$$

$$- \sum_{c \in \mathcal{C}_j} \underbrace{\beta_{c,k^*}\alpha_{m,j,k^*c}}_{\geq \tilde{\Omega}(1)} \underbrace{(\kappa_{m,j,c}^{t'})^{k^*}}_{\gtrsim d^{-k^*(1/4+1/(8k^*))}} |\xi_{m,j,g}^{t'}|)$$

$$\lesssim \frac{\eta}{CJ} |\xi_{m,j,g}^{t'}| \tilde{O}\left(d^{-(k^*-2)/2} - \tilde{\Omega}(d^{-k^*/4-1/8})\right)$$

$$\lesssim 0,$$

where we used the additional assumption $k^* \geq 5$, and the definition of $\mathcal{C}_j$. Therefore, the alignment $|{w_{m,j}^t}^\top w_g^*| + \sum_{c \notin \mathcal{C}_j} |{w_{m,j}^t}^\top w_c^*|$ is bounded by $\tilde{O}(d^{-1/2})$ for all $t$. See Theorem B.17 in Appendix for more rigorous discussions

## 4.2. Learning Dynamics of MoE

### 4.2.1. MAIN THEOREM

On the contrary, the MoE successfully learns the teacher model defined in Assumption 3.2 by enabling the router to appropriately partition the data among the teacher models for each cluster. This is stated formally in the following theorem. We further characterize the sample complexity of this learning process under Algorithm 1.

**Theorem 4.6.** *Under Assumptions 3.2, 3.3, and 3.4, set $J = O(\epsilon^{-1})$ as the number of neurons, $T_1 = \tilde{\Theta}(d^{k^*-1})$ as the number of training steps for Phase I, $T_2 = 1$ as the number of training steps and $n = \tilde{\Theta}(d)$ as the batch size for Phase II, $T_3 = \tilde{\Theta}(d^{k^*-1} \vee d\epsilon^{-2} \vee \epsilon^{-3})$ as the number of training steps for Phase III, and $T_4 = \tilde{\Theta}(\epsilon^{-2})$ as the number of training steps for Phase IV. Then, under the suitable choices of $\eta^t$ and $\lambda$, with probability at least $0.99$ over the randomness of the dataset and initialization,*

$$\mathbb{E}_{x_c}\left[|\hat{F}_M(x_c; \{\hat{a}_m\}_{m=1}^M) - f_c^*(x_c) - s_c g^*(x_c)|\right] \leq \epsilon.$$

We considered the case where each cluster possesses its own single-index model while collectively sharing a global single-index model across all clusters. This global task induces interference, which attenuates the signal of the shared model. This setting is potentially difficult for a vanilla neural network to learn as shown in Subsection 4.1. The total sample complexity is $\tilde{O}(d^{k^*-1})$ and the time complexity is polynomial in $d$. This complexity is the same as learning single-index model by a vanilla neural network (Arous et al., 2021) while kernel ridge regression requires $\tilde{O}(d^{p^*})$ (Ghorbani et al., 2021; Donhauser et al., 2021) with respect to $d$. After the weak recovery of Phase I, the router successfully divides the clusters and enables the expert to learn their target functions.

**Experiments** To illustrate the dynamics of the MoE following Algorithm 1, we focus on a synthetic problem where $C = 2$ in the problem setting of Assumption 3.2. We define $f_1^* = \text{He}_3 + \text{He}_5$ and $f_2^* = \text{He}_3 + \text{He}_4$ for the local tasks, and $g^* = \text{He}_3$ for the global task. The vectors $w_1^*, w_2^*$ and $w_g^*$ were of dimension 200, generated randomly and applied Gram-Schmidt orthogonalization to satisfy Assumption 3.3. As for the student model, the number of experts was set to 8, and the hidden dimension of each expert to 500. The learning rate was set to 1 for all optimization schemes, and $T_1 = 3.5 \times 10^6, T_2 = 300, T_3 = 10^7$.

The alignments of the experts and router at the end of Phase I, II and III are shown in Figures 1 and 2. As we can observe, in Phase I, differences among experts arise due to initialization, resulting in variations in the degree of weak recovery for local tasks. In Phase II, the router leverages these differences in recovery, which are reflected in the

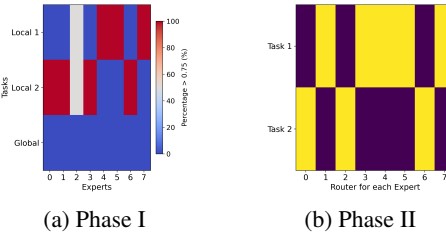

| (a) Phase I | (b) Phase II |

*Figure 1.* The alignment of the experts after Phase I (a) and router after Phase II (b) with the respective feature vectors of each task. In Figure (a), the alignment of $w_{m,j}$ and $w_c^*$ ($c = 1, \ldots, C$) or $w_g^*$ (vertical axis) is computed, and for each expert, the distribution of the number of $w_{m,j}$ with larger alignment than $\max_{j,c} w_{m,j}^\top w_c^*$ is reported. In Figure (b), we visualize for each router $h_m$ the task with the best alignment in yellow.

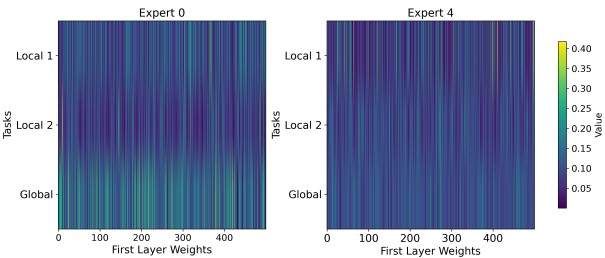

*Figure 2.* The alignment of two experts after Phase III. The alignment of $w_{m,j}$ (horizontal axis) and $w_c^*$ ($c = 1, \ldots, C$) (vertical axis) is computed. The last row is the alignment between $w_{m,j}$ and $w_g^*$.

gradients, as a signal to learn to dispatch the data from each cluster to the corresponding expert. In Phase III, once the router has learned to appropriately allocate the data, each expert can effectively learn both its assigned local task and the global task without signal interference across clusters.

### 4.2.2. PROOF SKETCH

In this section, we will provide an overview of how the MoE can detect and learn the latent cluster structure using population gradient flow, and how our intuition can be extended to the SGD. We proceed in five steps: initialization, Phase I, Phase II, Phase III, and Phase IV. Please refer to Appendix C for further details in empirical and discretized dynamics.

**Initialization.** At the initial state, experts are divided based on the task of the cluster with which they exhibit the highest alignment. We define (for each task) an expert that will eventually specialize in that task as follows:

**Definition 4.7** (The set of the professional experts for class $c$).

$$(j_m^*, c_m^*) := \operatorname{argmax}_{j,c} {w_{m,j}^0}^\top w_c^*,$$

$$\mathcal{M}_c := \{m \mid c = c_m^*\}.$$

${w_{m,j_m^*}^t}^\top w_{c_m^*}^*$ has a larger value by a constant factor with probability at least 0.999, which divides the experts into the exclusive subsets that are specialized to each cluster $c$:

**Lemma 4.8** (Following Chen et al. (2022); Oko et al. (2024a)). *If $M \gtrsim C \log C$, it holds that*

$$\mathbb{P}[|\mathcal{M}_c| \geq 1, \ \forall c] \geq 0.999.$$

*For all $m$, if $\sqrt{\log d} \gtrsim J \gtrsim C^{-1} \log M$, there are one neuron $w_{mj_m^*}$*

$${w_{mj}}^\top w_{c_m^*}^* \gtrsim \max_{c' \neq c_m^* \text{ or } j' \neq j_m^*} |{w_{mj'}}^\top w_{c'}^*| + \tilde{\Omega}(d^{-1/2}),$$

*with probability at least $0.999$.*

At the initialization, the inner products only differ by a constant. However, when two sequences have initial values that differ by a constant factor, this can cause differences in their growth rates, ultimately placing them in different asymptotic orders. Such a technique has been employed in various contexts (Ben Arous et al., 2022; Chen et al., 2022; Oko et al., 2024a). See Appendix C.1 for details.

**Phase I (Exploration Stage).** From this phase, we will take for granted that the conditions of Lemma 4.8 are satisfied and will use the term *with high probability* withing this scenario (i.e., conditional probability). In the exploration stage, one of the neurons in each cluster achieves faster weak recovery for its assigned cluster compared to other neurons, due to the alignment differences introduced during initialization. Now, for $m \in \mathcal{M}_c$, each ${w_{m,j}^t}^\top w_c^*$ follows a gradient flow as

$$\frac{\mathrm{d}}{\mathrm{d}t} |{w_{m,j}^t}^\top w_c^*| \simeq \eta |{w_{m,j}^t}^\top w_c^*|^{k^*-1}.$$

Then we have the following result:

**Lemma 4.9** (Informal). *For all $m \in \mathcal{M}_c$, there exists some time $t_1 \leq T_1 = \tilde{O}(d^{k^*-1})$ such that*

1. $|{w_{m,j_m^*}^{t_1}}^\top w_c^*| = \tilde{\Omega}(1)$,

2. $|{w_{m,j'}^{t_1}}^\top w_{c'}^*| = \tilde{O}(d^{-1/2})$ for all $(c', j') \neq (c_m^*, j_m^*)$,

3. $|{w_{m,j}^{t_1}}^\top w_g^*| = \tilde{O}(d^{-1/2})$ for all $j$.

Lemma 4.9 shows that the expert $m \in \mathcal{M}_c$ weakly specialize to the cluster $c$, enabling the router to identify experts via weak recovery. This result highlights that, in order for the router to effectively distinguish among experts, a weak recovery of the feature index is required. This, in turn, implies that a sample complexity of $\tilde{\Theta}(d^{k^*-1})$ may be required during the exploration phase, implying that a sufficiently long exploration stage is warranted before the router can engage

in meaningful learning. This contrasts with the linear expert setting of Li et al. (2024) and the classification framework in Chen et al. (2022); Chowdhury et al. (2023), as this finding is rooted in non-convex optimization in linear regression. To prove Lemma 4.9, we leverage the information exponent of the teacher models instead of using the cubic activation in Chen et al. (2022). Compared to the results for additive models in Oko et al. (2024a), we evaluated the growth of $w_{m,j}^t{}^\top w_c^*$ for all $j$.

**Phase II (Router Learning Stage).** Here, we discuss how the router extracts the cluster mean vector $v_c$ corresponding the cluster $c$ from the weak recovery of the experts. We show that the parameters $\theta_m$, for some $m \in M_c$, become positively correlated with $v_c$, while, on the other hand, $\theta_{m'}$, for all $m' \notin M_c$, become negatively correlated with it. This is enabled by the fact that the gradients of the gating network encode informative signals elicited by the weak recovery of the experts.

**Lemma 4.10.** *For all $c$, $m_c^* := \operatorname{argmax}_m h_m(v_c) \in \mathcal{M}_c$ and $m' \notin \mathcal{M}_c$,*

$$\theta_{m'}^{T_1+T_2}{}^\top v_c \le -\tilde{\Omega}(1) \le 0 \le \tilde{\Omega}(1) \le \theta_{m_c^*}^{T_1+T_2}{}^\top v_c.$$

*Proof.* (Sketch). Take $m \notin \mathcal{M}_c$. The population gradient for the gating network of the router is evaluated as

$$
\begin{aligned}
&- v_c^\top \nabla_{\theta_m} \mathbb{E}[\mathcal{L}] \\
&\simeq -\tilde{\Omega}\Big( \sum_{m'' \in \mathcal{M}_c, \forall j} \underbrace{|w_{m'',j}^t{}^\top w_c^*|^{k^*}}_{=\tilde{\Omega}(1), \text{ due to weak recovery}} \Big) \\
&\quad + \tilde{O}\Big( \sum_{m' \notin \mathcal{M}_c, \forall j, c} \underbrace{|w_{m',j}^t{}^\top w_c^*|^{k^*}}_{\simeq d^{-k^*/2}} \Big) \\
&\simeq -\tilde{\Omega}(1).
\end{aligned}
$$

Therefore, $v_c^\top \theta_m^{T_1+T_2} < -\tilde{\Omega}(1)$ and lastly we use $\sum_m \theta_m = 0$ to bound $h_{m_c^*}(x_c) = \theta_{m_c^*}^\top x_c$. $\qquad\square$

This lemma implies that, for $x_c = \rho v_c + z$,

$$
h_m(x_c) \begin{cases} \ge 0 & \text{if} \quad m = m_c^*, \\ < 0 & \text{if} \quad m \notin \mathcal{M}_c \end{cases}
$$

with high probability based on the assumption that $\rho = \operatorname{poly}\log d$ is sufficiently large. Interestingly, the concept of the information exponent and the weak recovery had essential roles in the router learning.

*Remark* 4.11. In Chen et al. (2022) and Li et al. (2024), the norm of the cluster signal $\rho v_c$ is as large as the norm of the noise independent of $v_c$. However, in our setting, $\|\rho v_c\|_2 = \operatorname{poly}\log d$ and $\|z\|_2 \simeq d^{1/2} \gg \|\rho v_c\|_2$ with high probability. Due to this setup, we had to employ a

much more subtle argument than theirs. Specifically, we carefully bounded $\|\nabla_m \mathbb{E}[\mathcal{L}]\|_2$ and $\theta_m^\top z = \tilde{O}(\|\theta_m\|_2 d^{1/2})$. Using Stein's lemma, $\|\nabla_m \mathbb{E}[\mathcal{L}]\|_2$ is bounded as $\tilde{O}(1)$.

*Remark* 4.12. We use different router algorithms in Phases I and II compared to Phases III and IV because the size of the set $\mathcal{M}_c$ is not fixed. Since there is a variation in $|\mathcal{M}_c|$ for each cluster $c$, employing a fixed-$k$ top-$k$ algorithm may fail in routing the data to the corresponding experts. On the one hand, if we set the $k$ of top-$k$ as $k > |\mathcal{M}_c|$ for some $c$, there might be some $c' \ne c$ such that the corresponding input $x_c$ is routed to $m \in \mathcal{M}_{c'}$. On the other hand, if we have the $k$ of top-$k$ as $k < |\mathcal{M}_c|$ for some $c$, then there may be no expert in the corresponding set $\mathcal{M}_c$ that is always selected (routed) when $x_c$ arrives.

**Phase III (Expert Learning Stage).** In this phase, as the router has learned to dispatch data appropriately, each expert receives and trains only on its designated cluster. Each expert first weakly recovers and then strongly recovers both the local and global tasks of its corresponding cluster. At this point, there exists at least one $m \in \mathcal{M}_c$ such that $h_m(x_c) \ge 0$ and for all $m \notin \mathcal{M}_c$, $h_m(x_c) < 0$ with high probability. Therefore, the teacher polynomials $s_c g^*(w_g^*{}^\top \cdot)$, where $\sum_c s_c = 0$, are successfully decomposed into $\tilde{\Omega}(1)$ functions and it enables the experts to learn $w_g^*$ and $g^*$. As for the MoE model, when the input $x_c$ is from the cluster $c$, the MoE model

$$\hat{F}_M(x_c; \{\hat{a}_m\}_{m=1}^M) := \sum_{m=1}^M \mathbb{1}[h_m(x_c) \ge 0] f_m(x_c)$$

is equivalent to

$$\hat{F}_{\mathcal{M}_c}(x_c; \{\hat{a}_m\}_{m \in \mathcal{M}_c}) := \sum_{m \in \mathcal{M}_c} \mathbb{1}[h_m(x_c) \ge 0] f_m(x_c)$$

with high probability. Thus, the MoE model was decomposed into $\{\hat{F}_{\mathcal{M}_c}\}_c$ which do not share the parameters because $\mathcal{M}_c \cap \mathcal{M}_{c'} = \emptyset$, $\forall c \ne c'$. Additionally, using $h_{m_c^*}(x_c) \ge 0$ with high probability where $m_c^* = \operatorname{argmax}_m h_m(v_c) \in \mathcal{M}_c$, it holds that

$$\nabla_{w_{m_c^*,j}} \hat{F}_{\mathcal{M}_c}(x_c; \{\hat{a}_m\}_{m \in \mathcal{M}_c}) = \nabla_{w_{m_c^*,j}} f_{m_c^*}(x_c; a_m)$$

with high probability. Hence, Phase III can be completely decomposed into the subproblem of the weak (to strong) recovery of $w_{m_c^*,j}^\top w_c^*$ and $w_{m_c^*,j}^\top w_g^*$ given the inputs $\{x_c^{t_c}\}_{t_c}$ in each cluster $c$. We show the strong recovery of neurons, in parallel with Oko et al. (2024a).

**Phase IV (Second Layer Optimization Stage).** In Phase IV, the experts with aligned vectors estimate the link functions $f_c^*$ and $g^*$ through second-layer optimization.

First, with some expert $m \in \mathcal{M}_c$ and $w_{mj} \simeq w_c^*$ and $w_{mj'} \simeq w_g^*$, we construct $\hat{a}_m$ such that

$$f_m(x_c) \simeq f_c^*(w_c^*{}^\top x_c) + s_c g^*(w_g^*{}^\top x_c)$$

as a feasible solution.

Next, we decompose the whole convex optimization problem into $c$ individual subproblems that do not share the experts to employ the results in the standard analysis for additive models in prior work (Oko et al., 2024a).

## 5. Conclusion

In this paper, we theoretically showed that a MoE can learn the latent cluster structure of a problem with a sample complexity that depends not on the information exponent of the whole task but on the local information exponent of each cluster. In addition, we have demonstrated that the vanilla neural network with polynomial time complexity fails to detect such a structure. While this work contributes to the further understanding of the underlying mechanism of MoE and its success, it is still unknown whether the MoE architecture is indeed effective to pursue the information-theoretic limit. We believe this constitutes a promising direction for future work.

**Implications and Future Directions** Our findings offer several insights for designing more effective MoE architectures. First, while our analysis demonstrates that MoEs mitigate gradient interference through explicit partitioning, the number of experts is typically chosen heuristically in practice. This raises the possibility that incorporating gradient-aware routing mechanisms could lead to more principled and efficient expert allocation strategies, as recently explored in Liu et al. (2024); Yang et al. (2025). Second, to prevent competition among professional experts, we employed top-$k$ routing to reduce potential load imbalance. This motivates the design of adaptive routing schemes that dynamically adjust $k$ during training—a perspective supported by our theoretical analysis in nonlinear regression and recent findings in NLP that adapt $k$ per token (Huang et al., 2024; Zeng et al., 2024). Third, freezing or pruning redundant experts may further alleviate competition and reduce deployment cost, aligning with recent proposals on expert merging (Zhang et al., 2024a).

Beyond architectural design, our analysis also informs the training process of MoE systems. In particular, we showed that learning a meaningful router relies on observable differences in the experts' weak recovery, which in turn requires a sufficiently long exploration stage due to the non-convex nature of the objective. This suggests that upcycling dense checkpoints pretrained on diverse domains may offer a practical means of accelerating convergence—an approach that has gained traction in recent large language models (Komatsuzaki et al., 2023; Wei et al., 2024). Finally, our analysis highlights that different phases of training pose distinct challenges. Specifically, the noise introduced during Phase II serves to ensure uniform gradient flow and provide suffi-

cient learning signals for all experts, whereas the adaptive top-$k$ routing employed in Phases III and IV is designed to mitigate competition among professional experts. These observations point to the potential of stage-specific routing strategies tailored to the evolving dynamics of MoE training.

## Impact Statement

This paper presents work whose goal is to advance the field of Machine Learning. There are many potential societal consequences of our work, none which we feel must be specifically highlighted here.

## Acknowledgment

RK and NN were supported by the FY 2024 Self-directed Research Activity Grant of the University of Tokyo's International Graduate Program "Innovation for Intelligent World" (IIW). KM was partially supported by JST CREST (JPMJCR2015). NN was partially supported by JST ACT-X (JPMJAX24CK) and JST BOOST (JPMJBS2418). TS was partially supported by JSPS KAKENHI (24K02905) and JST CREST (JPMJCR2115). YK was supported by JST BOOST, Japan Grant Number JPMJBS2418. This research is supported by the National Research Foundation, Singapore and the Ministry of Digital Development and Information under the AI Visiting Professorship Programme (award number AIVP-2024-004). Any opinions, findings and conclusions or recommendations expressed in this material are those of the author(s) and do not reflect the views of National Research Foundation, Singapore and the Ministry of Digital Development and Information.

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

# A. Preliminaries

## A.1. Hermite Polynomials

In this subsection, we present key properties of the probabilists' Hermite polynomials that are essential for analyzing functions under the Gaussian measure. For a more detailed treatment, we refer the reader to Section 11.2 of O'Donnell (2021).

Let $\mu$ be the standard Gaussian measure and $L^2(\mu)$ the corresponding square-integrable function space with respect to $\mu$. For $f, g \in L^2(\mu)$, the inner product is defined as $\langle f, g \rangle_\mu := \mathbb{E}_{z \sim \mu}[f(z)g(z)]$.

**Definition A.1.** The $i$th *Hermite polynomial* $\mathrm{He}_i : \mathbb{R} \to \mathbb{R}$, $i \in \mathbb{N}$ is defined as

$$\mathrm{He}_i(z) = (-1)^i \exp\left(\frac{z^2}{2}\right) \frac{\mathrm{d}^i}{\mathrm{d}z^i} \exp\left(-\frac{z^2}{2}\right).$$

**Lemma A.2.** *The normalized Hermite polynomials $\{\mathrm{He}_i/\sqrt{i!}\}_i$ form a complete orthonormal basis for $L^2(\mu)$.*

**Lemma A.3.** *The Hermite polynomials satisfy the following properties:*

1. ***Derivatives:***

$$\frac{\mathrm{d}}{\mathrm{d}z}\mathrm{He}_i(z) = i\mathrm{He}_{i-1}(z),$$

2. ***Integration by Parts:*** *For $f \in L^2(\mu)$ and $z, z' \sim \mathcal{N}(0, I_d)$ such that $\mathrm{Cov}(z, z') = \rho \in [-1, 1]$,*

$$\mathbb{E}_{z,z'}[\mathrm{He}_i(z)f(z')] = \rho\mathbb{E}_{z,z'}[\mathrm{He}_{i-1}(z)f'(z')],$$

3. ***Orthogonality:*** *For $z, z' \sim \mathcal{N}(0, I_d)$ such that $\mathrm{Cov}(z, z') = \rho \in [-1, 1]$,*

$$\mathbb{E}_{z,z'}[\mathrm{He}_i(z)\mathrm{He}_j(z')] = \begin{cases} (i!)\rho^i & if \quad i = j \\ 0 & otherwise, \end{cases}$$

4. ***Hermite expansion:*** *For $f \in L^2(\mu)$,*

$$f(z) \overset{L^2}{=} \sum_{i=0}^{\infty} \frac{\alpha_i}{i!}\mathrm{He}_i(z), \quad \alpha_i = \langle f, \mathrm{He}_i \rangle_\mu.$$

## A.2. Bihari-LaSalle Inequality and Gronwall Inequality.

In this subsection, we present the discrete version of Bihari-LaSalle Inequality and Gronwall Inequality, which serve as tools for analyzing the growth behavior of nonlinear recurrence relations. These inequalities will be used repeatedly throughout our analysis. The derivation is adapted from Ben Arous et al. (2022).

Let us consider the sequence $\{A_t\}_{t=0}^{\infty}$ defined as

$$A_{t+1} = A_t + B(A_t)^{k-1}$$

where $k > 3$ and $B > 0$. Then we have the following evaluations:

**Lemma A.4.** *We have*

$$A_t \geq \frac{A_0}{(1 - B(k-2)(A_0)^{k-2}t)^{\frac{1}{k-2}}}.$$

*Moreover, if $A_t \geq 1 \, \forall t \leq T$, we have*

$$A_t \leq \frac{A_0}{(1 - B(1+B)^{k-1}(k-2)(A_0)^{k-2}t)^{\frac{1}{k-2}}}.$$

Please note that if two sequences start off differing by a constant factor, their subsequent growth rates can diverge, leading them to differ in order of magnitude: Let us take two sequences as

$$A_{t+1} = A_t + B(A_t)^{k-1}, \quad A_0 = B_0 = o_d(1),$$

$$\tilde{A}_{t+1} = \tilde{A}_t + B(\tilde{A}_t)^{k-1}, \quad \tilde{A}_0 = \lambda B_0, \quad 0 < \lambda < \left(\frac{1}{1+B}\right)^{\frac{k-1}{k-2}}.$$

Then, it takes at most $B^{-1}(k-2)^{-1}(B_0)^{-(k-2)} =: t_1$ time to obtain $A_t \geq \Omega_d(1)$. Let $t_0 \leq t_1$ be the first time s.t. $A_t \geq \Omega_d(1)$. On the other hand,

$$\tilde{A}_{t_0} \leq \tilde{A}_{t_1} \leq \frac{\tilde{A}_0}{(1 - B(1+B)^{k-1}(k-2)(\lambda B_0)^{k-2}t_1)^{\frac{1}{k-2}}} \leq \frac{\tilde{A}_0}{\left(1 - \underbrace{(1+B)^{k-1}\lambda^{k-2}}_{<1}\right)^{\frac{1}{k-2}}} = o_d(1).$$

### A.3. Activation Functions

In this study, we consider the misspecified setting, where the target function and the activation function are different. However, in order to ensure the alignment between a neuron and a corresponding target task, we expect that the sign of the Hermite coefficients of the target function and the activation function to be the same. Remember that the Hermite expansion of the neuron $j$ is expressed as $a_{m,j}\sigma_m(w_{m,j}^\top z + b_{m,j}) = \sum_{i=0}^{\infty} \frac{\alpha_{m,j,i}}{\sqrt{i!}}\text{He}_i(w_{m,j}^\top z)$ and the Hermite expansion of the target function of cluster $c$ of the local task is expressed as $f_c^*(w_c^{*\top}z) = \sum_{i=k^*}^{p^*} \frac{\beta_{c,i}}{\sqrt{i!}}\text{He}_i(w_c^{*\top}z)$. We assume that at least a $\Omega(1)$ fraction of neurons ($j \in [J]$) satisfy $\alpha_{m,j,i}\beta_{c,i} \geq 0$ for $i = k^*$ and $\alpha_{m,j,i}\beta_{c,i} > 0$ for $k^* < i \leq p^*$. Similarly, for the global task $g$, we have $s_c\alpha_{m,j,i}\gamma_i \geq 0$ for $i = k^*$ and $s_c\alpha_{m,j,i}\gamma_i > 0$ for $k^* < i \leq p^*$. This condition is satisfied under certain activation functions.

For ReLU activations, the following lemma shows that $\alpha_{m,j,i}$ is positive for all $i$ with probability at least $\frac{1}{4}$ and the desired condition holds with probability at least $\frac{1}{8}$ over the randomness of the initialization of $a_{m,j}$.

**Lemma A.5** (Lemma 15 of Ba et al. (2023) and Lemma 17 of Oko et al. (2024a)). *Given degree $p^* \in \mathbb{N}$ and $b \sim [-C_b, C_b]$, the $i$-th Hermite coefficient of $\text{ReLU}(z + b_{m,j})$ is positive with probability $\frac{1}{4}$ for all $k^* \leq i \leq p^*$, if $C_b$ is larger than some constant that only depends on $p^*$.*

For polynomial functions, we randomize the activation functions as $\sigma_{m,j}(z) = \sum_{i=k^*}^{p^*} \frac{\epsilon_{i,j}}{\sqrt{i!}}\text{He}_i(z)$, where $\epsilon_{i,j}$ are independent Rademacher variables. The following lemma shows that the randomization of the activation functions ensure this condition.

**Lemma A.6** (Lemma 18 of Oko et al. (2024a)). *Given degree $p^* \in \mathbb{N}$ and $b \sim [-C_b, C_b]$, for each $k_{\min}^* \leq k^{*\prime} \leq k_{\max}^*$, the $i$-th Hermite coefficient of $a_{m,j}\sigma_m(z + b_{m,j})$ is non-zero with probability $\Omega(C_b^{-1})$, for all $k^{*\prime} \leq i \leq p^*$. Here, $\Omega$ hides constants only depending on $p^*$.*

## B. Proof of Limitations of the Vanilla Neural Network

In this chapter, we prove how spherical gradient descent using a standard neural network fails to learn some of the signals, introduced in Section 4.1. Here we have only one expert as

$$f_m(x; W_m) = \frac{1}{J}\sum_{j=1}^{J} a_{m,j}\sigma_m(w_{m,j}^\top x + b_{m,j}).$$

From here, we fix $m = 1$ and $j \in [1, \ldots, J]$. Let $\kappa_{m,j,c}^t = w_c^{*\top}w_{m,j}^t$ and $\xi_{m,j,g}^t = w_g^{*\top}w_{m,j}^t$. They represent alignment of the neuron $w_{m,j}^t$ with the signals $w_c^*$ and $w_g^*$ respectively.

**Definition B.1** (Restate). We consider the following teacher model:

- We have $C = O(1)$ classes and the strength of the cluster vector is $\rho = \text{poly}\log d$.

- All feature vectors are completely orthogonal.

- Additionally assume $k^* \geq 5$.

- Teacher models are defined as

  - $f_c^*(w_c^{*\top} x_c) = \beta_{c,k^*} \mathrm{He}_{k^*}(w_c^{*\top} x_c)$, for all $c \in [C]$,
  - $s_c g^*(x_c) = (-1)^{c+1} \mathrm{He}_{k^*}(w_g^{*\top} x_c)$ for $c = 1, 2$ and otherwise $0$,
  - $k^*$ is even.

  We assume that there exists at least one pair $(c, c')$ such that $\mathrm{sgn}\beta_{c,k^*} \neq \mathrm{sgn}\beta_{c'k^*}$.

- $s_c$ are set as

$$
s_c = \begin{cases} +1 & \text{if} \quad c = 1, \; j = 1, \\ -1 & \text{if} \quad c = 2, \; j = 1. \\ 0 & \quad \text{otherwise.} \end{cases}
$$

- In other words,

$$
y_c = \begin{cases} \beta_{1,k^*} \mathrm{He}_{k^*}(w_1^{*\top} x_c) + \mathrm{He}_{k^*}(w_g^{*\top} x_c) + \nu & \text{if} \quad c = 1, \\ \beta_{2,k^*} \mathrm{He}_{k^*}(w_2^{*\top} x_c) - \mathrm{He}_{k^*}(w_g^{*\top} x_c) + \nu & \text{if} \quad c = 2. \\ \beta_{c,k^*} \mathrm{He}_{k^*}(w_c^{*\top} x_c) + \nu & \text{if} \quad c > 2 \end{cases}
$$

- $|\alpha_{m,j,k^*}| = \Theta(1)$, where $\alpha_{m,j,k^*}$ is the $k^*$ th Hermite coefficient of $a_{m,j}\mathrm{ReLU}(\cdot + b_{m,j})$.

*Remark* B.2. We denote the $k^*$th Hermite coefficients of $\sigma_m(\cdot + w_{m,j}^\top v_c + b_{m,j})$ as $\alpha_{m,j,k^*,c}$. It may be possible that $\mathrm{sgn}\alpha_{m,j,k^*,c} \neq \mathrm{sgn}\alpha_{mj'k^*c}$ if $j \neq j'$ because we randomly initialize $a_{m,j}$ and $b_{m,j}$.

**Assumption B.3.** We assume that the size of one student to be at most $J = O(\mathrm{poly}d)$.

**Outline of the proof.** The outline of the proof is as follows:

1. We first show that

   - The Hermite coefficients corresponding to $\pm\mathrm{He}_{k^*}(w_g^{*\top} x_c)$ cancel out (Lemmas B.4 and C.8),
   - For all neurons $w_{m,j}$, there are some tasks $c \in [C]$ such that the signals of $w_c^*$ grow (Lemma B.7), the set of such $w_{m,j}$ is defined as $\mathcal{C}_j$.

2. For each $j$th neuron, the above points imply that there are three types of signals, as shown in Lemmas B.9 to B.11:

   (a) $w_c^*, c \in \mathcal{C}_j$: Learnable ($\frac{\mathrm{d}}{\mathrm{d}t}|\kappa_{m,j,c}^t|$ is positive),
   (b) $w_c^*, c \in [C] \setminus \mathcal{C}_j$: Not learnable ($\frac{\mathrm{d}}{\mathrm{d}t}|\kappa_{m,j,c}^t|$ is negative),
   (c) $w_g^*$: Not learnable (the growth rate of the product $w_j^\top w_g^*$ is too small compared to (a) because the Hermite coefficients cancel out (Lemma B.4)).

3. We show that all neurons tend to learn the features (a) $w_c^*, c \in \mathcal{C}_j$ (Lemma B.12).

4. In Lemma B.13, we repeat the argument in Lemma B.12 while keeping the condition of Hermite coefficients in Lemmas B.4 and B.5 until the products (a) become sufficiently large.

5. We finally show the growth of other products (b),(c) will be blocked (Lemma B.15) once the products corresponding to (a) become too large, additionally assuming $k^* \geq 5$

## B.1. Characterization of Hermite coefficients

Here we will show that the Hermite coefficients corresponding to $\pm\mathrm{He}_{k^*}(w_g^{*\top}x_c)$ cancel out. That is why $w_g^*$ is not learnable. Lemma C.8 informally implies that

$$\left|\frac{\mathrm{d}}{\mathrm{d}t}|w_j^{t\top}w_g^*|\right| \simeq \eta \left|(\underbrace{1\cdot\alpha_{m,j,k^*,1}}_{\text{from }\mathrm{He}_{k^*}(w_g^{*\top}x_1)} + \underbrace{(-1)\cdot\alpha_{m,j,k^*,2}}_{\text{from }-\mathrm{He}_{k^*}(w_g^{*\top}x_2)})(w_j^{t\top}w_g^*)^{k^*-1}\right|$$

$$\lesssim \eta\left|(\alpha_{m,j,k^*,1}-\alpha_{m,j,k^*,2})(w_j^{t\top}w_g^*)^{k^*-1}\right|$$

$$\lesssim \eta d^{-1/2}(w_j^{t\top}w_g^*)^{k^*-1}.$$

We see that the growth rate of $|\xi_{m,j,g}^t| = |w_j^{t\top}w_g^*|$ is small compared to $\kappa_{m,j,c}^t$ by a factor of $d^{-1/2}$. This results in the hardness of learning $w_g^*$ compared to $w_c^*$, $c \in \mathcal{C}_j$.

**Lemma B.4.** *At the initialization, the Hermite coefficients of $\sigma_m(\cdot + \rho w_{m,j}^\top v_c + b_{m,j})$ and $\sigma_m(\cdot + \rho w_{m,j}^\top v_{c'} + b_{m,j})$ are evaluated as*

$$|\alpha_{m,j,i,c} - \alpha_{mjic'}| \lesssim \rho\sqrt{\log d}/\sqrt{d}\ (\lesssim \tilde{O}(d^{-1/2}))$$

*by continuity, and $|\alpha_{m,j,k^*,c}| = \Theta(1)$ and $\mathrm{sgn}(\alpha_{m,j,c}) = \mathrm{sgn}(\alpha_{mjc'1})$ for all $c, c' \in [1, \ldots, C]$ with high probability over the randomness of the random initialization of $w_{m,j}$*

*Proof.* Remember that the inputs are generated as $x|c = \rho v_c + z$, $z \sim N(0, I)$. $w_{m,j}^\top v_c \lesssim \sqrt{\log d}/\sqrt{d}$ with high probability over the randomness of the random initialization of $w_{m,j}$ since $w_{m,j}^0 \sim \mathrm{Unif}(\mathbb{S}^{d-1})$ and $v_c \in \mathbb{S}^{d-1}$. Then we get

$$|\alpha_{m,j,k^*} - \alpha_{m,j,k^*,c}| \lesssim \rho\sqrt{\log d}/\sqrt{d}$$

for all $c \in [C]$ since

$$|\alpha_{m,j,k^*} - \alpha_{m,j,k^*,c}| = |\mathbb{E}_{z\sim\mathcal{N}(0,I)}[(\sigma_m(z) - \sigma_m(z + \rho w_{m,j}^\top v_{c'}))\mathrm{He}_{k^*}(z)]| \lesssim |\rho w_{m,j}^\top v_{c'}|.$$

Note that $\alpha_{m,j,k^*} = \mathbb{E}_{z\sim\mathcal{N}(0,I)}[\sigma_m(z)\mathrm{He}_{k^*}(z)]$ and use Lipschitz continuity for ReLU and use binomial expansion for polynomial activations. By the triangle inequality, we obtain

$$|\alpha_{m,j,k^*,c} - \alpha_{m,j,k^*,c'}| \le |\alpha_{m,j,k^*,c} - \alpha_{m,j,k^*}| + |\alpha_{m,j,k^*,c'} - \alpha_{m,j,k^*}| \lesssim \rho\sqrt{\log d}/\sqrt{d},$$

$$\alpha_{m,j,k^*,c} \gtrsim \alpha_{m,j,k^*} - O(\rho\sqrt{\log d}/\sqrt{d}) \quad \text{and} \quad \alpha_{m,j,k^*,c} \lesssim \alpha_{m,j,k^*} + O(\rho\sqrt{\log d}/\sqrt{d}).$$

Finally, use the assumption that $|\alpha_{m,j,k^*}| = \Theta(1)$. $\square$

We will show that the inequality in Lemma B.4 at the initialization continues to be satisfied:

**Lemma B.5.** *Let $\alpha_{m,j,k^*,c}^t$ be the $k^*$ th Hermite coefficient of $\sigma_m(\cdot + w_{m,j}^{t\top}v_c + b_{m,j})$. Note $\alpha_{m,j,k^*,c}^0 = \alpha_{m,j,k^*,c}$. Assume $k^* > 2$, $\sup_{c\in[C]}\kappa_{m,j,c}^s \lesssim d^{-1/2+1/(2k^*)}$ for all $s \le t$, and $t \lesssim \eta^{-1}Jd^{(k^*-2)/2}$. Then we have*

$$|\alpha_{m,j,k^*}^0 - \alpha_{m,j,k^*,c}^u| \lesssim \tilde{O}(d^{-1/2}) \quad \text{for all } c, j$$

*at arbitrary time $u \le t$.*

*Proof.* As discussed in Lemma C.8, we have $|w_j^{t\top}v_c| \lesssim \tilde{O}(d^{-1/2}) + \tilde{O}(\eta J^{-1}\int_0^t|\kappa_{m,j,c}^t|^{k^*}\mathrm{d}t) \lesssim \tilde{O}(d^{-1/2})$. Next, we repeat the same argument in Lemma B.4. $\square$

*Remark* B.6. We also assume $\left|\frac{\alpha_{m,j,k^*,c}^t}{\alpha_{m,j,k^*}}\right| = \tilde{\Theta}(1)$ and the sign does not change for all time $t$ (assumed in Assumption 3.4)

Lemma B.5 and Remark B.6 imply that we can temporarily ignore the dynamics of $\alpha_{m,j,k^*,c}^t$. So, we omit $t$ and denote the coefficient as $\alpha_{m,j,k^*,c}$ for now.

**Lemma B.7.** *For all $j$, there exists $c$ such that*

$$\beta_{c,k^*}\alpha_{m,j,k^*,c} > 0$$

*if $C \geq 2$ with high probability.*

*Proof.* Fix $j$. Use $\mathrm{sgn}\,\alpha_{m,j,k^*,c} = \mathrm{sgn}\,\alpha_{m,j,k^*c'}$ for all $c, c'$ with high probability and there exists at least one pair $(c, c')$ such that $\mathrm{sgn}\,\beta_{c,k^*} \neq \mathrm{sgn}\,\beta_{c'k^*}$ by assumption. These events imply that $\mathrm{sgn}\,\beta_{c,k^*}\alpha_{m,j,k^*,c} \neq \mathrm{sgn}\,\beta_{c'k^*}\alpha_{m,j,k^*c'}$. $\qquad\square$

Based on the above lemma, we define the set of $w_c^*$ which is "learnable":

**Definition B.8.** The set $\mathcal{C}_j \subset [C]$ consists of the class $c$ such that $\beta_{c,k^*}\alpha_{m,j,k^*,c} > 0$.

We roughly observe that

$$\frac{\mathrm{d}}{\mathrm{d}t}|\kappa_{m,j,c}^t| \simeq \eta J^{-1}\beta_{c,k^*}\alpha_{m,j,k^*,c}|\kappa_{m,j,c}^t|^{k^*-1} \simeq \begin{cases} \eta J^{-1}|\kappa_{m,j,c}^t|^{k^*-1} & \text{if } c \in \mathcal{C}_j, \\ -\eta J^{-1}|\kappa_{m,j,c}^t|^{k^*-1} & \text{if } c \notin \mathcal{C}_j, \end{cases}$$

which implies that $\mathcal{C}_j$ reflects the learnability of the tasks. We more formally have the following result:

### B.2. Evaluation of Spherical Gradient Flows

**Lemma B.9.** *Assume the conditions posed in Lemma B.5. We have*

$$\sum_{c \notin \mathcal{C}_j} \frac{\mathrm{d}}{\mathrm{d}t}|\kappa_{m,j,c}^t| \lesssim \frac{\eta}{CJ}\left(-|\kappa_{m,j,c}^t|^{k^*-1} + \underbrace{(\alpha_{m,j,k^*,1} - \alpha_{m,j,k^*,2})}_{\tilde{O}(d^{-1/2})}|\xi_{m,j,g}|^{k^*}\right).$$

*In addition, if*

- *$|\kappa_{m,j,c}^t| \gtrsim \tilde{\Theta}(d^{-1/2})$ for all $c \in \mathcal{C}_j$,*

- *$|\kappa_{m,j,c}^t| \lesssim \tilde{\Theta}(d^{-1/2+1/(2k^*)})$ for all $c \notin \mathcal{C}_j$,*

- *$|\xi_{m,j,g}^t| \lesssim \tilde{\Theta}(d^{-1/2+1/(2k^*)})$ for $c = 1, 2$*

*hold, then we have*

$$\sum_{c \in \mathcal{C}_j} \mathrm{sgn}(\kappa_{m,j,c}^t)\frac{\mathrm{d}}{\mathrm{d}t}\kappa_{m,j,c}^t \gtrsim \frac{\eta}{CJ}\left(\sum_{c=1}^C |\kappa_{m,j,c}^t|\right)^{k^*-1}.$$

*Proof.* By the standard argument of spherical gradient flow (please refer to Lemma C.7 for the parallel discussions in the discretized dynamics), we have

$$\begin{aligned}
\frac{\mathrm{d}}{\mathrm{d}t}\kappa_{m,j,c}^t \simeq & \frac{\eta k^*}{CJ}\Big(\beta_{c,k^*}\alpha_{m,j,k^*,c}(\kappa_{m,j,c})^{k^*-1}(1 - (\kappa_{m,j,c})^2) \\
& + (\sum_{c'} s_{c'}\alpha_{mjk^*c})(\xi_{m,j,g})^{k^*-1}((w_c^*)^\top(w_g^*) - \kappa_{m,j,c}\xi_{m,j,g}) \\
& + \sum_{c' \in \mathcal{C}_j \backslash \{c\}}^C \beta_{c'k^*}\alpha_{mjk^*c'}(\kappa_{m,j,c'})^{k^*-1}((w_{c'}^*)^\top(w_c^*) - \kappa_{mjc'1}\kappa_{m,j,c}) \\
& + \sum_{c' \notin \mathcal{C}_j \backslash \{c\}}^C \beta_{c'k^*}\alpha_{mjk^*c'}(\kappa_{m,j,c'})^{k^*-1}((w_{c'}^*)^\top(w_c^*) - \kappa_{m,j,c'}\kappa_{m,j,c})\Big) \\
= & \frac{\eta k^*}{CJ}\Big(\beta_{ck^*}\alpha_{m,j,k^*,c}(\kappa_{m,j,c})^{k^*-1}(1 - (\kappa_{m,j,c})^2) - (\alpha_{mjk^*1} - \alpha_{mjk^*2})(\xi_{m,j,g})^{k^*}\kappa_{m,j,c}
\end{aligned}$$

$$- \sum_{c' \in \mathcal{C}_j \setminus \{c\}} \underbrace{\beta_{c'k^*} \alpha_{mjk^*c'}}_{=+\tilde{\Theta}(1)} (\kappa_{m,j,c'})^{k^*} \kappa_{m,j,c} - \sum_{c' \notin \mathcal{C}_j \setminus \{c\}} \underbrace{\beta_{c'k^*} \alpha_{mjk^*c'}}_{=-\tilde{\Theta}(1)} (\kappa_{m,j,c'})^{k^*} \kappa_{m,j,c} \Bigg),$$

where we used Definition B.8 that is rewritten as

$$\mathrm{sgn}\beta_{c,k^*} \alpha_{m,j,k^*,c} = \begin{cases} 1 & \text{if} \quad c \in \mathcal{C}_j \\ -1 & \text{otherwise} \end{cases}$$

and $|\alpha_{mjk^*c}| = \tilde{\Theta}(1)$ in Assumption 3.4 at the last inequality. Now we have shown the first inequality in the statement.

Next, we will show the second inequality. Consider the sum for $c \notin \mathcal{C}_j$. We have the assumption that $|\kappa^t_{m,j,c}| \lesssim d^{-1/2+1/(2k^*)} \lesssim o_d(1)$ for all $c \notin \mathcal{C}_j$ and $k^*$ is even. Then,

$$\sum_{c \notin \mathcal{C}_j} \mathrm{sgn}(\kappa^t_{m,j,c}) \frac{\mathrm{d}}{\mathrm{dt}} \kappa^t_{m,j,c}$$

$$\lesssim \frac{\eta k^*}{CJ} \left( - \min_{c \notin \mathcal{C}_j} \{|\beta_{c,k^*} \alpha_{m,j,k^*,c}|\} (1 - o(1)^2) \sum_{c \notin \mathcal{C}_j} (\kappa_{m,j,c})^{k^*-1} + C \underbrace{|\alpha_{mjk^*1} - \alpha_{mjk^*2}|}_{\lesssim \tilde{O}(d^{-1/2}), \text{ from Lemma B.5}} (\xi_{m,j,g})^{k^*} \right.$$

$$\left. - \sum_{c \notin \mathcal{C}_j} \sum_{c' \in \mathcal{C}_j \setminus \{c\}} \underbrace{\beta_{c'k^*} \alpha_{mjk^*c'}}_{=+\Theta(1)} |\kappa_{m,j,c'}|^{k^*} |\kappa_{m,j,c}| - \sum_{c \notin \mathcal{C}_j} \sum_{c' \notin \mathcal{C}_j \setminus \{c\}} \underbrace{\beta_{c'k^*} \alpha_{mjk^*c'}}_{=-\Theta(1)} \underbrace{|\kappa_{m,j,c'}|^{k^*} |\kappa_{m,j,c}|}_{\leq \max_{c \notin \mathcal{C}_j} |\kappa^t_{m,j,c}|^{k^*+1}} \right).$$

$$\lesssim \frac{\eta}{CJ} \left( - \sum_{c \notin \mathcal{C}_j} |\kappa^t_{m,j,c}|^{k^*-1} + C^2 \max_{c \notin \mathcal{C}_j} |\kappa^t_{m,j,c}|^{k^*+1} + \tilde{O}(d^{-1/2}) |\xi_{m,j,g}|^{k^*} \right)$$

$$\lesssim \frac{\eta}{CJ} \left( - (1 - C^2 \cdot o_d(1)) \sum_{c \notin \mathcal{C}_j} |\kappa^t_{m,j,c}|^{k^*-1} + \tilde{O}(d^{-1/2}) |\xi_{m,j,g}|^{k^*} \right) \qquad (|\kappa^t_{m,j,c}| \lesssim o_d(1), \ \forall c)$$

$$\lesssim \frac{\eta}{CJ} \left( -C^{-k^*+2} \left( \sum_{c \notin \mathcal{C}_j} |\kappa^t_{m,j,c}| \right)^{k^*-1} + \tilde{O}(d^{-1/2}) |\xi_{m,j,g}|^{k^*} \right)$$

where we used $\left( \frac{1}{n} \sum_{i=1}^n |a_i| \right)^k \leq \frac{1}{n} \sum_{i=1}^n |a_i|^k$ for $k \in \mathbb{Z}_{\geq 1}$ and $a_1, \ldots, a_n \in \mathbb{R}$ by Jensen's inequality in the last inequality. As for the sum for $c \in \mathcal{C}_j$, we similarly have

$$\sum_{c \in \mathcal{C}_j} \mathrm{sgn}(\kappa^t_{m,j,c}) \frac{\mathrm{d}}{\mathrm{dt}} \kappa^t_{m,j,c}$$

$$\gtrsim \frac{\eta}{CJ} \left( C^{-k^*+2} \left( \sum_{c \in \mathcal{C}_j} |\kappa^t_{m,j,c}| \right)^{k^*-1} - C^2 \max_{c \in \mathcal{C}_j} |\kappa_{m,j,c}|^{k^*+1} - \tilde{O}(d^{-1/2}) (\xi_{m,j,g})^{k^*} \kappa_{m,j,c} \right)$$

$$\gtrsim \frac{\eta}{CJ} \left( \underbrace{\left( \sum_{c \in \mathcal{C}_j} |\kappa^t_{m,j,c}| \right)^{k^*-1}}_{\gtrsim d^{-\frac{k^*-1}{2}}} \left( C^{-k^*+2} - C^2 \underbrace{\max_{c \in \mathcal{C}_j} |\kappa^t_{m,j,c}|^2}_{o_d(1)} \right) - \underbrace{\tilde{O}(d^{-1/2}) (\xi_{m,j,g})^{k^*} |\kappa_{m,j,c}|}_{\tilde{O}(d^{-(k^*+2)/2}) \ll \tilde{\Theta}(d^{-(k^*-1)/2}) \leq \text{the first term}} \right).$$

$$\square$$

Next, we control the dynamics of $\xi_{m,j,g}^t$ corresponding to $w_g^*$. The growth rate is small because the signals cancel out:

**Lemma B.10.** *Assume the conditions posed in Lemma B.5. It holds that*

$$\frac{\mathrm{d}}{\mathrm{d}t}|\xi_{m,j,g}^t| \lesssim \frac{\eta}{CJ}\tilde{O}\left(\underbrace{(\alpha_{m,j,k^*,1} - \alpha_{m,j,k^*,2})}_{\lesssim d^{-1/2}}\xi_{m,j,g}^{k^*-1} + |\xi_{m,j,g}^t|(\sum_{c\notin\mathcal{C}_j}|\kappa_{m,j,c}|^{k^*} - \sum_{c\in\mathcal{C}_j}|\kappa_{m,j,c}|^{k^*})\right)$$

*for all $j$.*

*Proof.* We have the population GF as

$$\mathrm{sgn}(\xi_{m,j,g}^t)\frac{\mathrm{d}}{\mathrm{d}t}\xi_{m,j,g}^t$$

$$\simeq \mathrm{sgn}(\xi_{m,j,g}^t)\frac{\eta k^*}{CJ}\left((\alpha_{m,j,k^*,1} - \alpha_{m,j,k^*,2})(\xi_{m,j,g})^{k^*-1}(1 - (\xi_{mj1_1})^2)\right.$$

$$\left.+ \sum_{c=1}^{C}\beta_{c,k^*}\alpha_{mjk^*c}(\kappa_{m,j,c})^{k^*-1}(\underbrace{(w_c^*)^\top(w_g^*)}_{=0} - \kappa_{m,j,c}\xi_{m,j,g})\right)$$

$$= \mathrm{sgn}(\xi_{m,j,g}^t)\frac{\eta k^*}{CJ}\left(\underbrace{(\alpha_{m,j,k^*,1} - \alpha_{m,j,k^*,2})}_{\lesssim d^{-1/2},\text{ from Lemma B.5}}(\xi_{m,j,g})^{k^*-1}(1 - (\xi_{mj1_1})^2) - \sum_{c=1}^{C}\beta_{c,k^*}\alpha_{mjk^*c}(\kappa_{m,j,c})^{k^*}\xi_{m,j,g}\right)$$

$$\lesssim \frac{\eta}{CJ}\left(\tilde{O}(d^{-1/2})|\xi_{m,j,g}|^{k^*-1}(1 - (\xi_{mj1_1})^2) + \left(-\sum_{c\in\mathcal{C}_j}\underbrace{(\beta_{c,k^*}\alpha_{mjk^*c})}_{>0}(\kappa_{m,j,c})^{k^*} + \sum_{c\notin\mathcal{C}_j}\underbrace{(-\beta_{c,k^*}\alpha_{mjk^*c})}_{>0}(\kappa_{m,j,c})^{k^*}\right)|\xi_{m,j,g}|\right)$$

where we used

$$\mathrm{sgn}\beta_{c,k^*}\alpha_{m,j,k^*,c} = \begin{cases} 1 & \text{if } c \in \mathcal{C}_j \\ -1 & \text{otherwise} \end{cases}$$

at the last inequality. $\square$

Even if we ignore $|\kappa_{m,j,c}|$, $c \in \mathcal{C}_j$, which has the effect of reducing the gradient, the growth rate of $|\xi_{m,j,g}^t| + \sum_{c\notin\mathcal{C}_j}|\kappa_{m,j,c}^t|$ is small with the "information exponent" equals to $k^*$:

**Lemma B.11.** *Assume the conditions posed in Lemma B.5. Then we have*

$$\mathrm{sgn}(\xi_{m,j,g}^t)\frac{\mathrm{d}}{\mathrm{d}t}\xi_{m,j,g}^t + \sum_{c\notin\mathcal{C}_j}\mathrm{sgn}(\kappa_{m,j,c}^t)\frac{\mathrm{d}}{\mathrm{d}t}\kappa_{m,j,c}^t \lesssim \frac{\eta}{J}\tilde{O}\left(|\xi_{m,j,g}| + \sum_{c\notin\mathcal{C}_j}|\kappa_{m,j,c}|\right)^{k^*}.$$

*Therefore, if $t \leq \tilde{O}(J\eta^{-1}d^{(k^*-2)/2})$,*

$$|\xi_{m,j,g}^t| + \sum_{c\notin\mathcal{C}_j}|\kappa_{m,j,c}^t| \lesssim \tilde{O}(d^{-1/2}).$$

*Proof.* First, $|\kappa_{m,j,c}^0| \lesssim \tilde{\Theta}(d^{-1/2})$ for all $c \notin \mathcal{C}_j$ and $|\xi_{m,j,g}^t| + \sum_{c\notin\mathcal{C}_j}|\kappa_{m,j,c}^t| = \tilde{O}(d^{-1/2})$ at $t = 0$.

Next, we assume that there exists the time $\tau \lesssim \tilde{O}(J\eta^{-1}d^{(k^*-2)/2})$ such that $|\xi_{m,j,g}^\tau| + \sum_{c\notin\mathcal{C}_j}|\kappa_{m,j,c}^\tau| = \sup_{\tau'\in[t_1,\tau]}|\xi_{m,j,g}^{\tau'}| + \sum_{c\notin\mathcal{C}_j}|\kappa_{m,j,c}^{\tau'}| \simeq d^{-1/2+\delta}$ for some $1/(2k^*) > \delta > 0$[1] Then the assumptions in Lemma B.9

---

[1]We require $1/(2k^*) > \delta$ to satisfy $|w_j^\top v_c| \lesssim d^{-1/2}$.

are satisfied. Therefore, we have

$$\text{sgn}(\xi_{m,j,g}^t) \frac{d}{dt} \xi_{m,j,g}^t + \sum_{c \notin \mathcal{C}_j} \text{sgn}(\kappa_{m,j,c}^t) \frac{d}{dt} \kappa_{m,j,c}^t \lesssim \frac{\eta}{J} \tilde{O} \left( |\xi_{m,j,g}| + \sum_{c \notin \mathcal{C}_j} |\kappa_{m,j,c}| \right)^{k^*}$$

for $t \in [0, \tau]$.

Then we will show the contradiction. Let $|\xi_{m,j,g}^t| + \sum_{c \notin \mathcal{C}_j} |\kappa_{m,j,c}^t| = x_t$. the dynamics of $x^t$, $t \in [0, \tau]$ is evaluated as

$$\frac{d}{dt} x_t \leq \tilde{A} \frac{\eta}{J} (x_t)^{k^*},$$

where $\tilde{A} \lesssim \text{poly} \log d$ is a constant. By the Gronwall inequality,

$$x_\tau \leq \frac{x_0}{\left( 1 - (k^* - 1)^{-1} (x_0)^{k^*-1} \tilde{A} J^{-1} \eta \tau \right)^{1/(k^*-1)}}.$$

Therefore, we obtain

$$|\xi_{m,j,g}^\tau| + \sum_{c \notin \mathcal{C}_j} |\kappa_{m,j,c}^\tau|$$

$$\lesssim \frac{\tilde{O}(d^{-1/2})}{\left( 1 - \tilde{O} \left( \left( |\xi_{m,j,g}^0| + \sum_{c \notin \mathcal{C}_j} |\kappa_{m,j,c}^0| \right)^{k^*-1} J^{-1} \eta \tau \right) \right)^{1/(k^*-1)}}$$

$$\lesssim \frac{\tilde{O}(d^{-1/2})}{\left( 1 - \tilde{O} \left( d^{-(k^*-1)/2} d^{(k^*-2)/2} \right) \right)^{1/(k^*-1)}}$$

$$\lesssim \tilde{O}(d^{-1/2}),$$

which contradicts that $|\xi_{m,j,g}^\tau| + \sum_{c \notin \mathcal{C}_j} |\kappa_{m,j,c}^\tau| \gtrsim d^{-1/2+\delta}$. This implies that $|\kappa_{m,j,c}^t| \lesssim d^{-1/2}$ for all $c \notin \mathcal{C}_j$ holds if $t \lesssim J\eta^{-1} d^{(k^*-2)/2}$ and solving the ODE again leads to the desired result. $\square$

### B.3. Balancing the Race: Learning Before the Hermite Coefficients Deviate

We will show that the alignment $w_{m,j}^\top w_c^*$, $c \in \mathcal{C}_j$ becomes sufficiently large before the Hermite coefficients $\alpha_{m,j,k^*,c}^t$ deviate too much using a recursive argument.

The following lemma shows that $w_j$ tends to align with $w_c^*$, $c \in \mathcal{C}_j$:

**Lemma B.12.** *Assume the conditions posed in Lemma B.5. There exists $t_1 \lesssim \tilde{O}(J\eta^{-1} d^{(k^*-2)/2})$ such that*

1. *$\sum_{c \in \mathcal{C}_j} |\kappa_{m,j,c}^{t_1}| \simeq d^{-1/2+1/(2k^*)}$,*

2. *$|\kappa_{m,j,c}^{t_1}| \lesssim \tilde{O}(d^{-1/2})$ for all $c \notin \mathcal{C}_j$,*

3. *$|\xi_{m,j,g}^{t_1}| \lesssim \tilde{O}(d^{-1/2})$ for $c = 1, 2$.*

*Proof.* Combine the results in Lemma B.9 and Lemma B.11. Lemma B.9 implies the first condition by Gronwall inequality. Lemma B.11 leads to the second and the third conditions because $\max\{\max_{c \notin \mathcal{C}_j} |\kappa_{m,j,c}^{t_1}|, |\xi_{m,j,g}^{t_1}|\} \leq |\xi_{m,j,g}^{t_1}| + \sum_{c \notin \mathcal{C}_j} |\kappa_{m,j,c}^{t_1}|$. $\square$

The intuition of the final part in the proof of the above lemma is as follows: The differential equations of $x^t := \sum_{c \in \mathcal{C}_j} |\kappa_{m,j,c}^t|$ and $y^t := \sum_{c \notin \mathcal{C}_j} |\kappa_{m,j,c}^t| + |\xi_{m,j,g}|$ are

$$\frac{d}{dt} x^t \simeq \eta J^{-1} (x^t)^{k^*-1}, \quad x^0 \simeq d^{-1/2}$$

and

$$\frac{\mathrm{d}}{\mathrm{d}t} y^t \lesssim \eta J^{-1} (y^t)^{k^*}, \quad y^0 \simeq d^{-1/2}.$$

It takes at most $\eta^{-1} J(x^0)^{k^*-2} = \eta^{-1} J d^{(k^*-2)/2}$ time for $x^t$ to grow up to $d^{-\frac{1}{2}+\frac{1}{2k^*}}$ and on the other hand, it takes at least $\eta^{-1} J(y^0)^{k^*-1} = \eta^{-1} J d^{(k^*-1)/2}$ ($\gg \eta^{-1} J d^{(k^*-2)/2}$) time for $y^t$ to become larger than $\tilde{O}(d^{-1/2})$.

Repeating the same argument in Lemmas B.5 and B.9 to B.12, we have the following recurrence formula:

**Lemma B.13.** *Assume $k^* > 2$, $A_l \leq 1/2$ and there exists $t_l \lesssim \tilde{O}(J\eta^{-1} d^{A_l(k^*-2)})$ such that*

1. $\sum_{c \in \mathcal{C}_j} |\kappa_{m,j,c}^{\sum_{l'=1}^{l} t_{l'}}| \simeq d^{-A_l}$,

2. $\sum_{c \notin \mathcal{C}_j} |\kappa_{m,j,c}^{\sum_{l'=1}^{l} t_{l'}}| + |\xi_{m,j,g}| \lesssim d^{-1/2}$

3. $|\alpha_{m,j,k^*,c}^{\sum_{l'=1}^{l} t_{l'}} - \alpha_{m,j,k^*}| \lesssim \tilde{O}(d^{-1/2})$.

*Then, there exists $t_{l+1} \lesssim \eta^{-1} J d^{A_l(k^*-2)}$ such that*

1. $\sum_{c \in \mathcal{C}_j} |\kappa_{m,j,c}^{\sum_{l'=1}^{l+1} t_{l'}}| \simeq d^{-A_{l+1}}$ *where* $A_{l+1} = \frac{k^*-2}{k^*} A_l + \frac{1}{2k^*}$

2. $\sum_{c \notin \mathcal{C}_j} |\kappa_{m,j,c}^{\sum_{l'=1}^{l+1} t_{l'}}| + |\xi_{m,j,g}| \lesssim d^{-1/2}$

3. $|\alpha_{m,j,k^*,c}^{\sum_{l'=1}^{l+1} t_{l'}} - \alpha_{m,j,k^*}^0| \lesssim \tilde{O}(w_j^{t \top} v_c) \lesssim \tilde{O}(d^{-1/2})$.

*Proof.* Let $|\kappa_{m,j,c}^t| \lesssim d^{-\frac{k^*-2}{k^*} A_l - \frac{1}{2k^*}}$, $t \in [t_l, t_{l+1}]$. Following Lemmas B.5 and C.8, we have

$$\begin{aligned}
|w_j^{t \top} v_c| &\lesssim \tilde{O}(d^{-1/2}) + \tilde{O}(\eta J^{-1} \int_{t_l}^{t_{l+1}} |\kappa_{m,j,c}^t|^{k^*} \mathrm{d}t) \\
&\lesssim \tilde{O}(d^{-1/2}) + \tilde{O}\left(d^{-k^* \frac{k^*-2}{k^*} A_l - \frac{k^*}{2k^*} + A_l(k^*-2)}\right) \\
&\lesssim \tilde{O}(d^{-1/2}),
\end{aligned}$$

which implies the third inequality. Based on this, the first two inequalities follow from Gronwall's inequality. The differential equations of $x^t := \sum_{c \in \mathcal{C}_j} |\kappa_{m,j,c}^{\sum_{l'=1}^{l} t_{l'}+t}|$ and $y^t := \sum_{c \notin \mathcal{C}_j} |\kappa_{m,j,c}^{\sum_{l'=1}^{l} t_{l'}+t}| + |\xi_{m,j,g}|$ are

$$\frac{\mathrm{d}}{\mathrm{d}t} x^t \simeq \eta J^{-1} (x^t)^{k^*-1}, \quad x^0 \simeq d^{-A_l}$$

and

$$\frac{\mathrm{d}}{\mathrm{d}t} y^t \lesssim \eta J^{-1} (y^t)^{k^*}, \quad y^0 \simeq d^{-1/2}.$$

It takes at most $\eta^{-1} J(x^0)^{k^*-2} = \eta^{-1} J d^{A_l(k^*-2)}$ time for $x^t$ to grow up to $d^{-\frac{k^*-2}{k^*} A_l - \frac{1}{2k^*}}$ and on the other hand, it takes at least $\eta^{-1} J(y^0)^{k^*-1} = \eta^{-1} J d^{(k^*-1)/2}$ ($\gg \eta^{-1} J d^{A_l(k^*-2)}$) time for $y^t$ to become larger than $\tilde{O}(d^{-1/2})$. $\quad\square$

As shown in Lemma B.13, $A_l$ is shrinking as

$$A_{l+1} = \frac{k^* - 2}{k^*} A_l + \frac{1}{2k^*}, \quad A_0 = \frac{1}{2},$$

asymptotically approaching $1/4$. Repeating the above argument, we have the following result:

**Lemma B.14.** *Assume $k^* > 2$ and take arbitrary $\Delta > 0$. There exists $t_\Delta$ s.t.*

- $\sum_{c \in \mathcal{C}_j} |\kappa_{m,j,c}^{t_\Delta}| \simeq d^{-1/4-\Delta}$.

- $\sum_{c \notin \mathcal{C}_j} |\kappa_{m,j,c}^{t_\Delta}| + |\xi_{m,j,g}| \lesssim d^{-1/2}$

- $|\alpha_{m,j,k^*,c}^{t_\Delta} - \alpha_{m,j,k^*}^0| \lesssim \tilde{O}(w_j^{t \top} v_c) \lesssim \tilde{O}(d^{-1/2})$

**B.4. Blocking the alignment $w_{m,j}^t{}^\top w_g^*$**

We will show that the conditions in Lemma B.14 hold for $t > t_\Delta$. Note that $|\alpha_{m,j,k^*,c}^t - \alpha_{m,j,k^*}|$ is not assumed in $t > t_\Delta$:

**Lemma B.15.** *Take $\Delta = 1/(8k^*)$ defined in Lemma B.14 and take arbitrary $t > t_\Delta$. Additionally assume $k^* > 4$. Then we have*

$$\sum_{c \in \mathcal{C}_j} |\kappa_{m,j,c}^t| \gtrsim d^{-1/4-\Delta}$$

*and*

$$\frac{\mathrm{d}}{\mathrm{d}t}\left(|\xi_{m,j,g}^t| + \sum_{c \notin \mathcal{C}_j} |\kappa_{m,j,c}^t|\right) \lesssim 0.$$

*Proof.* Let $t_{\Delta'}$ is the first time that $\frac{\mathrm{d}}{\mathrm{d}t}\sum_{c \in \mathcal{C}_j}|\kappa_{m,j,c}^t| = 0$ (note that we can take $t_\Delta$ such that $\frac{\mathrm{d}}{\mathrm{d}t}\sum_{c \in \mathcal{C}_j}|\kappa_{m,j,c}^{t_\Delta}| \geq 0$ while satisfying the conditions in Lemma B.12 and this gradient flow stops at $t = t_{\Delta'} > t_\Delta$). First, because $\frac{\mathrm{d}}{\mathrm{d}t}\sum_{c \in \mathcal{C}_j}|\kappa_{m,j,c}^{t_\Delta}| \geq 0$ and the continuity of the derivative (except for $\sum_{c \in \mathcal{C}_j}|\kappa_{m,j,c}^{t_\Delta}| = 0$), we have $\frac{\mathrm{d}}{\mathrm{d}t}\sum_{c \in \mathcal{C}_j}|\kappa_{m,j,c}^t|\big|_{t=s} \geq 0$ for all $s \in [t_\Delta, t_{\Delta'}]$. This implies that $\sum_{c \in \mathcal{C}_j}|\kappa_{m,j,c}^t| \gtrsim d^{-1/4-\Delta}$ for all $t > t_\Delta$.

Next, we bound the derivative of $|\xi_{m,j,g}^s| + \sum_{c \notin \mathcal{C}_j}|\kappa_{m,j,c}^s|$. We assume that, there exists $t \in [t_\Delta, \infty)$ such that, $\sup_{t' \in [t_\Delta, t]}\left(|\xi_{m,j,g}^{t'}| + \sum_{c \notin \mathcal{C}_j}|\kappa_{m,j,c}^{t'}|\right) \simeq \tilde{O}(d^{-1/2+\delta})$ with $1/(6k^*) > \delta > 0$. However, for all $s \in [t_\Delta, t]$,

$$\mathrm{sgn}(\xi_{m,j,g}^s)\frac{d}{ds}\xi_{m,j,g}^s + \sum_{c \notin \mathcal{C}_j}\mathrm{sgn}(\kappa_{m,j,c}^s)\frac{d}{ds}\kappa_{m,j,c}^s$$

$$\lesssim \frac{\eta}{CJ}\left(\tilde{O}(\underbrace{(\alpha_{m,j,k^*,1}^t - \alpha_{m,j,k^*,2}^t)}_{\lesssim\tilde{O}(1),\ \text{NOT assuming Lemma B.5}}|\xi_{m,j,g}|^{k^*-1}(1 - (\xi_{mj1_1})^2)) + \sum_{c \notin \mathcal{C}_j}\underbrace{(-\beta_{c,k^*}\alpha_{mjk^*c}^t)}_{\leq\tilde{O}(1)}(\kappa_{m,j,c})^{k^*}|\xi_{m,j,g}|\right.$$

$$\left. - \sum_{c \in \mathcal{C}_j}\underbrace{\beta_{c,k^*}\alpha_{mjk^*c}^t}_{\geq\tilde{\Omega}(1)}\underbrace{(\kappa_{m,j,c})^{k^*}}_{\gtrsim d^{-1/4-\Delta}}|\xi_{m,j,g}|\right) + \frac{\eta}{CJ}\left(-\tilde{\Omega}\left(\sum_{c \notin \mathcal{C}_j}|\kappa_{m,j,c}^s|\right)^{k^*-1} + \underbrace{(\alpha_{m,j,k^*,1}^t - \alpha_{m,j,k^*,2}^t)}_{\lesssim\tilde{O}(1),\text{NOT assuming Lemma B.5}}\tilde{O}(|\xi_{m,j,g}|^{k^*})\right)$$

$$\lesssim \frac{\eta}{CJ}|\xi_{m,j,g}|\tilde{O}\left(d^{-(k^*-2)(1/2-\delta)} - \tilde{\Omega}(d^{-k^*(1/4+\Delta)})\right)$$

$$\lesssim 0$$

where we used $-(k^* - 2)(1/2 - \delta) = -(k^* - 2)(1/2 - 1/(6k^*)) < k^*(1/4 + 1/(8k^*)) = k^*(1/4 + \Delta)$ under the assumption that $k^* > 4$ (i.e. $k^* \geq 5$). This contradicts with the assumption that $\left(|\xi_{m,j,g}^t| + \sum_{c \notin \mathcal{C}_j}|\kappa_{m,j,c}^t|\right) \simeq (\text{poly}\log d)d^{-1/2+\delta}$. Therefore, we have $\left(|\xi_{m,j,g}^t| + \sum_{c \notin \mathcal{C}_j}|\kappa_{m,j,c}^t|\right) \lesssim (\text{poly}\log d)d^{-1/2}$ for all $t \in [t_\Delta, \infty)$ and this leads to $\mathrm{sgn}(\xi_{m,j,g}^t)\frac{\mathrm{d}}{\mathrm{d}t}\xi_{m,j,g}^t + \sum_{c \notin \mathcal{C}_j}\mathrm{sgn}(\kappa_{m,j,c}^t)\frac{\mathrm{d}}{\mathrm{d}t}\kappa_{m,j,c}^t \lesssim 0$ repeating the same calculation. □

By Lemma B.15, we have the following lemma:

**Lemma B.16.** *Assume $k^* > 4$. The following conditions hold true for all $t > t_\Delta$:*

1. $\sum_{c \in \mathcal{C}_j}|\kappa_{m,j,c}^t| \gtrsim d^{-1/4-1/(8k^*)}$,

2. $|\kappa_{m,j,c}^t| \lesssim \tilde{O}(d^{-1/2})$ *for all* $c \notin \mathcal{C}_j$,

3. $|\xi_{m,j,g}^t| \lesssim \tilde{O}(d^{-1/2})$ *for* $c = 1, 2$,

Finally we have the following theorem by Lemma B.16:

**Theorem B.17.** *For all $j = 1, \dots, J$, $c = 1, 2$, we have*

$$\sup_{t \geq 0} |\xi_{m,j,g}^t| \lesssim \tilde{O}(d^{-1/2})$$

*with high probability.*

## C. Proof of MoE Training

In this section, we present a formal proof that the MoE can learn teacher models (Theorem 4.6), under Assumption 3.2, Assumption 3.3, and Assumption 3.4. We execute the gradient-based optimization process outlined in Algorithm 1. Our proof builds upon and extends the reasoning presented by Oko et al. (2024a). We introduce polylogarithmic constants $A_i$ and $a_i$. $A_i$ is of the order $\text{polylog}(d)$, while $a_i$ is of the order $\frac{1}{\text{polylog}(d)}$, with the following order of strength:

$$0 < \begin{array}{c} A_1 \lesssim a_2^{-1} \lesssim A_2 \lesssim A_4 \lesssim a_4^{-1} \lesssim A_3 \\ a_2^{-1} \lesssim A_\rho \lesssim A_5, A_6 \lesssim a_5^{-1} \end{array} = \tilde{O}(1).$$

$A_1$ is derived from the high-probability bounds on the gradient of the experts. $a_2$ is derived from the threshold of the weak recovery of the neuron for the tasks. $A_2$ and $A_5$ are high-probability uniform bounds on the noise and small terms in the gradients of the experts and that of the gating network, respectively. $A_3$ is derived from the upper bounds of the specific components in the gradients of the experts. $A_4$ is derived from the upper bounds of the task correlation. $a_4$ and $a_5$ are derived from the upper bounds of the learning rates of the experts ($\eta_e$) and the router ($\eta_r$), respectively. $A_6$ originates from the lower bound of the Hermite coefficients in the router learning stage. $A_\rho$ reflects the order of the mean vector scaling in terms of $\rho$, as given by $\rho \simeq A_\rho$

An outline of the proof of MoE training is as follows:

1. **Initialization** (Appendix C.1)
   We first show that, after initialization, there exists a neuron $j_m^*$ within the set of professional experts $\mathcal{M}_c$ (Definition C.1) corresponding to the task $c$ that aligns with a constant factor stronger than other neurons (Lemma C.2 and Corollary C.3).

2. **Exploration Stage** (Appendix C.2)
   After the exploration stage, the neuron $j_m^*$, which was strongly aligned during initialization, undergoes weak recovery for the task $c_m^*$ it specializes in, whereas other neurons fail to achieve weak recovery and remains at a saddle point (Lemma C.6).

3. **Router Learning Stage** (Appendix C.3)
   After the router learning stage, the router directs the data $x_c$ from cluster corresponding to task $c$ to the experts in $\mathcal{M}_c$ (Lemma C.14).

4. **Expert Learning Stage** (Appendix C.4)
   After the router completes its learning and experts are reinitialized, the experts $m$ belonging to $\mathcal{M}_c$, which now receives the data $x_c$, achieve weak recovery (Lemma C.21) without being affected by inter-cluster interference and subsequently attain strong recovery (Lemma C.26).

5. **Second Layer Optimization Stage** (Appendix C.5)
   We finally show that by performing convex optimization on the second layer of the expert, the MoE achieves an $\epsilon$-error with respect to the teacher function (Lemma C.29).

To establish Theorem 4.6, we apply a union bound over multiple events. Given that $M = O(1)$ and $J \lesssim \sqrt{\log d}$, the set of events that hold with high probability remains closed under certain union bounds. By combining the aforementioned events, we conclude that the event in Theorem 4.6 holds with probability at least 0.99.

### C.1. Initialization

To start off, we consider the initial alignment between the neurons and the index features. The following lemma shows that a constant fraction of the neurons aligns with the task of their corresponding cluster by a constant factor more strongly than the remaining neurons. We provide a definition of the set of professional experts that depends on initialization.

**Definition C.1** (The set of the professional experts for class $c$)**.**

$$(j_m^*, c_m^*) := \operatorname{argmax}_{j,c} w_{m,j}^\top w_c^*,$$

$$\mathcal{M}_c := \{m \mid c = c_m^*\}.$$

The following Lemma holds with initialization.

**Lemma C.2.** *Assume $C$ is $O(1)$. Take arbitrary constants $\delta > 0$ and $A_d = 1 + O((\log d)^{-1})$. If*

$$M \simeq C \log(C/\delta)$$

*and*

$$\sqrt{\log d} \gtrsim J \gtrsim \frac{1}{C} \log \frac{M}{\delta},$$

*then $|\mathcal{M}_c| \geq 1$ for all $c$ and*

$$w_{m,j_m^*}^\top w_{c_m^*}^* \geq \frac{2}{\sqrt{d}}, \quad w_{m,j_m^*}^\top w_{c_m^*}^* \geq \max_{c \neq c_m^* \text{ or } j \neq j_m^*} A_d |w_{m,j}^\top w_c^*| + \frac{1}{\sqrt{d}\log d}$$

*with probability at least $1 - \delta$ with sufficiently large $d$.*

*Proof.* Fix $m$. By the symmetry, we have

$$\mathbb{P}[|\mathcal{M}_c| \geq 1] \geq 1 - (1 - C^{-1})^M.$$

By union bound,

$$\mathbb{P}[|\mathcal{M}_c| \geq 1 \; \forall c] \geq 1 - M(1 - C^{-1})^M \geq 1 - M \exp(-M/C) \geq 1 - \delta/3$$

where $M \simeq C \log(C/\delta)$.

$w_{m,j} \sim \mathrm{Unif}(\mathbb{S}^{d-1}(1))$ is obtained by $\frac{\tilde{w}_{m,j}}{\|\tilde{w}_{m,j}\|}$, where $\tilde{w}_{m,j} \sim N(0, \frac{1}{d}I)$. We have $\|\tilde{w}\| \sim 1$ with high probability (the same argument as Oko et al. (2024a)). Consider the value of $\tilde{\kappa}_{m,j,c} = \tilde{w}_{m,j}^\top (I - \sum_{c' \neq c} w_{c'}^* w_{c'}^{*\top}) w_c$ and $\sum_{c' \neq c} \tilde{w}_{m,j}^\top w_{c'}^* (w_{c'}^*)^\top w_c$. Then, for each $\tilde{\kappa}_{m,j,c}$, there exists $\bar{\kappa}_{m,j,c} \overset{\text{i.i.d.}}{\sim} \mathcal{N}(0, d^{-1})$ such that $\frac{\bar{\kappa}_{m,j,c}}{\tilde{\kappa}_{m,j,c}} = 1 + O(d^{-1/2})$ because $\tilde{\kappa}_{m,j,c}$ are independent and $\sum_{c' \neq c} \tilde{w}_{m,j}^\top w_{c'}^* (w_{c'}^*)^\top w_c \lesssim d^{-1}$. Therefore, we evaluate the values of $\bar{\kappa}_{m,j,c}$ instead of $w_{m,j}^\top w_c^*$.

We show that there is some $j$ s.t. $\bar{\kappa}_{m,j,c}$ is large enough: First,

$$\mathbb{P}[\bar{\kappa}_{m,j,c} < 2d^{-1/2}] < 0.9.$$

Then,

$$\mathbb{P}[\max_c \bar{\kappa}_{m,j,c} > 2d^{-1/2} \text{ for some } j] \geq 1 - (0.9)^{JC}.$$

Taking

$$J \gtrsim \frac{1}{C} \log \frac{M}{\delta},$$

$$\mathbb{P}[\max_c \bar{\kappa}_{m,j,c} > 2d^{-1/2} \text{ for some } j] \geq 1 - \frac{\delta}{3M}.$$

Following Chen et al. (2022), we have

$$\left(1 - \frac{\delta}{3MJ^2C^2}\right) w_{m,j_m^*}^\top w_{c_m^*}^* \geq \max_{c \neq c_m^* \text{ or } j \neq j_m^*} |w_{mj}^\top w_c^*| - O(d^{-1})$$

and therefore

$$w_{m,j_m^*}^\top w_{c_m^*}^* \geq \max_{c \neq c_m^* \text{ or } j \neq j_m^*} \left(1 + \frac{\delta}{6MJ^2C^2}\right) |w_{m,j}^\top w_c^*| + \frac{\delta}{6MJ^2C^2} w_{m,j_m^*}^\top w_{c_m^*}^* - O(d^{-1})$$

with probability at least $1 - \delta/(3M)$ and the desired result follows, using $\frac{\delta}{6MJ^2C^2} \gtrsim (\log d)^{-1}$ and $w_{m,j_m^*}^\top w_{c_m^*}^* \geq 2d^{-1/2}$.

$\square$

Since $w_{m,j}^\top w_c^* = O(\sqrt{\log d}/\sqrt{d})$ with high probability and $\frac{\max_c |\tilde{\alpha}_{m,j,k^*,c}^0 \beta_{c,k^*}|}{\min_{c'} |\tilde{\alpha}_{m,j,k^*,c'}^0 \beta_{c',k^*}|} = 1 + \tilde{O}(d^{-1/2})$ with high probability, by taking $\delta$ as sufficiently small, we have the following inequality:

**Corollary C.3** (Following (Chen et al., 2022; Oko et al., 2024a))**.** *When $J \gtrsim C^{-1} \log M$ and $M \gtrsim C \log C$, for all $m$, we have at least one neuron $w_{m,j}$ such that*

$$w_{m,j_m^*}^\top w_{c_m^*}^* \geq \frac{1}{\sqrt{d}}$$

*and*

$$
|\tilde{\alpha}_{m,j_m^*,k^*,c_m^*}^t \beta_{c_m^*,k^*}| (w_{m,j_m^*}^\top w_{c_m^*}^*)^{k^*-2}
$$
$$
\geq \max\{\max_{j,c} |\tilde{\alpha}_{m,j,k^*,c}^t \beta_{c,k^*}|, |\max_j \sum_{c' \in [C]} s_{c'} \tilde{\alpha}_{m,j,k^*,c'}^t \gamma_{k^*}|\} \max_{c \neq c_m^* \text{ or } j \neq j_m^*} |w_{m,j}^\top w_c^*|^{k^*-2} + a(w_{m,j_m^*}^\top w_{c_m^*}^*)^{k^*-2}.
$$

*with probability at least $0.999$, where $a$ is a small constant $\lesssim (\log d)^{k^*-2}$.*

*Remark* C.4. The term $\tilde{\alpha}_{m,j,k^*,c}^t$, defined in Lemma C.7, varies with time $t$ as it is a Hermite coefficient influenced by the mean vector $\rho v_c$ of the data $x_c$. Nevertheless, Corollary C.3 holds for all $t$ throughout the exploration stage. This is ensured by the bounds on the Hermite coefficients provided in Lemma C.8 and Lemma C.9.

*Remark* C.5. From this section, we will discuss Phase I to IV on the event that the initialization was successful.

## C.2. Exploration Stage

We train the first layer of the experts. We employ the correlation loss to eliminate the interactions between neurons. The alignment at time $t$, denoted as $\kappa_{c,m,j}^t$, is defined as the inner product of the feature index $w_c^*$ and the weight of $j$-th neuron $w_{m,j}^t$ at time $t$, expressed as $\kappa_{c,m,j}^t := w_c^{*\top} w_{m,j}^t$. Similarly, $\kappa_{g,m,j}^t := w_g^{*\top} w_{m,j}^t$ is defined in the same manner. The purpose of this subsection is to establish Lemma C.6, which serves as the formal statement of Lemma 4.9. Within the expert set $\mathcal{M}_c$, there exists a neuron $j$ such that the alignment magnitude $\kappa_{c,m,j}$ with the feature index $w_c^*$ satisfies $\kappa_{c,m,j} \geq a_2$ for some constant $a_2 > 0$. In contrast, for all other expert sets $\mathcal{M}_{c'}$ with $c' \neq c$, no neuron achieves such alignment; that is, $\kappa_{c',m,j} \leq \tilde{O}(d^{-\frac{1}{2}}) < a_2$ for all $j \in \mathcal{M}_{c'}$. Moreover, the remaining neurons in $\mathcal{M}_c$ also do not reach this level of alignment. To prove Lemma C.6, we first decompose the stochastic gradient update into its population and noise components. Then, by introducing auxiliary sequences, we establish a lower bound for $\kappa_{c_m^*,m,j_m^*}^t$ (i.e., weak recovery) and upper bounds for $|\kappa_{c,m,j}^t|$ for all $(c,j) \neq (c_m^*, j_m^*)$ and $|\kappa_{g,m,j}^t|$ for all $j \in [J]$.

**Lemma C.6** (Formal)**.** *Consider the expert $m \in \mathcal{M}_c$. Let $w_c^{*\top} w_{c'}^* \leq A_4 d^{-\frac{1}{2}} = \tilde{O}(d^{-1/2})$ for all $c \neq c'$, $w_c^{*\top} w_g^* \leq A_4 d^{-\frac{1}{2}} = \tilde{O}(d^{-1/2})$ for all $c$ and $\eta^t = \eta_e \leq a_4 d^{-\frac{k^*}{2}}$. Then, with high probability, there exists some time $t_1 \leq T_1 = \tilde{\Theta}(\eta_e^{-1} d^{\frac{k^*-2}{2}})$ such that the following conditions hold:*

- $\kappa_{c_m^*,m,j_m^*}^{t_1} \geq a_2$,

- $|\kappa_{c,m,j}^{t_1}| \leq 5 A_3 d^{-\frac{1}{2}} = \tilde{O}(d^{-1/2})$, *for all $(c,j) \neq (c_m^*, j_m^*)$,*

- $|\kappa_{g,m,j}^{t_1}| \leq 5 A_3 d^{-\frac{1}{2}} = \tilde{O}(d^{-1/2})$, *for all $j \in [J]$.*

**Gradient update decomposition.** First, we assess the evolution of the alignment by analyzing its population and stochastic contributions. Note that at the exploration stage, the router distributes the data to the experts with an equal probability of $\frac{1}{M}$ because the weights $\theta_m$ of the gating network are initialized to zero.

In Lemma C.7, we will evaluate the update of the spherical gradient descent

$$
\kappa_{c,m,j}^{t+1} = w_c^{*\top} w_{c,m,j}^{t+1} = w_c^{*\top} \frac{[w_{c,m,j}^t - \eta^t (I_d - w_{m,j}^t w_{m,j}^{t\top}) \nabla_{w_{m,j}^t} \mathbf{1}(m(x_c) = m) \pi_m(x_c) y_c a_{m,j} \sigma_m(w_{m,j}^{t\top} x_c + b_{m,j})]}{\|w_{m,j}^t - \eta^t (I_d - w_{m,j}^t w_{m,j}^{t\top}) \nabla_{w_{m,j}^t} \mathbf{1}(m(x_c) = m) \pi_m(x_c) y_c a_{m,j} \sigma_m(w_{m,j}^{t\top} x_c + b_{m,j})\|}.
$$

Note that $I_d - w_{m,j}^t w_{m,j}^{t\top}$ is a projection matrix used to project the gradient $\nabla_{w_{m,j}^t} \mathbf{1}(m(x_c) = m) \pi_m(x_c) y_c a_{m,j} \sigma_m(w_{m,j}^{t\top} x_c + b_{m,j})$ onto the tangent space of the sphere $\mathbb{S}^{d-1}$. Additionally, normalization is performed to return the vector $w_{c,m,j}^{t+1}$ to the unit sphere.

**Lemma C.7.** *Suppose that $\eta^t = \eta_e \leq a_4 d^{-\frac{k^*}{2}}$ and $\kappa^0_{c^*_m, m, j^*_m} \geq \frac{1}{2} d^{-\frac{1}{2}}$. With high probability, the update of $\kappa^t_{c,m,j}$ and $\kappa^t_{g,m,j}$ satisfies the following bounds.*

$$\kappa^{t+1}_{c^*_m, m, j^*_m} \geq \kappa^t_{c^*_m, m, j^*_m} + \frac{\eta^t}{CM^2} \sum_{c' \in [C]} \sum_{i=k^*}^{p^*} \left[ i\tilde{\alpha}_{m,j^*_m,i,c'} \beta_{c',i} (\kappa^t_{c',m,j^*_m})^{i-1} (w^*_{c^*_m}{}^\top w^*_{c'} - \kappa^t_{c^*_m, m, j^*_m} \kappa^t_{c',m,j}) \right.$$

$$\left. + is_{c'} \tilde{\alpha}_{m,j^*_m,i,c'} \gamma_i (\kappa^t_{g,m,j^*_m})^{i-1} (w^*_{c^*_m}{}^\top w^*_g - \kappa^t_{c^*_m, m, j^*_m} \kappa^t_{g,m,j^*_m}) \right] - \kappa^t_{c^*_m, m, j^*_m} (\eta^t)^2 A_1{}^2 d$$

$$+ \eta^t w^*_c{}^\top (I_d - w^t_{m,j^*_m} w^t_{m,j^*_m}{}^\top) \Xi^t_{w_{m,j^*_m}}.$$

*Let $c \neq c^*_m$ or $j \neq j^*_m$, we have*

$$\kappa^t_{c,m,j} + \frac{\eta^t}{CM^2} \sum_{c' \in [C]} \sum_{i=k^*}^{p^*} \left[ i\tilde{\alpha}_{m,j,i,c'} \beta_{c',i} (\kappa^t_{c',m,j})^{i-1} (w^*_c{}^\top w^*_{c'} - \kappa^t_{c,m,j} \kappa^t_{c',m,j}) + is_{c'} \tilde{\alpha}_{m,j,i,c'} \gamma_i (\kappa^t_{g,m,j})^{i-1} \right.$$

$$\left. \cdot (w^*_c{}^\top w^*_g - \kappa^t_{c,m,j} \kappa^t_{g,m,j}) \right] - \frac{|\kappa^t_{c,m,j}| (\eta^t)^2 A_1{}^2 d}{2} - \frac{(\eta^t)^3 A_1{}^3 d^{\frac{3}{2}}}{2} + \eta^t w^*_c{}^\top (I_d - w^t_{m,j} w^t_{m,j}{}^\top) \Xi^t_{w_{m,j}}$$

$$\leq \kappa^{t+1}_{c,m,j} \leq$$

$$\kappa^t_{c,m,j} + \frac{\eta^t}{CM^2} \sum_{c' \in [C]} \sum_{i=k^*}^{p^*} \left[ i\tilde{\alpha}_{m,j,i,c'} \beta_{c',i} (\kappa^t_{c',m,j})^{i-1} (w^*_c{}^\top w^*_{c'} - \kappa^t_{c,m,j} \kappa^t_{c',m,j}) + is_{c'} \tilde{\alpha}_{m,j,i,c'} \gamma_i (\kappa^t_{g,m,j})^{i-1} \right.$$

$$\left. \cdot (w^*_c{}^\top w^*_g - \kappa^t_{c,m,j} \kappa^t_{g,m,j}) \right] + \frac{|\kappa^t_{c,m,j}| (\eta^t)^2 A_1{}^2 d}{2} + \frac{(\eta^t)^3 A_1{}^3 d^{\frac{3}{2}}}{2} + \eta^t w^*_c{}^\top (I_d - w^t_{m,j} w^t_{m,j}{}^\top) \Xi^t_{w_{m,j}}.$$

*Let $j \in [J]$, we have*

$$\kappa^t_{g,m,j} + \frac{\eta^t}{CM^2} \sum_{c' \in [C]} \sum_{i=k^*}^{p^*} \left[ i\tilde{\alpha}_{m,j,i,c'} \beta_{c',i} (\kappa^t_{c',m,j})^{i-1} (w^*_g{}^\top w^*_{c'} - \kappa^t_{g,m,j} \kappa^t_{c',m,j}) + is_{c'} \tilde{\alpha}_{m,j,i,c'} \gamma_i (\kappa^t_{g,m,j})^{i-1} \right.$$

$$\left. \cdot (1 - (\kappa^t_{g,m,j})^2) \right] - \frac{|\kappa^t_{g,m,j}| (\eta^t)^2 A_1{}^2 d}{2} - \frac{(\eta^t)^3 A_1{}^3 d^{\frac{3}{2}}}{2} + \eta^t w^*_g{}^\top (I_d - w^t_{m,j} w^t_{m,j}{}^\top) \Xi^t_{w_{m,j}}$$

$$\leq \kappa^{t+1}_{g,m,j} \leq$$

$$\kappa^t_{g,m,j} + \frac{\eta^t}{CM^2} \sum_{c' \in [C]} \sum_{i=k^*}^{p^*} \left[ i\tilde{\alpha}_{m,j,i,c'} \beta_{c',i} (\kappa^t_{c',m,j})^{i-1} (w^*_g{}^\top w^*_{c'} - \kappa^t_{g,m,j} \kappa^t_{c',m,j}) + is_{c'} \tilde{\alpha}_{m,j,i,c'} \gamma_i (\kappa^t_{g,m,j})^{i-1} \right.$$

$$\left. \cdot (1 - (\kappa^t_{g,m,j})^2) \right] + \frac{|\kappa^t_{g,m,j}| \eta^t{}^2 A_1{}^2 d}{2} + \frac{(\eta^t)^3 A_1{}^3 d^{\frac{3}{2}}}{2} + \eta^t w^*_g{}^\top (I_d - w^t_{m,j} w^t_{m,j}{}^\top) \Xi^t_{w_{m,j}}.$$

$\Xi^t_{w_{m,j}}$ *represents a mean-zero random variable satisfying $\|\Xi^t_{w_{m,j}}\| = \tilde{O}(d^{\frac{1}{2}})$ and $|u^\top \Xi^t_{w_{m,j}}| = \tilde{O}(1)$, where $u \sim$ Unif $(\mathbb{S}^{d-1})$, with high probability. We can also obtain $|\kappa^{t+1}_{c,m,j} - \kappa^t_{c,m,j}| = \tilde{O}(\eta_e)$ and $|\kappa^{t+1}_{g,m,j} - \kappa^t_{g,m,j}| = \tilde{O}(\eta_e)$ with high probability.*

*Proof.* The population gradient for the first layer of the expert can be represented as a decomposition in the following manner.

$$\nabla_{w_{m,j}} \mathbb{E}_c \mathbb{E}_{x_c} \left[ \mathbf{1}\left(m(x_c) = m\right) \pi_m(x_c) y_c a_{m,j} \sigma_m (w_{m,j}{}^\top x_c + b_{m,j}) \right]$$

$$= \mathbb{E}_c \mathbb{E}_{x_c} \left[ \mathbf{1}\left(m(x_c) = m\right) \pi_m(x_c) y_c a_{m,j} \sigma_m'(w_{m,j}{}^\top x_c + b_{m,j}) x_c \right]$$

$$= \frac{1}{CM^2} \sum_{c \in [C]} \mathbb{E}_{x_c} \left[ \left( \sum_{i=k^*}^{p^*} \frac{\beta_{c,i}}{\sqrt{i!}} \mathrm{He}_i(w^*_c{}^\top x_c) + s_c \sum_{i=k^*}^{p^*} \frac{\gamma_i}{\sqrt{i!}} \mathrm{He}_i(w^*_g{}^\top x_c) \right) \left( \sum_{i=1}^{\infty} \frac{i\alpha_{m,j,i}}{\sqrt{i!}} \mathrm{He}_{i-1}(w_{m,j}{}^\top x_c) \right) x_c \right]$$

$$
= \frac{1}{CM^2} \sum_{c \in [C]} \Bigg( \mathbb{E}_z \Bigg[ \Bigg( \sum_{i=k^*}^{p^*} \frac{\beta_{c,i}}{\sqrt{i!}} \mathrm{He}_i(w_c^{*\top}z) + s_c \sum_{i=k^*}^{p^*} \frac{\gamma_i}{\sqrt{i!}} \mathrm{He}_i(w_g^{*\top}z) \Bigg) \Bigg( \sum_{i=1}^{\infty} \frac{i\alpha_{m,j,i}}{\sqrt{i!}} \sum_{l=0}^{i-1} \binom{i-1}{l} \mathrm{He}_{i-l-1}(w_{m,j}^{\top}z)(\rho w_{m,j}^{\top}v_c)^l \Bigg) z \Bigg]
$$

$$
+ \mathbb{E}_z \Bigg[ \Bigg( \sum_{i=k^*}^{p^*} \frac{\beta_{c,i}}{\sqrt{i!}} \mathrm{He}_i(w_c^{*\top}z) + s_c \sum_{i=k^*}^{p^*} \frac{\gamma_i}{\sqrt{i!}} \mathrm{He}_i(w_g^{*\top}z) \Bigg) \Bigg( \sum_{i=1}^{\infty} \frac{i\alpha_{m,j,i}}{\sqrt{i!}} \sum_{l=0}^{i-1} \binom{i-1}{l} \mathrm{He}_{i-l-1}(w_{m,j}^{\top}z)(\rho w_{m,j}^{\top}v_c)^l \Bigg) \rho v_c \Bigg] \Bigg)
$$

$$
= \frac{1}{CM^2} \sum_{c \in [C]} \Bigg( \underbrace{ \mathbb{E}_z \Bigg[ \Bigg( \sum_{i=k^*}^{p^*} \frac{i\beta_{c,i}}{\sqrt{i!}} \mathrm{He}_{i-1}(w_c^{*\top}z) \Bigg) \Bigg( \sum_{i=1}^{\infty} \frac{i\alpha_{m,j,i}}{\sqrt{i!}} \sum_{l=0}^{i-1} \binom{i-1}{l} \mathrm{He}_{i-l-1}(w_{m,j}^{\top}z)(\rho w_{m,j}^{\top}v_c)^l \Bigg) w_c^* \Bigg] }_{(\mathrm{I})}
$$

$$
+ \underbrace{ \mathbb{E}_z \Bigg[ \Bigg( s_c \sum_{i=k^*}^{p^*} \frac{i\gamma_i}{\sqrt{i!}} \mathrm{He}_{i-1}(w_g^{*\top}z) \Bigg) \Bigg( \sum_{i=1}^{\infty} \frac{i\alpha_{m,j,i}}{\sqrt{i!}} \sum_{l=0}^{i-1} \binom{i-1}{l} \mathrm{He}_{i-l-1}(w_{m,j}^{\top}z)(\rho w_{m,j}^{\top}v_c)^l \Bigg) w_g^* \Bigg] }_{(\mathrm{II})}
$$

$$
+ \underbrace{ \mathbb{E}_z \Bigg[ \Bigg( \sum_{i=k^*}^{p^*} \frac{\beta_{c,i}}{\sqrt{i!}} \mathrm{He}_i(w_c^{*\top}z) + s_c \sum_{i=k^*}^{p^*} \frac{\gamma_i}{\sqrt{i!}} \mathrm{He}_i(w_g^{*\top}z) \Bigg) \Bigg( \sum_{i=2}^{\infty} \frac{i\alpha_{m,j,i}}{\sqrt{i!}} \sum_{l=0}^{i-2} \binom{i-1}{l}(i-l-1)\mathrm{He}_{i-l-2}(w_{m,j}^{\top}z)(\rho w_{m,j}^{\top}v_c)^l \Bigg) w_{m,j} \Bigg] }_{(\mathrm{III})}
$$

$$
+ \underbrace{ \mathbb{E}_z \Bigg[ \Bigg( \sum_{i=k^*}^{p^*} \frac{\beta_{c,i}}{\sqrt{i!}} \mathrm{He}_i(w_c^{*\top}z) + s_c \sum_{i=k^*}^{p^*} \frac{\gamma_i}{\sqrt{i!}} \mathrm{He}_i(w_g^{*\top}z) \Bigg) \Bigg( \sum_{i=1}^{\infty} \frac{i\alpha_{m,j,i}}{\sqrt{i!}} \sum_{l=0}^{i-1} \binom{i-1}{l} \mathrm{He}_{i-l-1}(w_{m,j}^{\top}z)(\rho w_{m,j}^{\top}v_c)^l \Bigg) \rho v_c \Bigg] }_{(\mathrm{IV})} \Bigg).
$$

where the second equality is due to $\mathbb{P}(m(x_c) = m) = \frac{1}{M}$ and $\pi_{m(x_c)} = \frac{1}{M}$. The third equality follows from the definition $x_c := z + \rho v_c$ where $z \sim \mathcal{N}(0, I_d)$, along with the condition $v_c^{\top}w_{c'}^* = 0$ and $v_c^{\top}w_g^* = 0$ for all $(c, c')$, as well as the binomial expansion. The fourth equality is due to Stein's Lemma and integration by parts.

With the spherical gradient, (III) is negligible since $w_{m,j}$ is unit-norm and $w_c^{*\top}(I_d - w_{m,j}w_{m,j}^{\top})w_{m,j} = 0$. When considering $v_c^{\top}\nabla_{w_{m,j}}\mathbb{E}[\mathbf{1}\left(m(x_c) = m\right)\pi_m(x_c)y_c a_{m,j}\sigma_m(w_{m,j}^{\top}x_c + b_{m,j})]$, (IV) can be ignored because $v_c^{\top}w_{c'}^* = 0$ for all $(c, c')$. Thus, we expand (I) and (II).

$$
(\mathrm{I}) = \sum_{i=k^*}^{p^*} \sum_{l=i}^{\infty} \Big( \frac{l\alpha_{m,j,l}}{\sqrt{l!}} \binom{l-1}{l-i}(\rho w_{m,j}^{\top}v_c)^{l-i} \Big) \frac{i\beta_{c,i}}{\sqrt{i!}}(i-1)!(w_c^{*\top}w_{m,j})^{i-1}w_c^*.
$$

$$
(\mathrm{II}) = \sum_{i=k^*}^{p^*} \sum_{l=i}^{\infty} \Big( \frac{l\alpha_{m,j,l}}{\sqrt{l!}} \binom{l-1}{l-i}(\rho w_{m,j}^{\top}v_c)^{l-i} \Big) \frac{is_c\gamma_{c,i}}{\sqrt{i!}}(i-1)!(w_g^{*\top}w_{m,j})^{i-1}w_g^*.
$$

Here, we introduce the discrepancy $\Xi^t{}_{w_{m,j}}$ between the population and the empirical gradient.

$$
\Xi^t{}_{w_{m,j}} = -\nabla_{w_{m,j}^t}\mathbf{1}\left(m(x_c) = m\right)\pi_m(x_c)y_c a_{m,j}\sigma_m(w_{m,j}^{t\top}x_c + b_{m,j})
$$
$$
+ \nabla_{w_{m,j}^t}\mathbb{E}[\mathbf{1}\left(m(x_c) = m\right)\pi_m(x_c)y_c a_{m,j}\sigma_m(w_{m,j}^{t\top}x_c + b_{m,j})]
$$

We evaluate the empirical update of the alignment.

$$
\kappa_{c,m,j}^{t+1} = \frac{\kappa_{c,m,j}^t - \eta^t w_c^{*\top}(I_d - w_{m,j}^t w_{m,j}^{t\top})\nabla_{w_{m,j}^t}\mathbf{1}\left(m(x_c) = m\right)\pi_m(x_c)y_c a_{m,j}\sigma_m(w_{m,j}^{t\top}x_c + b_{m,j})}{\left\| w_{m,j}^t - \eta^t(I_d - w_{m,j}^t w_{m,j}^{t\top})\nabla_{w_{m,j}^t}\mathbf{1}\left(m(x_c) = m\right)\pi_m(x_c)y_c a_{m,j}\sigma_m(w_{m,j}^{t\top}x_c + b_{m,j})\right\|}
$$
$$
\overset{(i)}{\geq} \kappa_{c,m,j}^t - \eta^t w_c^{*\top}(I_d - w_{m,j}^t w_{m,j}^{t\top})\nabla_{w_{m,j}^t}\mathbf{1}\left(m(x_c) = m\right)\pi_m(x_c)y_c a_{m,j}\sigma_m(w_{m,j}^{t\top}x_c + b_{m,j})
$$
$$
- \frac{|\kappa_{c,m,j}^t|(\eta^t)^2}{2}\left\| \nabla_{w_{m,j}^t}\mathbf{1}\left(m(x_c) = m\right)\pi_m(x_c)y_c a_{m,j}\sigma_m(w_{m,j}^{t\top}x_c + b_{m,j})\right\|^2
$$

$$- \frac{(\eta^t)^3}{2} \| \nabla_{w_{m,j}^t} \mathbf{1}\left(m(x_c) = m\right) \pi_m(x_c) y_c a_{m,j} \sigma_m({w_{m,j}^t}^\top x_c + b_{m,j}) \|^3$$

$$\overset{(ii)}{\geq} \kappa_{c,m,j}^t + \eta^t {w_c^*}^\top (I_d - w_{m,j}^t {w_{m,j}^t}^\top) \frac{1}{CM^2} \sum_{c' \in [C]} \sum_{i=k^*}^{p^*} \left( \sum_{l=i}^\infty \left( \frac{l\alpha_{m,j,l}}{\sqrt{l!}} \binom{l-1}{l-i} (\rho {w_{m,j}^t}^\top v_{c'})^{l-i} \right) \right.$$

$$\left. \cdot \frac{i\beta_{c',i}}{\sqrt{i!}} (i-1)! ({w_{c'}^*}^\top w_{m,j}^t)^{i-1} \right) w_{c'}^* + \eta^t {w_c^*}^\top (I_d - w_{m,j}^t {w_{m,j}^t}^\top)$$

$$\cdot \frac{1}{CM^2} \sum_{c' \in [C]} \sum_{i=k^*}^{p^*} \left( \sum_{l=i}^\infty \left( \frac{l\alpha_{m,j,l}}{\sqrt{l!}} \binom{l-1}{l-i} (\rho {w_{m,j}^t}^\top v_{c'})^{l-i} \right) \frac{is_{c'}\gamma_i}{\sqrt{i!}} (i-1)! ({w_g^*}^\top w_{m,j}^t)^{i-1} \right) w_g^*$$

$$- \frac{|\kappa_{c,m,j}^t|(\eta^t)^2 {A_1}^2 d}{2} - \frac{(\eta^t)^3 {A_1}^3 d^{\frac{3}{2}}}{2} + \eta^t {w_c^*}^\top (I_d - w_{m,j}^t {w_{m,j}^t}^\top) \Xi_{w_{m,j}}^t$$

$$= \kappa_{c,m,j}^t + \eta^t {w_c^*}^\top (I_d - w_{m,j}^t {w_{m,j}^t}^\top) \frac{1}{CM^2} \sum_{c' \in [C]} \sum_{i=k^*}^{p^*} \left( \frac{i\alpha_{m,j,i}}{\sqrt{i!}} + \sum_{l=i+1}^\infty \left( \frac{l\alpha_{m,j,l}}{\sqrt{l!}} \binom{l-1}{l-i} (\rho {w_{m,j}^t}^\top v_c')^{l-i} \right) \right.$$

$$\left. \cdot \frac{i\beta_{c',i}}{\sqrt{i!}} (i-1)! ({w_{c'}^*}^\top w_{m,j}^t)^{i-1} \right) w_{c'}^* + \eta^t {w_c^*}^\top (I_d - w_{m,j}^t {w_{m,j}^t}^\top)$$

$$\cdot \frac{1}{CM^2} \sum_{c' \in [C]} \sum_{i=k^*}^{p^*} \left( \frac{i\alpha_{m,j,i}}{\sqrt{i!}} + \sum_{l=i+1}^\infty \left( \frac{l\alpha_{m,j,l}}{\sqrt{l!}} \binom{l-1}{l-i} (\rho {w_{m,j}^t}^\top v_{c'})^{l-i} \right) \frac{is_{c'}\gamma_i}{\sqrt{i!}} (i-1)! ({w_g^*}^\top w_{m,j}^t)^{i-1} \right) w_g^*$$

$$- \frac{|\kappa_{c,m,j}^t|(\eta^t)^2 {A_1}^2 d}{2} - \frac{(\eta^t)^3 {A_1}^3 d^{\frac{3}{2}}}{2} + \eta^t {w_c^*}^\top (I_d - w_{m,j}^t {w_{m,j}^t}^\top) \Xi_{w_{m,j}}^t$$

$$= \kappa_{c,m,j}^t + \frac{\eta^t}{CM^2} \sum_{c' \in [C]} \sum_{i=k^*}^{p^*} \left( i\tilde{\alpha}_{m,j,i,c'}^t \beta_{c',i} (\kappa_{c',m,j}^t)^{i-1} ({w_c^*}^\top w_{c'}^* - \kappa_{c,m,j}^t \kappa_{c',m,j}^t) + is_{c'} \tilde{\alpha}_{m,j,i,c'}^t \gamma_i (\kappa_{g,m,j}^t)^{i-1} \right.$$

$$\left. \cdot ({w_c^*}^\top w_g^* - \kappa_{c,m,j}^t \kappa_{g,m,j}^t) \right) - \frac{|\kappa_{c,m,j}^t|(\eta^t)^2 {A_1}^2 d}{2} - \frac{(\eta^t)^3 {A_1}^3 d^{\frac{3}{2}}}{2} + \eta^t {w_c^*}^\top (I_d - w_{m,j}^t {w_{m,j}^t}^\top) \Xi_{w_{m,j}}^t .$$

where (i) is due to Taylor expansion, Cauchy-Schwarz inequality, and the orthogonality property of the Hermite polynomials. In (ii), we used the expansion of (I) and (II) and $\| \nabla_{w_{m,j}^t} \mathbf{1}\left(m(x_c) = m\right) \pi_m(x_c) y_c a_{m,j} \sigma_m({w_{m,j}^t}^\top x_c + b_{m,j}) \| \leq A_1 d^{\frac{1}{2}}$ which holds with high probability. Here, we introduced $\tilde{\alpha}_{m,j,i,c}^t$ defined by

$$\frac{i\tilde{\alpha}_{m,j,i,c}^t}{\sqrt{i!}} := \frac{i\alpha_{m,j,i}}{\sqrt{i!}} + \sum_{l=i+1}^\infty \left( \frac{l\alpha_{m,j,l}}{\sqrt{l!}} \binom{l-1}{l-i} (\rho {w_{m,j}^t}^\top v_c)^{l-i} \right).$$

Furthermore, it can be equivalently rewritten as follows:

$$\tilde{\alpha}_{m,j,i,c}^t = \frac{1}{\sqrt{i!}} \mathbb{E}_z \left[ a_{m,j} \sigma_m'({w_{m,j}^t}^\top z + \rho {w_{m,j}^t}^\top v_c + b_{m,j}) \mathrm{He}_i({w_{m,j}^t}^\top z) \right].$$

Note that the definition of $\tilde{\alpha}_{m,j,i,c}$ given here differs from that of $\alpha_{m,j,i,c}$ in Appendix B. In Appendix C, we define the Hermite coefficients by incorporating the perturbation induced by $\rho {w_{m,j}}^\top v_c$, and formulate them in a manner involving a first-order derivative. To distinguish this modified definition, we introduced the tilde notation.

In the same way, we obtain an upper bound as follows:

$$\kappa_{c,m,j}^{t+1} \leq \kappa_{c,m,j}^t + \frac{\eta^t}{CM^2} \sum_{c' \in [C]} \sum_{i=k^*}^{p^*} \left( i\tilde{\alpha}_{m,j,i,c'}^t \beta_{c',i} (\kappa_{c',m,j}^t)^{i-1} ({w_c^*}^\top w_{c'}^* - \kappa_{c,m,j}^t \kappa_{c',m,j}^t) + is_{c'} \tilde{\alpha}_{m,j,i,c'}^t \gamma_i (\kappa_{g,m,j}^t)^{i-1} \right.$$

$$\left. \cdot ({w_c^*}^\top w_g^* - \kappa_{c,m,j}^t \kappa_{g,m,j}^t) \right) + \frac{|\kappa_{c,m,j}^t|(\eta^t)^2 {A_1}^2 d}{2} + \frac{(\eta^t)^3 {A_1}^3 d^{\frac{3}{2}}}{2} + (I_d - w_{m,j}^t {w_{m,j}^t}^\top) {w_c^*}^\top \Xi_{w_{m,j}}^t .$$

When $(c, j) = (c_m^*, j_m^*)$, $\frac{\kappa_{c,m,j}^t (\eta^t)^2 A_1{}^2 d}{2} + \frac{(\eta^t)^3 A_1{}^3 d^{\frac{3}{2}}}{2} \leq \kappa_{c,m,j}^t (\eta^t)^2 A_1{}^2 d$ since we have $\kappa_{c_m^*,m,j_m^*}^t \geq d^{-\frac{1}{2}}$ and $\eta^t \leq a_4 d^{-\frac{k^*}{2}}$.

We obtain an upper bound on the difference of $\kappa_{c,m,j}$ over a single step.

$$
\begin{aligned}
|\kappa_{c,m,j}^{t+1} - \kappa_{c,m,j}^t| \leq{}& \eta^t |w_c^{*\top}(I_d - w_{m,j}^t w_{m,j}^{t\top}) \nabla_{w_{m,j}^t} \mathbf{1}\,(m(x_c) = m)\, \pi_m(x_c) y_c a_{m,j} \sigma_m(w_{m,j}^{t\top} x_c + b_{m,j})| \\
&+ \frac{|\kappa_{c,m,j}^t|(\eta^t)^2}{2} \|\nabla_{w_{m,j}^t} \mathbf{1}\,(m(x_c) = m)\, \pi_m(x_c) y_c a_{m,j} \sigma_m(w_{m,j}^{t\top} x_c + b_{m,j})\|^2 \\
&+ \frac{(\eta^t)^3}{2} |w_c^{*\top}(I_d - w_{m,j}^t w_{m,j}^{t\top}) \nabla_{w_{m,j}^t} \mathbf{1}\,(m(x_c) = m)\, \pi_m(x_c) y_c a_{m,j} \sigma_m(w_{m,j}^{t\top} x_c + b_{m,j})| \\
&\cdot \|\nabla_{w_{m,j}^t} \mathbf{1}\,(m(x_c) = m)\, \pi_m(x_c) y_c a_{m,j} \sigma_m(w_{m,j}^{t\top} x_c + b_{m,j})\|^2 \\
={}& \tilde{O}(\eta_e).
\end{aligned}
$$

All terms on the RHS are bounded by $\tilde{O}(\eta_e)$ since
$|w_c^{*\top}(I_d - w_{m,j}^t w_{m,j}^{t\top}) \nabla_{w_{m,j}^t} \mathbf{1}\,(m(x_c) = m)\, \pi_m(x_c) y_c a_{m,j} \sigma_m(w_{m,j}^{t\top} x_c + b_{m,j})| = \tilde{O}(1)$,
$\|\nabla_{w_{m,j}^t} \mathbf{1}\,(m(x_c) = m)\, \pi_m(x_c) y_c a_{m,j} \sigma_m(w_{m,j}^{t\top} x_c + b_{m,j})\|^2 = \tilde{O}(d)$, and
$|w_c^{*\top}(I_d - w_{m,j}^t w_{m,j}^{t\top}) \nabla_{w_{m,j}^t} \mathbf{1}\,(m(x_c) = m)\, \pi_m(x_c) y_c a_{m,j} \sigma_m(w_{m,j}^{t\top} x_c + b_{m,j})|$
$\cdot \|\nabla_{w_{m,j}^t} \mathbf{1}\,(m(x_c) = m)\, \pi_m(x_c) y_c a_{m,j} \sigma_m(w_{m,j}^{t\top} x_c + b_{m,j})\|^2 = \tilde{O}(d)$ with high probability.

We establish similar statements for $\kappa_{g,m,j}$. We obtain a lower bound as follows:

$$
\begin{aligned}
\kappa_{g,m,j}^{t+1} ={}& \frac{\kappa_{g,m,j}^t - \eta^t w_g^{*\top}(I_d - w_{m,j}^t w_{m,j}^{t\top}) \nabla_{w_{m,j}^t} \mathbf{1}\,(m(x_c) = m)\, \pi_m(x_c) y_c a_{m,j} \sigma_m(w_{m,j}^{t\top} x_c + b_{m,j})}{\|w_{m,j}^t - \eta^t(I_d - w_{m,j}^t w_{m,j}^{t\top}) \nabla_{w_{m,j}^t} \mathbf{1}\,(m(x_c) = m)\, \pi_m(x_c) y_c a_{m,j} \sigma_m(w_{m,j}^{t\top} x_c + b_{m,j})\|} \\
\overset{(i)}{\geq}{}& \kappa_{g,m,j}^t - \eta^t w_g^{*\top}(I_d - w_{m,j}^t w_{m,j}^{t\top}) \nabla_{w_{m,j}^t} \mathbf{1}\,(m(x_c) = m)\, \pi_m(x_c) y_c a_{m,j} \sigma_m(w_{m,j}^{t\top} x_c + b_{m,j}) \\
&- \frac{|\kappa_{g,m,j}^t|(\eta^t)^2}{2} \|\nabla_{w_{m,j}^t} \mathbf{1}\,(m(x_c) = m)\, \pi_m(x_c) y_c a_{m,j} \sigma_m(w_{m,j}^{t\top} x_c + b_{m,j})\|^2 \\
&- \frac{(\eta^t)^3}{2} \|\nabla_{w_{m,j}^t} \mathbf{1}\,(m(x_c) = m)\, \pi_m(x_c) y_c a_{m,j} \sigma_m(w_{m,j}^{t\top} x_c + b_{m,j})\|^3 \\
\overset{(ii)}{\geq}{}& \kappa_{g,m,j}^t + \eta^t w_g^{*\top}(I_d - w_{m,j}^t w_{m,j}^{t\top}) \frac{1}{CM^2} \sum_{c' \in [C]} \sum_{i=k^*}^{p^*} \Big( \sum_{l=i}^{\infty} \Big( \frac{l\alpha_{m,j,l}}{\sqrt{l!}} \binom{l-1}{l-i} (\rho w_{m,j}^{t\top} v_{c'})^{l-i} \Big) \\
&\cdot \frac{i\beta_{c',i}}{\sqrt{i!}} (i-1)! (w_{c'}^{*\top} w_{m,j}^t)^{i-1} \Big) w_{c'}^* + \eta^t w_g^{*\top}(I_d - w_{m,j}^t w_{m,j}^{t\top}) \\
&\cdot \frac{1}{CM^2} \sum_{c' \in [C]} \sum_{i=k^*}^{p^*} \Big( \sum_{l=i}^{\infty} \Big( \frac{l\alpha_{m,j,l}}{\sqrt{l!}} \binom{l-1}{l-i} (\rho w_{m,j}^{t\top} v_{c'})^{l-i} \Big) \frac{is_{c'}\gamma_i}{\sqrt{i!}} (i-1)! (w_g^{*\top} w_{m,j}^t)^{i-1} \Big) w_g^* \\
&- \frac{|\kappa_{g,m,j}^t|(\eta^t)^2 A_1{}^2 d}{2} - \frac{(\eta^t)^3 A_1{}^3 d^{\frac{3}{2}}}{2} + \eta^t w_g^{*\top}(I_d - w_{m,j}^t w_{m,j}^{t\top}) \Xi^t{}_{w_{m,j}} \\
={}& \kappa_{g,m,j}^t + \eta^t w_g^{*\top}(I_d - w_{m,j}^t w_{m,j}^{t\top}) \frac{1}{CM^2} \sum_{c' \in [C]} \sum_{i=k^*}^{p^*} \Big( \frac{i\alpha_{m,j,i}}{\sqrt{i!}} + \sum_{l=i+1}^{\infty} \Big( \frac{l\alpha_{m,j,l}}{\sqrt{l!}} \binom{l-1}{l-i} (\rho w_{m,j}^{t\top} v_{c'}')^{l-i} \Big) \\
&\cdot \frac{i\beta_{c',i}}{\sqrt{i!}} (i-1)! (w_{c'}^{*\top} w_{m,j}^t)^{i-1} \Big) w_{c'}^* + \eta^t w_g^{*\top}(I_d - w_{m,j}^t w_{m,j}^{t\top}) \\
&\cdot \frac{1}{CM^2} \sum_{c' \in [C]} \sum_{i=k^*}^{p^*} \Big( \frac{i\alpha_{m,j,i}}{\sqrt{i!}} + \sum_{l=i+1}^{\infty} \Big( \frac{l\alpha_{m,j,l}}{\sqrt{l!}} \binom{l-1}{l-i} (\rho w_{m,j}^{t\top} v_{c'})^{l-i} \Big) \frac{is_{c'}\gamma_i}{\sqrt{i!}} (i-1)! (w_g^{*\top} w_{m,j}^t)^{i-1} \Big) w_g^*
\end{aligned}
$$

$$- \frac{|\kappa_{g,m,j}^t|(\eta^t)^2 A_1^2 d}{2} - \frac{(\eta^t)^3 A_1^3 d^{\frac{3}{2}}}{2} + \eta^t w_g^{*\top}(I_d - w_{m,j}^t w_{m,j}^{t\top})\Xi^t_{w_{m,j}}$$

$$= \kappa_{g,m,j}^t + \frac{\eta^t}{CM^2} \sum_{c' \in [C]} \sum_{i=k^*}^{p^*} \left( i\tilde{\alpha}_{m,j,i,c'}^t \beta_{c',i}(\kappa_{c',m,j}^t)^{i-1}(w_g^{*\top} w_{c'}^* - \kappa_{g,m,j}^t \kappa_{c',m,j}^t) + is_{c'}\tilde{\alpha}_{m,j,i,c'}^t \gamma_i(\kappa_{g,m,j}^t)^{i-1} \right.$$

$$\left. \cdot (1 - (\kappa_{g,m,j}^t)^2) \right) - \frac{|\kappa_{g,m,j}^t|(\eta^t)^2 A_1^2 d}{2} - \frac{(\eta^t)^3 A_1^3 d^{\frac{3}{2}}}{2} + \eta^t w_g^{*\top}(I_d - w_{m,j}^t w_{m,j}^{t\top})\Xi^t_{w_{m,j}}.$$

where (i) is due to Taylor expansion, Cauchy-Schwarz inequality, and the orthogonality property of the Hermite polynomials. In (ii), we used the expansion of (I) and (II) and $\|\nabla_{w_{m,j}^t} \mathbf{1}(m(x_c) = m)\pi_m(x_c)y_c a_{m,j}\sigma_m(w_{m,j}^{t\top}x_c + b_{m,j})\| \leq A_1 d^{\frac{1}{2}}$ which holds with high probability.

In the same way, we obtain an upper bound as follows:

$$\kappa_{g,m,j}^{t+1} \leq \kappa_{g,m,j}^t + \frac{\eta^t}{CM^2} \sum_{c' \in [C]} \sum_{i=k^*}^{p^*} \left( i\tilde{\alpha}_{m,j,i,c'}^t \beta_{c',i}(\kappa_{c',m,j}^t)^{i-1}(w_g^{*\top} w_{c'}^* - \kappa_{g,m,j}^t \kappa_{c',m,j}^t) + is_{c'}\tilde{\alpha}_{m,j,i,c'}^t \gamma_i(\kappa_{g,m,j}^t)^{i-1} \right.$$

$$\left. \cdot (1 - (\kappa_{g,m,j}^t)^2) \right) - \frac{|\kappa_{g,m,j}^t|(\eta^t)^2 A_1^2 d}{2} - \frac{(\eta^t)^3 A_1^3 d^{\frac{3}{2}}}{2} + \eta^t w_g^{*\top}(I_d - w_{m,j}^t w_{m,j}^{t\top})\Xi^t_{w_{m,j}}.$$

Similar to $\kappa_{c,m,j}$, we obtain an upper bound on the difference of $\kappa_{g,m,j}$ over a single step.

$$|\kappa_{g,m,j}^{t+1} - \kappa_{g,m,j}^t| \leq \eta^t |w_g^{*\top}(I_d - w_{m,j}^t w_{m,j}^{t\top})\nabla_{w_{m,j}^t} \mathbf{1}(m(x_c) = m)\pi_m(x_c)y_c a_{m,j}\sigma_m(w_{m,j}^{t\top}x_c + b_{m,j})|$$

$$+ \frac{|\kappa_{g,m,j}^t|(\eta^t)^2}{2}\|\nabla_{w_{m,j}^t} \mathbf{1}(m(x_c) = m)\pi_m(x_c)y_c a_{m,j}\sigma_m(w_{m,j}^{t\top}x_c + b_{m,j})\|^2$$

$$+ \frac{(\eta^t)^3}{2}|w_g^{*\top}(I_d - w_{m,j}^t w_{m,j}^{t\top})\nabla_{w_{m,j}^t} \mathbf{1}(m(x_c) = m)\pi_m(x_c)y_c a_{m,j}\sigma_m(w_{m,j}^{t\top}x_c + b_{m,j})|$$

$$\cdot \|\nabla_{w_{m,j}^t} \mathbf{1}(m(x_c) = m)\pi_m(x_c)y_c a_{m,j}\sigma_m(w_{m,j}^{t\top}x_c + b_{m,j})\|^2$$

$$= \tilde{O}(\eta_e).$$

$\square$

Note that $\Xi^t_{w_{m,j}}$ are mean-zero sub-Weibull random variables, and their partial sums exhibit strong concentration behavior.

**Weak recovery for the corresponding cluster.** Building on Lemma C.7, we establish Lemma C.6.

Now we show that the mean vector $\rho v_c$ does not significantly alter the Hermite coefficients of the activation function when $\kappa_{c,m,j}^s = \tilde{O}(d^{-\frac{1}{2}})$.

**Lemma C.8.** *Suppose that* $|v_c^{\top} w_{m,j}^s| = \tilde{O}(d^{-\frac{1}{2}})$, $|\kappa_{c,m,j}^s| = \tilde{O}(d^{-\frac{1}{2}})$ *and* $|\kappa_{g,m,j}^s| = \tilde{O}(d^{-\frac{1}{2}})$ *for all* $s = 0, 1, \ldots, t \leq \tilde{O}(d^{k^*-1})$. *Then, by setting* $\eta^t = \eta_e \leq a_4 d^{-\frac{k^*}{2}}$, *we obtain that* $|\tilde{\alpha}_{m,j,i,c}^{t+1} - \alpha_{m,j,i}| = \tilde{O}(d^{-\frac{1}{2}})$ *with high probability.*

*Proof.* Consider the case where $\alpha_{m,j,i} > 0$. Suppose that $|v_c^{\top} w_{m,j}^s| = \tilde{O}(d^{-\frac{1}{2}})$, $|\kappa_{c,m,j}^s| = \tilde{O}(d^{-\frac{1}{2}})$ and $|\kappa_{g,m,j}^s| = \tilde{O}(d^{-\frac{1}{2}})$ for all $s = 0, 1, \ldots, t$. Then, by leveraging the evaluation of the gradient update in Lemma C.7, we obtain that

$$|v_c^{\top} w_{m,j}^{t+1}| \leq |v_c^{\top} w_{m,j}^t| + \left|\frac{\eta^t \rho}{CM^2} \sum_{i=k^*}^{p^*} \sqrt{i+1}\tilde{\alpha}_{m,j,i,c}^t \beta_{c,i}(\kappa_{c,m,j}^t)^i (1 - (v_c^{\top} w_{m,j}^t)^2)\right|$$

$$+ \left|\frac{\eta^t \rho}{CM^2} \sum_{i=k^*}^{p^*} \sqrt{i+1}s_c \tilde{\alpha}_{m,j,i,c}^t \gamma_i(\kappa_{g,m,j}^t)^i (1 - (v_c^{\top} w_{m,j}^t)^2)\right|$$

$$+ \frac{|v_c^{\top} w_{m,j}^t|(\eta^t)^2 A_1^2 d}{2} + \frac{(\eta^t)^3 A_1^3 d^{\frac{3}{2}}}{2} + \eta^t v_c^{\top}(I_d - w_{m,j}^t w_{m,j}^{t\top})\Xi^t_{w_{m,j}}$$

$$\leq |v_c^\top w_{m,j}^0| + \sum_{s=0}^{t} \Big[ \frac{\eta^s \rho}{CM^2} p^* \sqrt{p^*+1} (\max_i |\tilde{\alpha}_{m,j,i,c}^s \beta_{c,i}| (\kappa_{c,m,j}^s)^{k^*} + \max_i |s_c \tilde{\alpha}_{m,j,i,c}^s \gamma_i| (\kappa_{g,m,j}^s)^{k^*})$$

$$+ \frac{|v_c^\top w_{m,j}^s|(\eta^s)^2 A_1^2 d}{2} + \frac{(\eta^s)^3 A_1^3 d^{\frac{3}{2}}}{2} + \eta^s v_c^\top (I_d - w_{m,j}^s w_{m,j}^{s\top}) \Xi^s{}_{w_{m,j}^s} \Big]$$

$$\leq |v_c^\top w_{m,j}^0| + t \frac{\eta_e \rho}{CM^2} p^* \sqrt{p^*+1} (\max_{i,s} |\tilde{\alpha}_{m,j,i,c}^s \beta_{c,i}| \max_s |\kappa_{c,m,j}^s|^{k^*} + \max_{i,s} |s_c \tilde{\alpha}_{m,j,i,c}^s \gamma_i| \max_s |\kappa_{g,m,j}^s|^{k^*})$$

$$+ t \frac{\max_s |v_c^\top w_{m,j}^s|(\eta_e)^2 A_1^2 d}{2} + \frac{(\eta_e)^3 A_1^3 d^{\frac{3}{2}}}{2} + \sum_{s=0}^{t} \eta_e v_c^\top (I_d - w_{m,j}^s w_{m,j}^{s\top}) \Xi^s{}_{w_{m,j}^s}$$

$$= \tilde{O}(d^{-\frac{1}{2}})$$

where the last inequality is due to $\frac{\eta_e \rho}{CM^2} p^* \sqrt{p^*+1} (\max_{i,s} |\tilde{\alpha}_{m,j,i,c}^s \beta_{c,i}| \max_s |\kappa_{c,m,j}^s|^{k^*} +$ $\max_{i,s} |s_c \tilde{\alpha}_{m,j,i,c}^s \gamma_i| \max_s |\kappa_{g,m,j}^s|^{k^*}) = \tilde{O}(d^{-k^*})$, $\frac{\max_s |v_c^\top w_{m,j}^s|(\eta_e)^2 A_1^2 d}{2} = \tilde{O}(d^{-k^* + \frac{1}{2}})$, and $\frac{(\eta_e)^3 A_1^3 d^{\frac{3}{2}}}{2} = \tilde{O}(d^{-\frac{3k^*}{2} + \frac{3}{2}})$, since we have $\eta_e \leq a_4 d^{-\frac{k^*}{2}}$, $|v_c^\top w_{m,j}^s| = \tilde{O}(d^{-\frac{1}{2}})$, and $\kappa_{c,m,j}^s = \tilde{O}(d^{-\frac{1}{2}})$. Note that $|v_c^\top w_{m,j}^0| = \tilde{O}(d^{-\frac{1}{2}})$ with high probability.

Additionally, when $t \leq \tilde{O}(d^{k^*-1})$,

$$|\sum_{s=0}^{t} \eta_e v_c^\top (I_d - w_{m,j}^s w_{m,j}^{s\top}) \Xi^s{}_{w_{m,j}}| = \tilde{O}(\eta_e \sqrt{t}) = \tilde{O}(d^{-\frac{1}{2}})$$

with high probability.

Recall that

$$\frac{i\alpha_{m,j,i}}{\sqrt{i!}} + \sum_{l=i+1}^{\infty} \Big( \frac{l\alpha_{m,j,l}}{\sqrt{l!}} \binom{l-1}{l-i} (\rho w_{m,j}^t {}^\top v_{c'})^{l-i} \Big) = \frac{i\tilde{\alpha}_{m,j,i,c'}^t}{\sqrt{i!}}.$$

Since $\rho v_c^\top w_{m,j}^s = \tilde{O}(d^{-\frac{1}{2}})$, the series $\sum_{l=i+1}^{\infty} \Big( \frac{l\alpha_{m,j,l}}{\sqrt{l!}} \binom{l-1}{l-i} (\rho w_{m,j}^t {}^\top v_{c'})^{l-i} \Big)$ decays exponentially with respect to $d$. Together with $v_c^\top w_{m,j}^0 = \tilde{O}(d^{-\frac{1}{2}})$ and $\kappa_{c,m,j}^0 = \tilde{O}(d^{-\frac{1}{2}})$ with high probability over initialization randomness, we obtain $|\tilde{\alpha}_{m,j,i,c}^{t+1} - \alpha_{m,j,i}| = \tilde{O}(d^{-\frac{1}{2}})$ with high probability by induction. The proof can be similarly established for the case where $\alpha_{m,j,i} < 0$ since $k^*$ is even. $\qquad\square$

Furthermore, we demonstrate that when $\kappa_{c,m,j}^t$ grows asymptotically larger than those of other neurons whose $\kappa_{c,m,j}^t$ have not increased significantly, $v_c^\top w_{m,j}^t$ also becomes larger, leading to an increase in the Hermite coefficient $\tilde{\alpha}_{m,j,i,c}^t$.

**Lemma C.9.** *Consider a neuron which satisfies $\alpha_{m,j,k^*} \beta_{c,k^*} > 0$ and $\alpha_{m,j,i} \beta_{c,i} > 0$ for $k^* < i \leq p^*$. Suppose that $|v_c^\top w_{m,j}^s| = \tilde{\Omega}(d^{-\frac{1}{2}})$, $\kappa_{c,m,j}^s = \tilde{\Omega}(d^{-\frac{1}{2}})$, and $|\kappa_{g,m,j}^s| = \tilde{O}(d^{-\frac{1}{2}})$ for all $s = 0, 1, \ldots, t \leq \tau = \tilde{O}(d^{k^*-1})$. Then, by setting $\eta^t = \eta_e \leq a_4 d^{-\frac{k^*}{2}}$, we obtain $|\tilde{\alpha}_{m,j,i,c}^{t+1}| - |\alpha_{m,j,i}| = \tilde{\Omega}(d^{-\frac{1}{2}})$ with high probability.*

*Proof.* Consider the case where $\alpha_{m,j,i} > 0$. Suppose that $\kappa_{c,m,j}^s = \tilde{\Omega}(d^{-\frac{1}{2}}) \geq 0$ and $|\kappa_{g,m,j}^s| = \tilde{O}(d^{-\frac{1}{2}})$ for all $s = 0, 1, \ldots, t$, we have

$$v_c^\top w_{m,j}^{t+1} \geq v_c^\top w_{m,j}^0 + \frac{\eta_e \rho}{CM^2} \sum_{s=0}^{t} \Big[ \sum_{i=k^*}^{p^*} \sqrt{i+1} \tilde{\alpha}_{m,j,i,c}^s \beta_{c,i} (\kappa_{c,m,j}^s)^i (1 - (v_c^\top w_{m,j}^s)^2)$$

$$+ \sum_{i=k^*}^{p^*} \sqrt{i+1} s_c \tilde{\alpha}_{m,j,i,c}^s \gamma_i (\kappa_{g,m,j}^s)^i (1 - (v_c^\top w_{m,j}^s)^2) \Big]$$

$$- t \frac{\max_s |v_c^\top w_{m,j}^s|(\eta_e)^2 A_1^2 d}{2} - t \frac{(\eta_e)^3 A_1^3 d^{\frac{3}{2}}}{2} + \sum_{s=0}^{t} \eta_e v_c^\top (I_d - w_{m,j}^s w_{m,j}^{s\top}) \Xi^s{}_{w_{m,j}}$$

$$\geq v_c^\top w_{m,j}^0 + \frac{\eta_e \rho}{CM^2} \sum_{s=0}^{t} \sqrt{k^*+1} \tilde{\alpha}_{m,j,k^*,c}^s \beta_{c,k^*} (\kappa_{c,m,j}^s)^i (1 - (v_c^\top w_{m,j}^s)^2)$$

$$- \frac{\eta_e \rho}{CM^2} \sum_{s=0}^{t} p^* \sqrt{p^*+1} \max_c |s_c \tilde{\alpha}_{m,j,i,c}^s \gamma_i||\kappa_{g,m,j}^s|^{k^*} (1 - (v_c^\top w_{m,j}^s)^2)$$

$$- t \frac{\max_s |v_c^\top w_{m,j}^s|(\eta_e)^2 A_1{}^2 d}{2} - t \frac{(\eta_e)^3 A_1{}^3 d^{\frac{3}{2}}}{2} - \Big| \sum_{s=0}^{t} \eta_e v_c^\top (I_d - w_{m,j}^s w_{m,j}^s{}^\top) \Xi^s{}_{w_{m,j}} \Big|$$

$$= \tilde{\Omega}(d^{-\frac{1}{2}}),$$

where we used $\frac{\eta_e \rho}{CM^2} p^* \sqrt{p^*+1} \max_c |s_c \tilde{\alpha}_{m,j,i,c}^s \gamma_i||\kappa_{g,m,j}^s|^{k^*} = \tilde{O}(d^{-k^*})$, $\frac{\max_s |v_c^\top w_{m,j}^s|(\eta_e)^2 A_1{}^2 d}{2} = \tilde{O}(d^{-k^*+\frac{1}{2}})$, and $\frac{(\eta_e)^3 A_1{}^3 d^{\frac{3}{2}}}{2} = \tilde{O}(d^{-\frac{3k^*}{2}+\frac{3}{2}})$, since we have $\eta_e \leq a_4 d^{-\frac{k^*}{2}}$, $|v_c^\top w_{m,j}^s| = \tilde{O}(d^{-\frac{1}{2}})$, and $\kappa_{c,m,j}^s = \tilde{O}(d^{-\frac{1}{2}})$. In addition, when $t \leq \tilde{O}(d^{k^*-1})$, we have that $|\sum_{s=0}^{t} \eta_e v_c^\top (I_d - w_{m,j}^s w_{m,j}^s{}^\top) \Xi^s{}_{w_{m,j}}| = \tilde{O}(\eta_e \sqrt{t}) = \tilde{O}(d^{-\frac{1}{2}})$ with high probability.

Note that $|v_c^\top w_{m,j}^0| = \tilde{O}(d^{-\frac{1}{2}})$ with high probability. Thus, combined with the assumption that $\mathrm{sgn}(\alpha_{m,j,i})$ is the same for all $i$ and $\rho = \tilde{O}(1)$, it holds that

$$\frac{i \tilde{\alpha}_{m,j,i,c}^t}{\sqrt{i!}} = \frac{i \alpha_{m,j,i}}{\sqrt{i!}} + \underbrace{\sum_{l=i+1}^{\infty} \left( \frac{l \alpha_{m,j,l}}{\sqrt{l!}} \binom{l-1}{l-i} (\rho w_{m,j}^t{}^\top v_c)^{l-i} \right)}_{= \tilde{\Omega}(d^{-\frac{1}{2}})} = \frac{i \alpha_{m,j,i}}{\sqrt{i!}} + \tilde{\Omega}(d^{-\frac{1}{2}}).$$

The same holds even when $\alpha_{m,j,i} < 0$ by considering the upper bound in the same manner. $\qquad \square$

We show that even as $w_{m,j}^t{}^\top v_c$ increases, the coefficient $\tilde{\alpha}_{c,m,j,i}^t$ remains bounded by a polylogarithmic function in $d$.

**Lemma C.10.** $\tilde{\alpha}_{m,j,i,c}^t = \frac{1}{\sqrt{i!}} \mathbb{E}_z \left[ a_{m,j,i} \sigma_m'(w_{m,j}^t{}^\top z + \rho w_{m,j}^t{}^\top v_c) \mathrm{He}_i(w_{m,j}^t{}^\top z) \right] = \tilde{O}(1)$, where $z \sim \mathcal{N}(0, I_d)$ and $\rho w_{m,j}^t{}^\top v_c = \tilde{O}(1)$.

*Proof.* Define $Z := w_{m,j}^t{}^\top z \sim \mathcal{N}(0,1)$, since $\|w_{m,j}^t\| = 1$. Also note that $\rho w_{m,j}^t{}^\top v_c = \tilde{O}(1)$, given $\|v_c\| = 1$ and $\rho = \tilde{O}(1)$. By the Cauchy–Schwarz inequality, we have

$$\mathbb{E}_Z \left[ a_{m,j} \sigma_m'(Z + \rho w_{m,j}^t{}^\top v_c + b_{m,j}) \mathrm{He}_i(Z) \right] \leq \sqrt{\mathbb{E}_Z[a_{m,j}{}^2 \sigma_m'(Z + \rho w_{m,j}^t{}^\top v_c + b_{m,j})^2]} \sqrt{\mathbb{E}_Z[\mathrm{He}_i(Z)^2]}$$

$$= \sqrt{i!} \sqrt{\mathbb{E}_Z[a_{m,j}{}^2 \sigma_m'(Z + \rho w_{m,j}^t{}^\top v_c + b_{m,j})^2]}.$$

We now provide casewise bounds according to the activation function $\sigma_m$.
If $\sigma_m(\cdot)$ is the ReLU function $\sigma_m(\cdot) = \max(0, \cdot)$,

$$\sqrt{\mathbb{E}_Z[a_{m,j}{}^2 \sigma_m'(Z + \rho w_{m,j}^t{}^\top v_c + b_{m,j})^2]} \leq |a_{m,j}|.$$

If $\sigma_m(\cdot)$ is a degree-$p$ polynomial $\sigma_m'(\cdot + b_{m,j}) = \sum_{q=0}^{p-1} C_q(\cdot)^q$ with $p = O(1)$,

$$\sqrt{\mathbb{E}_Z \left[ a_{m,j}^2 \sigma_m'(Z + \rho w_{m,j}^t{}^\top v_c + b_{m,j})^2 \right]} = |a_{m,j}| \sqrt{\mathbb{E}_Z \left[ \left( \sum_{q=0}^{p-1} C_q(Z + \rho w_{m,j}^t{}^\top v_c)^q \right)^2 \right]}$$

$$\leq |a_{m,j}| \sqrt{\sum_{q=0}^{p-1} \sum_{r=0}^{p-1} |C_q C_r| \mathbb{E}_Z \left[ (Z + \rho w_{m,j}^t{}^\top v_c)^{q+r} \right]}$$

$$\leq |a_{m,j}| \sqrt{\sum_{u=0}^{2(p-1)} C_u \mathbb{E}_Z \left[ (Z + \rho w_{m,j}^t{}^\top v_c)^u \right]}$$

$$= |a_{m,j}| \sqrt{\sum_{u=0}^{2(p-1)} C_u \sum_{j=0}^{u} \binom{u}{j} \mathbb{E}[Z^j] |\rho w_{m,j}^t{}^\top v_c|^{u-j}}$$

$$= \tilde{O}(1)|a_{m,j}|$$

where the constants $C_q$, $C_r$, and $C_u$ arise from the binomial expansion. Combining both cases, we obtain

$$\tilde{\alpha}_{c,m,j,i}^t = \frac{1}{\sqrt{i!}} \mathbb{E}_Z \left[ a_{m,j} \sigma_m'(Z + \rho w_{m,j}^t{}^\top v_c + b_{m,j}) \mathrm{He}_i(Z) \right] = \tilde{O}(1)$$

as desired. $\qquad\square$

*Remark* C.11. For the sake of conciseness in the exposition of the proof, we omit the superscript $t$ in $\tilde{\alpha}_{m,j,i,c}^t$. Based on Lemma C.8, Lemma C.9, and Lemma C.10, the bounds in the subsequent lemmas are properly justified, regardless of the variations in the coefficients $\tilde{\alpha}_{c,m,j,i}^t$.

To prove Lemma C.6, We introduce auxiliary sequences that provide the following bounds.

**Lemma C.12.** *Consider the expert $m \in \mathcal{M}_c$. Let $w_c^*{}^\top w_{c'}^* \leq A_4 d^{-\frac{1}{2}} = \tilde{O}(d^{-1/2})$ for all $c \neq c'$, $w_c^*{}^\top w_g^* \leq A_4 d^{-\frac{1}{2}} = \tilde{O}(d^{-1/2})$ for all $c$, and $\eta^t = \eta_e \leq a_4 d^{-\frac{k^*}{2}}$. For all $s = 0, 1, \ldots, t$, suppose that*

- $\kappa_{c_m^*, m, j_m^*}^s \leq a_2$,

- $|\kappa_{c,m,j}^s| \leq \kappa_{c_m^*, m, j_m^*}^s$ *for all* $(c,j) \neq (c_m^*, j_m^*)$,

- $|\kappa_{c,m,j}^s| \leq A_2 A_3 d^{-\frac{1}{2}}$ *for all* $(c,j) \neq (c_m^*, j_m^*)$,

- $|\kappa_{g,m,j}^s| \leq \kappa_{c_m^*, m, j_m^*}^s$ *for all* $j$,

- $|\kappa_{g,m,j}^s| \leq A_2 A_3 d^{-\frac{1}{2}}$ *for all* $j$.

*Then, by introducing auxiliary sequences $(P_I^s)_{s=0}^{t+1}$ and $(Q_I^s)_{s=0}^{t+1}$ characterized as follows:*

$$P_I^{s+1} = P_I^s + \frac{\eta^s}{CM^2} k^* \tilde{\alpha}_{m,j_m^*,k^*,c_m^*} \beta_{c_m^*,k^*} (P_I^s)^{k^*-1} \text{ with } P_I^0 = (1-a_2)\kappa_{c_m^*,m,j_m^*}^0$$

*and*

$$Q_I^{s+1} = Q_I^s + (1+a_2)\frac{\eta^s}{CM^2} k^* \max \left\{ \max_{c'} |\tilde{\alpha}_{m,j,k^*,c'} \beta_{c',k^*}|, \Big| \sum_{c' \in [C]} s_{c'} \tilde{\alpha}_{m,j,k^*,c'} \gamma_{k^*} \Big| \right\} (Q_I^s)^{k^*-1}$$

$$+ A_3 \frac{\eta^s}{CM^2} k^* \tilde{\alpha}_{m,j,k^*,c_m^*} \beta_{c_m^*,k^*} (\kappa_{c_m^*,m,j_m^*}^{s+1})^{k^*-1} d^{-\frac{1}{2}} \text{ with } Q_I^0 = (1+a_2) \max\{\max_c |\kappa_{c,m,j}^s|, |\kappa_{g,m,j}^s|, \frac{1}{2}d^{-\frac{1}{2}}\},$$

$\kappa_{c_m^*,m,j_m^*}^s$ *is lower bounded by $P_I^s$ for all $s = 0, 1, \ldots, t+1$ with high probability. For all $(c,j) \neq (c_m^*, j_m^*)$, $|\kappa_{c,m,j}^s|$ and $|\kappa_{g,m,j}^s|$ are upper bounded by $Q_I^s$ for all $s = 0, 1, \ldots, t+1$ with high probability.*

*Proof.* Suppose that $\kappa_{c_m^*,m,j_m^*}^s \leq a_2$, $|\kappa_{c,m,j}^s| \leq \kappa_{c_m^*,m,j_m^*}^s$ for all $c \neq c_m^*$ or $j \notin \mathcal{J}_m^*$, and $|\kappa_{g,m,j}^s| \leq A_2 A_3 d^{-\frac{1}{2}}$ for all $j \in [J]$ for all $s = 0, 1, ..., t$.

We first deduce the auxiliary sequence $(P_I^s)_{s=0}^{t+1}$.

$$\kappa_{c_m^*,m,j_m^*}^{s+1} \geq \kappa_{c_m^*,m,j_m^*}^s + \frac{\eta^s}{CM^2} \sum_{c \in [C]} \sum_{i=k^*}^{p^*} \left[ i\tilde{\alpha}_{m,j_m^*,i,c} \beta_{c,i} (\kappa_{c,m,j_m^*}^s)^{i-1} (w_{c_m^*}^*{}^\top w_c^* - \kappa_{c_m^*,m,j_m^*}^s \kappa_{c,m,j_m^*}^s) \right.$$

$$+ is_c \tilde{\alpha}_{m,j_m^*,i,c} \gamma_i (\kappa_{g,m,j_m^*}^s)^{i-1} (w_{c_m^*}^{* \top} w_g^* - \kappa_{c_m^*,m,j_m^*}^s \kappa_{g,m,j_m^*}^s) \Big]$$

$$- \frac{\kappa_{c_m^*,m,j_m^*}^s (\eta^s)^2 A_1{}^2 d}{2} - \frac{\eta^{s3} A_1{}^3 d^{\frac{3}{2}}}{2} - w_{c_m^*}^{* \top} (I_d - w_{m,j_m^*}^s w_{m,j_m^*}^{s \top}) \Xi^s{}_{w_{m,j}^s}$$

$$\geq \kappa_{c_m^*,m,j_m^*}^s + \frac{\eta^s}{CM^2} k^* \tilde{\alpha}_{m,j_m^*,k^*,c_m^*} \beta_{c_m^*,k^*} (1 - \kappa_{c_m^*,m,j_m^*}^s{}^2)(\kappa_{c_m^*,m,j_m^*}^s)^{k^*-1}$$

$$- \frac{\eta^s p^{*2}}{M^2} \max_{c,i} |\tilde{\alpha}_{m,j_m^*,i,c} \beta_{c,i}| \max_{c \neq c_m^*} |\kappa_{c,m,j_m^*}^s|^{k^*-1} \max_{c \neq c_m^*} |w_{c_m^*}^{* \top} w_c^*|$$

$$- \frac{\eta^s p^{*2}}{M^2} \max_{c,i} |\tilde{\alpha}_{m,j_m^*,i,c} \beta_{c,i}| \max_{c \neq c_m^*} |\kappa_{c,m,j_m^*}^s|^{k^*} - \frac{\eta^s p^{*2}}{M^2} \max_c |s_c| \max_{i,c} |\tilde{\alpha}_{m,j_m^*,i,c} \gamma_i| |\kappa_{g,m,j_m^*}^s|^{k^*-1} \max_{c \neq c_m^*} |w_c^{\top} w_g^*|$$

$$- \frac{\eta^s p^{*2}}{M^2} \max_c |s_c| \max_{i,c} |\tilde{\alpha}_{m,j_m^*,i,c} \gamma_i| |\kappa_{g,m,j_m^*}^s|^{k^*} - \kappa_{c_m^*,m,j_m^*}^s (\eta^s)^2 A_1{}^2 d + \eta^s w_{c_m^*}^{* \top} (I_d - w_{m,j_m^*}^s w_{m,j_m^*}^{s \top}) \Xi^s{}_{w_{m,j}}.$$

$$\geq \kappa_{c_m^*,m,j_m^*}^s + \frac{\eta^s}{CM^2} k^* \tilde{\alpha}_{m,j_m^*,k^*,c_m^*} \beta_{c_m^*,k^*} (1 - \kappa_{c_m^*,m,j_m^*}^s{}^2)(\kappa_{c_m^*,m,j_m^*}^s)^{k^*-1}$$

$$- \frac{\eta^s p^{*2}}{M^2} \max \left\{ \max_{c,i} |\tilde{\alpha}_{m,j_m^*,i,c} \beta_{c,i}|, \max_c |s_c| \max_{i,c} |\tilde{\alpha}_{m,j_m^*,i,c} \gamma_i| \right\} (\kappa_{c_m^*,m,j_m^*}^s)^{k^*-1}$$

$$\cdot (\max_{c \neq c_m^*} |w_{c_m^*}^{* \top} w_c^*| + \max_{c \neq c_m^*} |\kappa_{c,m,j_m^*}^s| + \max_{c \neq c_m^*} |w_c^{* \top} w_g^*| + |\kappa_{g,m,j_m^*}^s|)$$

$$- \kappa_{c_m^*,m,j_m^*}^s \eta^s a A_1{}^2 d^{-\frac{k^*-2}{2}} + \eta^s w_{c_m^*}^{* \top} (I_d - w_{m,j_m^*}^s w_{m,j_m^*}^{s \top}) \Xi^s{}_{w_{m,j}}.$$

We used conditions $w_c^{* \top} w_{c'}^* = \tilde{O}(d^{-\frac{1}{2}})$ for all $c \neq c'$ and $w_g^{* \top} w_c^* = \tilde{O}(d^{-\frac{1}{2}})$ for all $c$, $(\kappa_{c_m^*,m,j_m^*}^t)^2 \leq a_2{}^2 \leq \frac{1}{4} a_2$, $\frac{\eta^s p^{*2}}{M^2} \max \left\{ \max_{c,i} |\tilde{\alpha}_{m,j_m^*,i,c} \beta_{c,i}|, \max_c |s_c| \max_{i,c} |\tilde{\alpha}_{m,j_m^*,i,c} \gamma_i| \right\} (\kappa_{c_m^*,m,j_m^*}^s)^{k^*-1} (\max_{c \neq c_m^*} |w_{c_m^*}^{* \top} w_c^*| + \max_{c \neq c_m^*} |\kappa_{c,m,j_m^*}^s| + \max_{c \neq c_m^*} |w_c^{* \top} w_g^*| + |\kappa_{g,m,j_m^*}^s|) \leq \frac{a_2 \eta^s}{4CM^2} k^* \tilde{\alpha}_{m,j_m^*,k^*,c_m^*} \beta_{c_m^*,k^*} (\kappa_{c_m^*,m,j_m^*}^s)^{k^*-1}$, and $\kappa_{c_m^*,m,j_m^*}^s \eta^s a_4 A_1{}^2 d^{-\frac{k^*-2}{2}} \leq \frac{a_2 \eta^s}{4CM^2} k^* \tilde{\alpha}_{m,j_m^*,k^*,c_m^*} \beta_{c_m^*,k^*} (\kappa_{c_m^*,m,j_m^*}^s)^{k^*-1}$.

Hence,

$$\kappa_{c_m^*,m,j_m^*}^{s+1} \geq \kappa_{c_m^*,m,j_m^*}^s + (1 - \frac{3}{4} a_2) \frac{\eta^s}{CM^2} k^* \tilde{\alpha}_{m,j_m^*,k^*,c} \beta_{c_m^*,k^*} \left( \kappa_{c_m^*,m,j_m^*}^s \right)^{k^*-1}$$

$$+ \eta^s w_{c_m^*}^{* \top} (I_d - w_{m,j_m^*}^s w_{m,j_m^*}^{s \top}) \Xi^s{}_{w_{m,j}}$$

$$\geq \kappa_{c_m^*,m,j_m^*}^0 + \sum_{s'=0}^s \Big[ (1 - \frac{3}{4} a_2) \frac{\eta^{s'}}{CM^2} k^* \tilde{\alpha}_{m,j_m^*,k^*,c} \beta_{c_m^*,k^*} \left( \kappa_{c_m^*,m,j_m^*}^{s'} \right)^{k^*-1}$$

$$+ \eta^{s'} w_{c_m^*}^{* \top} (I_d - w_{m,j_m^*}^{s'} w_{m,j_m^*}^{s' \top}) \Xi^{s'}{}_{w_{m,j}} \Big].$$

We bound the noise term. Note that $\Xi^s{}_{w_{m,j}}$ has a sub-Weibull tail.

If $s \leq A_2 (\kappa_{c_m^*,m,j_m^*}^0)^{2-2k^*}$,

$$\sum_{s'=0}^s \eta^{s'} w_{c_m^*}^{* \top} (I_d - w_{m,j_m^*}^{s'} w_{m,j_m^*}^{s' \top}) \Xi^{s'}{}_{w_{m,j}^{s'}} \leq \eta_e A_1 \sqrt{s} \leq a_4 A_1 \kappa_{c_m^*,m,j_m^*}^0 \leq a_2 \kappa_{c_m^*,m,j_m^*}^0$$

with high probability.

If $s > A_2 (\kappa_{c_m^*,m,j_m^*}^0)^{2-2k^*}$,

$$\sum_{s'=0}^s \eta^{s'} w_{c_m^*}^{* \top} (I_d - w_{m,j_m^*}^{s'} w_{m,j_m^*}^{s' \top}) \Xi^{s'}{}_{w_{m,j}} \leq \eta^s A_1 s^{-\frac{1}{2}} s \leq \eta_e s A_1 A_2^{-\frac{1}{2}} (\kappa_{c_m^*,m,j_m^*}^{s'})^{k^*-1}$$

$$\leq \sum_{s'=0}^s \frac{a_2 \eta_e}{4CM^2} k^* \tilde{\alpha}_{m,j_m^*,k^*,c_m^*} \beta_{c_m^*,k^*} (\kappa_{c_m^*,m,j_m^*}^{s'})^{k^*-1}$$

with high probability.

Therefore, for all $s = 0, 1, \ldots, t$, $\kappa^s_{c^*_m, m, j^*_m}$ can be lower bounded as

$$\kappa^{s+1}_{c^*_m, m, j^*_m} \geq (1 - a_2)\kappa^s_{c^*_m, m, j^*_m} + \sum_{s'=0}^{s} \frac{(1 - a_2)\eta^{s'}}{CM^2} k^* \tilde{\alpha}_{m, j^*_m, k^*, c^*_m} \beta_{c^*_m, k^*} (\kappa^{s'}_{c^*_m, m, j^*_m})^{k^*-1}.$$

With the aid of an auxiliary sequence $(P^s_I)^{t+1}_{s=0}$, where $P^0_I = (1 - a_2)\kappa^0_{c^*_m, m, j^*_m}$, and

$$P^{s+1}_I = P^s_I + \frac{\eta^s}{CM^2} k^* \tilde{\alpha}_{m, j^*_m, k^*, c^*_m} \beta_{c^*_m, k^*} (P^s_I)^{k^*-1},$$

$\kappa^s_{c^*_m, m, j^*_m}$ is lower bounded by $P^s_I$ for all $s = 0, 1, \ldots, t+1$.

Next, we deduce the auxiliary sequence $(Q^s_I)^{t+1}_{s=0}$.

For the upper bound of $\max_{c \neq c^*_m \text{ or } j \notin \mathcal{J}^*_m} |\kappa^{s+1}_{c, m, j}|$, we have $|\kappa^{s+1}_{c, m, j} - \kappa^s_{c, m, j}| \leq A_1 \eta_e$ with high probability. Thus, the sign of $\kappa^{s+1}_{c, m, j}$ is the same as that of $\kappa^s_{c, m, j}$, or $|\kappa^{s+1}_{c, m, j}| \leq A_1 \eta_e$. Similarly, the sign of $\kappa^{s+1}_{g, m, j}$ is the same as that of $\kappa^s_{g, m, j}$, or $|\kappa^{s+1}_{g, m, j}| \leq A_1 \eta_e$.

Fix $m \in [M]$.

We show that the following bounds hold for all $s = 0, 1, \ldots, t+1$.

- $\max_{c \neq c^*_m} |\kappa^s_{c, m, j}|$ is upper bounded by $Q^s_{I, c}$ for all $j \in \mathcal{J}^*_m$,
  where the sequence $(Q^s_{I, c})^{t+1}_{s=0}$ is defined recursively as follows:

$$Q^0_{I, c} = (1 + a_2) \max\{|\kappa^s_{c, m, j}|, \frac{1}{2} d^{-\frac{1}{2}}\},$$

  and for $s \geq 0$,

$$Q^{s+1}_{I, c} = Q^s_{I, c} + (1 + a_2) \frac{\eta^s}{CM^2} k^* \max_{c'} |\tilde{\alpha}_{m, j, k^*, c'} \beta_{c', k^*}| (Q^s_{I, c})^{k^*-1} + A_3 \frac{\eta^s}{CM^2} k^* \tilde{\alpha}_{m, j, k^*, c^*_m} \beta_{c^*_m, k^*} (\kappa^s_{c^*_m, m, j^*_m})^{k^*-1} d^{-\frac{1}{2}},$$

  with high probability.

- $|\kappa^s_{g, m, j}|$ is upper bounded by $Q^s_{I, g}$ for all $j \in \mathcal{J}^*_m$,
  where the sequence $(Q^s_{I, g})^{t+1}_{s=0}$ is defined recursively as follows:

$$Q^0_{I, g} = (1 + a_2) \max\{|\kappa^s_{g, m, j}|, \frac{1}{2} d^{-\frac{1}{2}}\},$$

  and for $s \geq 0$,

$$Q^{s+1}_{I, g} = Q^s_{I, g} + (1 + a_2) \frac{\eta^s}{CM^2} k^* |\sum_{c' \in [C]} s_{c'} \tilde{\alpha}_{m, j, k^*, c'} \gamma_{k^*}| (Q^s_{I, g})^{k^*-1} + A_3 \frac{\eta^s}{CM^2} k^* \tilde{\alpha}_{m, j, k^*, c^*_m} \beta_{c^*_m, k^*} (\kappa^s_{c^*_m, m, j^*_m})^{k^*-1} d^{-\frac{1}{2}},$$

  with high probability.

- $\max_{c \in [C]} |\kappa^s_{c, m, j}|$ is upper bounded by $R^s_{I, c}$ for all $j \notin \mathcal{J}^*_m$,
  where the sequence $(R^s_{I, c})^{t+1}_{s=0}$ is defined recursively as follows:

$$R^0_{I, c} = (1 + a_2) \max\{\max_{c \neq c^*_m} |\kappa^s_{c, m, j}|, \frac{1}{2} d^{-\frac{1}{2}}\},$$

and for $s \geq 0$,

$$R_{\mathrm{I},c}^{s+1} = R_{\mathrm{I},c}^s + (1+a_2)\frac{\eta^s}{CM^2}k^* \max_{c'}|\tilde{\alpha}_{m,j,k^*,c'}\beta_{c',k^*}|(R_{\mathrm{I},c}^s)^{k^*-1},$$

with high probability.

- $|\kappa_{g,m,j}^s|$ is upper bounded by $R_{\mathrm{I},g}^s$ for all $j \notin \mathcal{J}_m^*$,
  where the sequence $\left(R_{\mathrm{I},g}^s\right)_{s=0}^{t+1}$ is defined recursively as follows:

$$R_{\mathrm{I},g}^0 = (1+a_2)\max\{|\kappa_{g,m,j}^s|, \tfrac{1}{2}d^{-\frac{1}{2}}\},$$

and for $s \geq 0$,

$$R_{\mathrm{I},g}^{s+1} = R_{\mathrm{I},g}^s + (1+a_2)\frac{\eta^s}{CM^2}k^*|\sum_{c'\in[C]} s_{c'}\tilde{\alpha}_{m,j,k^*,c'}\gamma_{k^*}|(R_{\mathrm{I},g}^s)^{k^*-1},$$

with high probability.

Furthermore, for all $s = 0, 1, \ldots, t+1$, $Q_{\mathrm{I},c}^s$, $Q_{\mathrm{I},g}^s$, $R_{\mathrm{I},c}^s$, and $R_{\mathrm{I},g}^s$ are upper bounded by $Q_{\mathrm{I}}^s$.

We sequentially present each bound.

For $c \neq c_m^*$ and $j \in \mathcal{J}_m^*$,

$$|\kappa_{c,m,j}^{s+1}| \leq \max\Big\{ A_1\eta_e, \Big|\kappa_{c,m,j}^s + \frac{\eta^s}{CM^2}\sum_{c\in[C]}\sum_{i=k^*}^{p^*}\Big(i\tilde{\alpha}_{m,j,i,c'}\beta_{c',i}(\kappa_{c',m,j}^s)^{i-1}\big(w_c^{*\top}w_{c'}^* - \kappa_{c,m,j}^s\kappa_{c',m,j}^s\big)$$

$$+ i\tilde{\alpha}_{m,j,i,c'}s_{c'}\gamma_i(\kappa_{g,m,j}^s)^{i-1}\big(w_c^{*\top}w_g^* - \kappa_{c,m,j}^s\kappa_{g,m,j}^s\big)\Big) + \frac{|\kappa_{c,m,j}^s|\eta^{s2}A_1{}^2 d}{2} + \frac{\eta^{s3}A_1{}^3 d^{\frac{3}{2}}}{2}$$

$$+ w_c^{*\top}(I_d - w_{m,j}^s w_{m,j}^s{}^\top)\Xi^s{}_{w_{m,j}}\Big|\Big\}$$

$$\leq \max\Big\{ A_1\eta_e, \Big|\kappa_{c,m,j}^s + \frac{\eta^s}{CM^2}\Big(|\sum_{i=k^*}^{p^*}i\tilde{\alpha}_{m,j,i,c_m^*}\beta_{c_m^*,i}(\kappa_{c_m^*,m,j}^s)^{i-1}(w_c^{*\top}w_{c_m^*}^*)|$$

$$+ |\sum_{i=k^*}^{p^*}i\tilde{\alpha}_{m,j,i,c}\beta_{c,i}(\kappa_{c,m,j}^s)^{i-1}(1-(\kappa_{c,m,j}^s)^2)|$$

$$+ |\sum_{c'\neq c,c_m^*}\sum_{i=k^*}^{p^*}i\tilde{\alpha}_{m,j,i,c'}\beta_{c',i}(\kappa_{c',m,j}^s)^{i-1}(w_c^{*\top}w_{c'}^* - \kappa_{c,m,j}^s\kappa_{c',m,j}^s)|$$

$$+ |\sum_{c'\in[C]}\sum_{i=k^*}^{p^*}is_{c'}\tilde{\alpha}_{m,j,i,c'}\gamma_i(\kappa_{g,m,j}^s)^{i-1}(w_c^{*\top}w_g^*)| + |\sum_{c'\in[C]}\sum_{i=k^*}^{p^*}is_{c'}\tilde{\alpha}_{m,j,i,c'}\gamma_i(\kappa_{g,m,j}^s)^i|\Big)$$

$$+ \frac{|\kappa_{c,m,j}^s|(\eta^s)^2 A_1{}^2 d}{2} + \frac{(\eta^s)^3 A_1{}^3 d^{\frac{3}{2}}}{2} + w_c^{*\top}(I_d - w_{m,j}^s w_{m,j}^s{}^\top)\Xi^s{}_{w_{m,j}}\Big|\Big\}$$

$$\leq \max\Big\{ A_1\eta_e, \Big|\kappa_{c,m,j}^s + \frac{\eta^s}{CM^2}\Big(p^{*2}\max_i|\tilde{\alpha}_{m,j,i,c_m^*}\beta_{c_m^*,i}|(\kappa_{c_m^*,m,j}^s)^{k^*-1}(w_c^{*\top}w_{c_m^*}^*)$$

$$+ k^*\max_i|\tilde{\alpha}_{m,j,k^*,c}\beta_{c,k^*}||\kappa_{c,m,j}^s|^{k^*-1} + p^{*2}\max_{c',i}|\tilde{\alpha}_{m,j,i,c'}\beta_{c',i}||\kappa_{c,m,j}^s|^{k^*}$$

$$+ Cp^{*2}\max_{c',i}|\tilde{\alpha}_{m,j,i,c'}\beta_{c',i}|\max_{c'\neq c_m^*}|\kappa_{c,m,j}^s|^{k^*}(\max_{c'\neq c}|w_c^{*\top}w_{c'}^*| + \max_{c'\neq c,c_m^*}|\kappa_{c',m,j}^s|)$$

$$+ Cp^{*2}\max_{c',i}|s_{c'}\tilde{\alpha}_{m,j,i,c'}\gamma_i||\kappa_{g,m,j}^s|^{k^*-1}(|w_c^{*\top}w_g^*| + |\kappa_{g,m,j}^s|)\Big)$$

$$+ \frac{|\kappa^s_{c,m,j}|(\eta^s)^2 A_1^2 d}{2} + \frac{(\eta^s)^3 A_1^3 d^{\frac{3}{2}}}{2} + \eta^s w_c^{*\top}(I_d - w^s_{m,j}w^{s\,\top}_{m,j})\Xi^s{}_{w^s_{m,j}}\bigg|\bigg\}$$

$$\leq \max\bigg\{ A_1\eta_e, \bigg|\kappa^s_{c,m,j} + (1 + \frac{1}{3}a_2)\frac{\eta^s}{CM^2}k^* \max_{c'}|\tilde{\alpha}_{m,j,k^*,c'}\beta_{c',k^*}||\kappa^s_{c,m,j}|^{k^*-1}$$

$$+ A_3\frac{\eta^s}{CM^2}k^*\tilde{\alpha}_{m,j,k^*,c^*_m}\beta_{c^*_m,k^*}(\kappa^s_{c^*_m,m,j^*_m})^{k^*-1}d^{-\frac{1}{2}} + \frac{|\kappa^s_{c,m,j}|(\eta^s)^2 A_1^2 d}{2} + \frac{(\eta^s)^3 A_1^3 d^{\frac{3}{2}}}{2}$$

$$+ \eta^s w_c^{*\top}(I_d - w^s_{m,j}w^{s\,\top}_{m,j})\Xi^s{}_{w_{m,j}}\bigg|\bigg\}$$

$$\leq \max\bigg\{ A_1\eta_e, \max_{c\neq c^*_m}|\kappa^0_{c,m,j}|$$

$$+ (1 + \frac{2}{3}a_2)\sum_{s'=0}^s \frac{\eta^{s'}}{CM^2}k^* \max_{c'}|\tilde{\alpha}_{m,j,k^*,c'}\beta_{c',k^*}|\max\bigg\{\max_{c\neq c^*_m}|\kappa^{s'}_{c,m,j}|^{k^*-1}, (\frac{1}{2}d^{-\frac{1}{2}})^{k^*-1}\bigg\}$$

$$+ A_3\sum_{s'=0}^s \frac{\eta^{s'}}{CM^2}k^*\tilde{\alpha}_{m,j,k^*,c^*_m}\beta_{c^*_m,k^*}(\kappa^{s'}_{c^*_m,m,j^*_m})^{k^*-1}d^{-\frac{1}{2}} + \max_{c\neq c^*_m}\bigg|\sum_{s'=0}^s \eta^{s'} w_c^{*\top}(I_d - w^{s'}_{m,j}w^{s'\,\top}_{m,j})\Xi^{s'}{}_{w_{m,j}}\bigg|\bigg\}.$$

Since, $w_c^{*\top}w^*_{c'} = \tilde{O}(d^{-\frac{1}{2}})$ for all $c \neq c'$, $w_g^{*\top}w^*_c = \tilde{O}(d^{-\frac{1}{2}})$ for all $c$, $\kappa^s_{c,m,j} = \tilde{O}(d^{-\frac{1}{2}})$ for all $(c,j) \neq (c^*_m, j^*_m)$ and $\kappa^s_{g,m,j} = \tilde{O}(d^{-\frac{1}{2}})$ for all $j \in [J]$, the term $k^* \max_i|\tilde{\alpha}_{m,j,k^*,c}\beta_{c,k^*}||\kappa^s_{c,m,j}|^{k^*-1}$ subsumes the remaining terms within the expression $\frac{\eta^s}{CM^2}(\cdot)$ with wight $\frac{1}{3}a_2$ in the fourth inequality. $\frac{|\kappa^s_{c,m,j}|(\eta^s)^2 A_1^2 d}{2} + \frac{(\eta^s)^3 A_1^3 d^{\frac{3}{2}}}{2}$ are subsumed with weight $\frac{1}{3}a_2$ by $\eta^s \leq a_4 d^{-\frac{k^*}{2}}$ and $|\kappa^s_{c,m,j}| \leq A_2 A_3 d^{-\frac{1}{2}}$ in the fifth inequality.

For the noise term, if $s \leq A_2 d^{k^*-1}$,

$$\max_{c\neq c^*_m}\bigg|\sum_{s'=0}^s \eta^{s'} w_c^{*\top}(I_d - w^{s'}_{m,j}w^{s'\,\top}_{m,j})\Xi^{s'}{}_{w_{m,j}}\bigg| \leq \eta_e A_1\sqrt{s} \leq \frac{1}{2}a_2 d^{-\frac{1}{2}}$$

with high probability.

If $s > A_2 d^{k^*-1}$,

$$\max_{c\neq c^*_m}\bigg|\sum_{s'=0}^s \eta^{s'} w_c^{*\top}(I_d - w^{s'}_{m,j}w^{s'\,\top}_{m,j})\Xi^{s'}{}_{w_{m,j}}\bigg| \leq \eta_e A_1\sqrt{s} \leq \frac{a_2\eta_e s}{3CM^2}k^* \max_{c'}|\tilde{\alpha}_{m,j,k^*,c'}\beta_{c',k^*}|(\frac{1}{2}d^{-\frac{1}{2}})^{k^*-1}$$

with high probability.

Thus,

$$\max_{c\neq c^*_m}|\kappa^s_{c,m,j}| \leq (1 + a_2)\max\bigg\{\max_{c\neq c^*_m}|\kappa^0_{c,m,j}|, \frac{1}{2}d^{-\frac{1}{2}}\bigg\}$$

$$+ (1 + a_2)\sum_{s'=0}^s \frac{\eta^{s'}}{CM^2}k^* \max_{c'}|\tilde{\alpha}_{m,j,k^*,c'}\beta_{c',k^*}|\max\bigg\{\max_{c\neq c^*_m}|\kappa^{s'}_{c,m,j}|, Q^{s'}_{\mathrm{I},c}\bigg\}^{k^*-1}$$

$$+ A_3\sum_{s'=0}^s \frac{\eta^{s'}}{CM^2}k^*\tilde{\alpha}_{m,j,k^*,c^*_m}\beta_{c^*_m,k^*}(\kappa^{s'}_{c^*_m,m,j^*_m})^{k^*-1}d^{-\frac{1}{2}}.$$

In contrast, the following inequality holds for $Q^s_{\mathrm{I},c}$:

$$Q^{s+1}_{\mathrm{I},c} \leq (1 + a_2)\max\{\max_{c\neq c^*_m}|\kappa^s_{c,m,j}|, \frac{1}{2}d^{-\frac{1}{2}}\} + (1 + a_2)\sum_{s'=0}^s \frac{\eta^{s'}}{CM^2}k^* \max_{c'}|\tilde{\alpha}_{m,j,k^*,c'}\beta_{c',k^*}|(Q^{s'}_{\mathrm{I},c})^{k^*-1}$$

$$+ A_3\sum_{s'=0}^s \frac{\eta^{s'}}{CM^2}k^*\tilde{\alpha}_{m,j,k^*,c^*_m}\beta_{c^*_m,k^*}(\kappa^{s'}_{c^*_m,m,j^*_m})^{k^*-1}d^{-\frac{1}{2}}.$$

Therefore, by induction, we establish that $\max_{c\neq c^*_m}|\kappa^s_{c,m,j}|$ is upper bounded by $Q^s_{\mathrm{I},c}$ for all $j \in \mathcal{J}^*_m$ and $s = 0, 1, \ldots, t+1$ with high probability.

We apply a similar procedure for $g$ and $j \in \mathcal{J}_m^*$.

$$
\begin{aligned}
|\kappa_{g,m,j}^{s+1}| &\leq \max \Big\{ A_1 \eta_e, \Big| \kappa_{g,m,j}^s + \frac{\eta^s}{CM^2} \sum_{c' \in [C]} \sum_{i=k^*}^{p^*} \Big[ i\tilde{\alpha}_{m,j,i,c'} \beta_{c',i} (\kappa_{c',m,j}^s)^{i-1} (w_g^{*\top} w_{c'}^* - \kappa_{g,m,j}^s \kappa_{c',m,j}^s) \\
&\quad + i\tilde{\alpha}_{m,j,i,c'} s_{c'} \gamma_i (\kappa_{g,m,j}^s)^{i-1} (1 - (\kappa_{g,m,j}^s)^2) \Big] + \frac{|\kappa_{g,m,j}^s|(\eta^s)^2 A_1^2 d}{2} + \frac{(\eta^s)^3 A_1^3 d^{\frac{3}{2}}}{2} \\
&\quad + w_g^{*\top} (I_d - w_{m,j}^s w_{m,j}^{s\top}) \Xi^s{}_{w_{m,j}} \Big| \Big\} \\
&\leq \max \Big\{ A_1 \eta_e, \Big| \kappa_{g,m,j}^s + \frac{\eta^s}{CM^2} \Big( | \sum_{i=k^*}^{p^*} i\tilde{\alpha}_{m,j,i,c_m^*} \beta_{c_m^*,i} (\kappa_{c_m^*,m,j}^s)^{i-1} (w_g^{*\top} w_{c_m^*}^* - \kappa_{g,m,j}^t \kappa_{c_m^*,m,j}^t)| \\
&\quad + | \sum_{c' \neq c_m^*} \sum_{i=k^*}^{p^*} i\tilde{\alpha}_{m,j,i,c'} \beta_{c',i} (\kappa_{c',m,j}^s)^{i-1} (w_g^{*\top} w_{c'}^* - \kappa_{g,m,j}^s \kappa_{c',m,j}^s)| \\
&\quad + | \sum_{c' \in [C]} \sum_{i=k^*}^{p^*} i s_{c'} \tilde{\alpha}_{m,j,i,c'} \gamma_i (\kappa_{g,m,j}^s)^{i-1} (1 - (\kappa_{g,m,j}^s)^2)| \Big) \\
&\quad + \frac{|\kappa_{g,m,j}^s|(\eta^s)^2 A_1^2 d}{2} + \frac{(\eta^s)^3 A_1^3 d^{\frac{3}{2}}}{2} + w_g^{*\top} (I_d - w_{m,j}^s w_{m,j}^{s\top}) \Xi^s{}_{w_{m,j}} \Big| \Big\} \\
&\leq \max \Big\{ A_1 \eta_e, \Big| \kappa_{g,m,j}^s + \frac{\eta^s}{CM^2} \Big( k^* \tilde{\alpha}_{m,j,k^*,c_m^*} \beta_{c_m^*,k^*} (\kappa_{c_m^*,m,j}^s)^{k^*-1} (|w_g^{*\top} w_{c_m^*}^*| + |\kappa_{g,m,j}^s|) \\
&\quad + p^{*2} \max_i |\tilde{\alpha}_{m,j,i,c_m^*} \beta_{c_m^*,i}| (\kappa_{c_m^*,m,j}^t)^{k^*} (|w_g^{*\top} w_{c_m^*}^*| + |\kappa_{g,m,j}^s|) \\
&\quad + Cp^{*2} \max_{c' \neq c_m^*, i} |\tilde{\alpha}_{m,j,i,c'} \beta_{c',i}| \max_{c' \neq c_m^*} |\kappa_{c',m,j}^s|^{k^*-1} (\max_{c' \neq c_m^*} |w_g^{*\top} w_{c'}^*| + \max_{c' \neq c_m^*} |\kappa_{c',m,j}^t|) \\
&\quad + k^* | \sum_{c' \in [C]} s_{c'} \tilde{\alpha}_{m,j,k^*,c'} \gamma_{k^*} | |\kappa_{g,m,j}^t|^{k^*-1} + Cp^{*2} \max_{c',i} |s_{c'} \tilde{\alpha}_{m,j,i,c'} \gamma_i| |\kappa_{g,m,j}^s|^{k^*} \Big) \\
&\quad + \frac{|\kappa_{g,m,j}^s|(\eta^s)^2 A_1^2 d}{2} + \frac{(\eta^s)^3 A_1^3 d^{\frac{3}{2}}}{2} + \eta^s w_g^{*\top} (I_d - w_{m,j}^s w_{m,j}^{s\top}) \Xi^s{}_{w_{m,j}} \Big| \Big\} \\
&\leq \max \Big\{ A_1 \eta_e, |\kappa_{g,m,j}^0| + (1 + \frac{2}{3} a_2) \sum_{s'=0}^s \frac{\eta^{s'}}{CM^2} k^* | \sum_{c' \in [C]} s_{c'} \tilde{\alpha}_{m,j,k^*,c'} \gamma_{k^*} | \max \{ |\kappa_{g,m,j}^{s'}|^{k^*-1}, (\frac{1}{2} d^{-\frac{1}{2}})^{k^*-1} \} \\
&\quad + A_3 \sum_{s'=0}^s \frac{\eta^s}{CM^2} k^* \tilde{\alpha}_{m,j,k^*,c_m^*} \beta_{c_m^*,k^*} (\kappa_{c_m^*,m,j_m^*}^s)^{k^*-1} d^{-\frac{1}{2}} + \Big| \sum_{s'=0}^s \eta^s w_g^{*\top} (I_d - w_{m,j}^s w_{m,j}^{s\top}) \Xi^s_{w_{m,j}} \Big| \Big\}.
\end{aligned}
$$

Since, $w_c^{*\top} w_{c'}^* = \tilde{O}(d^{-\frac{1}{2}})$ for all $c \neq c'$, $w_g^{*\top} w_c^* = \tilde{O}(d^{-\frac{1}{2}})$ for all $c$, $\kappa_{c,m,j}^s = \tilde{O}(d^{-\frac{1}{2}})$ for all $(c,j) \neq (c_m^*, j_m^*)$ and $\kappa_{g,m,j}^s = \tilde{O}(d^{-\frac{1}{2}})$ for all $j \in [J]$, the term $k^* \max_i |\sum_{c' \in [C]} s_{c'} \tilde{\alpha}_{m,j,k^*,c'} \gamma_{k^*}| |\kappa_{g,m,j}^s|^{k^*-1}$ subsumes the remaining terms within the expression $\frac{\eta^s}{CM^2} (\cdot)$ with weight $\frac{1}{3} a_2$ in the fourth inequality. $\frac{|\kappa_{g,m,j}^s|(\eta^s)^2 A_1^2 d}{2} + \frac{(\eta^s)^3 A_1^3 d^{\frac{3}{2}}}{2}$ are also subsumed with weight $\frac{1}{3} a_2$ by $\eta^s \leq a_4 d^{-\frac{k^*}{2}}$ and $|\kappa_{c,m,j}^s| \leq A_2 A_3 d^{-\frac{1}{2}}$ in the fourth inequality.

By providing an upper bound for the noise term in the same way,

$$
\Big| \sum_{s'=0}^s \eta^s w_g^{*\top} (I_d - w_{m,j}^s w_{m,j}^{s\top}) \Xi^s{}_{w_{m,j}} \Big| \leq \begin{cases} \frac{1}{2} a_2 d^{-\frac{1}{2}}, & \text{if } s \leq A_2 d^{k^*-1}, \\ \frac{a_2 \eta_e s}{3CM^2} k^* | \sum_{c' \in [C]} s_{c'} \tilde{\alpha}_{m,j,k^*,c'} \gamma_{k^*} | (\frac{1}{2} d^{-\frac{1}{2}})^{k^*-1}, & \text{if } s > A_2 d^{k^*-1} \end{cases}
$$

with high probability.

Thus,

$$
|\kappa_{g,m,j}^s| \leq (1 + a_2) \max \Big\{ |\kappa_{g,m,j}^0|, \frac{1}{2} d^{-\frac{1}{2}} \Big\}
$$

$$+ (1 + a_2) \sum_{s'=0}^{s} \frac{\eta^{s'}}{CM^2} k^* | \sum_{c' \in [C]} s_{c'} \tilde{\alpha}_{m,j,k^*,c'} \gamma_{k^*} | \max \left\{ |\kappa_{g,m,j}^{s'}|, Q_{I,g}^{s'} \right\}^{k^*-1}$$

$$+ A_3 \sum_{s'=0}^{s} \frac{\eta^{s'}}{CM^2} k^* \tilde{\alpha}_{m,j,k^*,c_m^*} \beta_{c_m^*,k^*} (\kappa_{c_m^*,m,j_m^*}^{s'})^{k^*-1} d^{-\frac{1}{2}}.$$

In contrast, the following inequality holds for $Q_{I,g}^s$:

$$Q_{I,g}^{s+1} \le (1 + a_2) \max\{ \max_{c \ne c_m^*} |\kappa_{g,m,j}^s|, \frac{1}{2} d^{-\frac{1}{2}} \} + (1 + a_2) \sum_{s'=0}^{s} \frac{\eta^{s'}}{CM^2} k^* | \sum_{c' \in [C]} s_{c'} \tilde{\alpha}_{m,j,k^*,c'} \gamma_{k^*} | (Q_{I,g}^{s'})^{k^*-1}$$

$$+ A_3 \sum_{s'=0}^{s} \frac{\eta^{s'}}{CM^2} k^* \tilde{\alpha}_{m,j,k^*,c_m^*} \beta_{c_m^*,k^*} (\kappa_{c_m^*,m,j_m^*}^{s'})^{k^*-1} d^{-\frac{1}{2}}.$$

Therefore, by induction, we establish that $|\kappa_{g,m,j}^s|$ is upper bounded by $Q_{I,g}^s$ for all $j \in \mathcal{J}_m^*$ and $s = 0, 1, \dots, t+1$ with high probability.

We also consider a similar argument for $j \notin \mathcal{J}_m^*$.

$$|\kappa_{c,m,j}^{s+1}| \le \max \Big\{ A_1 \eta_e, \Big| \kappa_{c,m,j}^s + \frac{\eta^s}{CM^2} \sum_{c \in [C]} \sum_{i=k^*}^{p^*} \Big[ i \tilde{\alpha}_{m,j,i,c'} \beta_{c',i} (\kappa_{c',m,j}^s)^{i-1} (w_c^{*\top} w_{c'}^* - \kappa_{c,m,j}^s \kappa_{c',m,j}^s)$$

$$+ i \tilde{\alpha}_{m,j,i,c'} s_{c'} \gamma_i (\kappa_{g,m,j}^s)^{i-1} (w_c^{*\top} w_g^* - \kappa_{c,m,j}^s \kappa_{g,m,j}^s) \Big] + \frac{|\kappa_{c,m,j}^s|(\eta^s)^2 A_1^2 d}{2} + \frac{(\eta^s)^3 A_1^3 d^{\frac{3}{2}}}{2}$$

$$+ \eta^s w_c^{*\top} (I_d - w_{m,j}^s w_{m,j}^{s\top}) \Xi^s{}_{w_{m,j}} \Big| \Big\}$$

$$\le \max \Big\{ A_1 \eta_e, \Big| \kappa_{c,m,j}^s + \frac{\eta^s}{CM^2} \Big( | \sum_{i=k^*}^{p^*} i \tilde{\alpha}_{m,j,i,c} \beta_{c,i} (\kappa_{c,m,j}^s)^{i-1} (1 - (\kappa_{c,m,j}^s)^2) |$$

$$+ | \sum_{c' \ne c} \sum_{i=k^*}^{p^*} i \tilde{\alpha}_{m,j,i,c'} \beta_{c',i} (\kappa_{c',m,j}^s)^{i-1} (w_c^{*\top} w_{c'}^* - \kappa_{c,m,j}^s \kappa_{c',m,j}^s) |$$

$$+ | \sum_{c' \in [C]} \sum_{i=k^*}^{p^*} i s_{c'} \tilde{\alpha}_{m,j,i,c'} \gamma_i (\kappa_{g,m,j}^s)^{i-1} (w_c^{*\top} w_g^*) + \sum_{c' \in [C]} \sum_{i=k^*}^{p^*} i s_{c'} \tilde{\alpha}_{m,j,i,c'} \gamma_i (\kappa_{g,m,j}^s)^i | \Big)$$

$$+ \frac{|\kappa_{c,m,j}^s|(\eta^s)^2 A_1^2 d}{2} + \frac{(\eta^s)^3 A_1^3 d^{\frac{3}{2}}}{2} + w_c^{*\top} (I_d - w_{m,j}^s w_{m,j}^{s\top}) \Xi^s{}_{w_{m,j}} \Big| \Big\}$$

$$\le \max \Big\{ A_1 \eta_e, \Big| \kappa_{c,m,j}^s + \frac{\eta^s}{CM^2} \Big( k^* |\tilde{\alpha}_{m,j,k^*,c} \beta_{c,k^*}| |\kappa_{c,m,j}^s|^{k^*-1} + p^{*2} \max_i |\tilde{\alpha}_{m,j,i,c} \beta_{c,i}| |\kappa_{c,m,j}^s|^{k^*}$$

$$+ Cp^{*2} \max_{c' \ne c,i} |\tilde{\alpha}_{m,j,i,c'} \beta_{c',i}| \max_{c' \ne c} |\kappa_{c',m,j}^s|^{k^*-1} (\max_{c' \ne c} |w_c^{*\top} w_{c'}^*| + |\kappa_{c,m,j}^s|)$$

$$+ Cp^{*2} \max_{c' \ne c,i} |\tilde{\alpha}_{m,j,i,c'} \beta_{c',i}| |\kappa_{g,m,j}^s|^{k^*-1} (|w_c^{*\top} w_g^*| + |\kappa_{g,m,j}^s|) \Big)$$

$$+ \frac{|\kappa_{c,m,j}^s|(\eta^s)^2 A_1^2 d}{2} + \frac{(\eta^s)^3 A_1^3 d^{\frac{3}{2}}}{2} + \eta^s w_c^{*\top} (I_d - w_{m,j}^s w_{m,j}^{s\top}) \Xi^s{}_{w_{m,j}} \Big| \Big\}$$

$$\le \max \Big\{ A_1 \eta_e, \max_{c \ne c_m^*} |\kappa_{c,m,j}^0| + (1 + \frac{2}{3} a_2) \sum_{s'=0}^{s} \frac{\eta^{s'}}{CM^2} k^* \max_{c'} |\tilde{\alpha}_{m,j,k^*,c'} \beta_{c',k^*}|$$

$$\cdot \max \Big\{ \max_{c \ne c_m^*} |\kappa_{c,m,j}^{s'}|^{k^*-1}, (\frac{1}{2} d^{-\frac{1}{2}})^{k^*-1} \Big\} + \max_{c \ne c_m^*} \Big| \sum_{s'=0}^{s} \eta^{s'} w_c^{*\top} (I_d - w_{m,j}^s w_{m,j}^{s\top}) \Xi^s{}_{w_{m,j}} \Big| \Big\}.$$

Since, $w_c^{*\top} w_{c'}^* = \tilde{O}(d^{-\frac{1}{2}})$ for all $c \ne c'$, $w_g^{*\top} w_c^* = \tilde{O}(d^{-\frac{1}{2}})$ for all $c$, $\kappa_{c,m,j}^s = \tilde{O}(d^{-\frac{1}{2}})$ for all $(c,j) \ne (c_m^*, j_m^*)$ and $\kappa_{g,m,j}^s = \tilde{O}(d^{-\frac{1}{2}})$ for all $j \in [J]$, the term $k^* \max_i |\tilde{\alpha}_{m,j,k^*,c} \beta_{c,k^*}| |\kappa_{c,m,j}^s|^{k^*-1}$ subsumes the remaining terms within the

expression $\frac{\eta^s}{CM^2}(\cdot)$ with wight $\frac{1}{3}a_2$ in the fourth inequality. $\frac{|\kappa^s_{c,m,j}|(\eta^s)^2A_1{}^2d}{2} + \frac{(\eta^s)^3A_1{}^3d^{\frac{3}{2}}}{2}$ are also subsumed with weight $\frac{1}{3}a_2$ by $\eta^s \leq a_4 d^{-\frac{k^*}{2}}$ and $|\kappa^s_{c,m,j}| \leq A_2A_3d^{-\frac{1}{2}}$ in the fourth inequality.

By upper bounding the noise term in the same way, we have

$$\Big|\sum_{s'=0}^{s}\eta^s w_c^{*\top}(I_d - w^s_{m,j}w^s_{m,j}{}^\top)\Xi^s w_{m,j}\Big| \leq \begin{cases} \frac{1}{2}a_2 d^{-\frac{1}{2}}, & \text{if } s \leq A_2 d^{k^*-1}, \\ \frac{a_2\eta_e s}{3CM^2}k^*\max_{c'}|\tilde{\alpha}_{m,j,k^*,c'}\beta_{c',k^*}|(\frac{1}{2}d^{-\frac{1}{2}})^{k^*-1}, & \text{if } s > A_2 d^{k^*-1} \end{cases}$$

with high probability.

Thus,

$$\max_{c\in[C]}|\kappa^s_{c,m,j}| \leq (1+a_2)\max\{\max_{c\neq c^*_m}|\kappa^0_{c,m,j}|, \frac{1}{2}d^{-\frac{1}{2}}\}$$
$$+ (1+a_2)\sum_{s'=0}^{s}\frac{\eta^{s'}}{CM^2}k^*\max_{c'}|\tilde{\alpha}_{m,j,k^*,c'}\beta_{c',k^*}|\max\Big\{\max_{c\neq c^*_m}|\kappa^{s'}_{c,m,j}|, R^{s'}_{I,c}\Big\}^{k^*-1}.$$

In contrast, the following inequality holds for $R^s_{I,c}$:

$$R^{s+1}_{I,c} \leq (1+a_2)\max\{\max_{c\neq c^*_m}|\kappa^s_{c,m,j}|, \frac{1}{2}d^{-\frac{1}{2}}\} + (1+a_2)\sum_{s'=0}^{s}\frac{\eta^{s'}}{CM^2}k^*\max_{c'}|\tilde{\alpha}_{m,j,k^*,c'}\beta_{c',k^*}|(R^{s'}_{I,c})^{k^*-1}.$$

Therefore, by induction, we establish that $|\kappa^s_{c,m,j}|$ is upper bounded by $R^s_{I,c}$ for all $j \notin \mathcal{J}^*_m$ and $s = 0, 1, \ldots, t+1$ with high probability.

$$|\kappa^{s+1}_{g,m,j}| \leq \max\Big\{A_1\eta_e, \Big|\kappa^s_{g,m,j} + \frac{\eta^s}{CM^2}\sum_{c'\in[C]}\sum_{i=k^*}^{p^*}\Big(i\tilde{\alpha}_{m,j,i,c'}\beta_{c',i}(\kappa^s_{c',m,j})^{i-1}(w_g^{*\top}w_{c'}^* - \kappa^s_{g,m,j}\kappa^s_{c',m,j})$$
$$+ i\tilde{\alpha}_{m,j,i,c'}s_{c'}\gamma_i(\kappa^s_{g,m,j})^{i-1}(1 - (\kappa^s_{g,m,j})^2)\Big) + \frac{|\kappa^s_{g,m,j}|(\eta^s)^2A_1{}^2d}{2} + \frac{(\eta^s)^3A_1{}^3d^{\frac{3}{2}}}{2}$$
$$+ w_g^{*\top}(I_d - w^s_{m,j}w^s_{m,j}{}^\top)\Xi^s w_{m,j}\Big|\Big\}$$
$$\leq \max\Big\{A_1\eta_e, \Big|\kappa^s_{g,m,j} + \frac{\eta^s}{CM^2}\Big(|\sum_{c'\in[C]}\sum_{i=k^*}^{p^*}i\tilde{\alpha}_{m,j,i,c'}\beta_{c',i}(\kappa^s_{c',m,j})^{i-1}(w_g^{*\top}w_{c'}^* - \kappa^s_{g,m,j}\kappa^s_{c',m,j})|$$
$$+ |\sum_{c'\in[C]}\sum_{i=k^*}^{p^*}is_{c'}\alpha_{m,j,i,c'}\gamma_i(\kappa^s_{g,m,j})^{i-1}(1 - (\kappa^s_{g,m,j})^2)|\Big)$$
$$+ \frac{|\kappa^t_{g,m,j}|(\eta^t)^2A_1{}^2d}{2} + \frac{(\eta^s)^3A_1{}^3d^{\frac{3}{2}}}{2} + \eta^s w_c^{*\top}(I_d - w^s_{m,j}w^s_{m,j}{}^\top)\Xi^s w_{m,j}\Big|\Big\}$$
$$\leq \max\Big\{A_1\eta_e, \Big|\kappa^s_{g,m,j} + \frac{\eta^s}{CM^2}\Big(Cp^{*2}\max_{c',i}|\tilde{\alpha}_{m,j,i,c'}\beta_{c',i}|\max_{c'}|\kappa^s_{c',m,j}|^{k^*-1}(\max_{c'}|w_g^{*\top}w_{c'}^*| + |\kappa^s_{g,m,j}|)$$
$$+ k^*|\sum_{c'\in[C]}s_{c'}\tilde{\alpha}_{m,j,k^*,c'}\gamma_{k^*}||\kappa^s_{g,m,j}|^{k^*-1} + Cp^{*2}\max_{c',i}|s_{c'}\tilde{\alpha}_{m,j,i,c'}\gamma_i||\kappa^s_{g,m,j}|^{k^*}\Big)$$
$$+ \frac{|\kappa^s_{g,m,j}|(\eta^s)^2A_1{}^2d}{2} + \frac{(\eta^s)^3A_1{}^3d^{\frac{3}{2}}}{2} + \eta^s w_g^{*\top}(I_d - w^s_{m,j}w^s_{m,j}{}^\top)\Xi^s w_{m,j}\Big|\Big\}$$
$$\leq \max\Big\{A_1\eta_e, \max_{c\neq c^*_m}|\kappa^0_{c,m,j}| + (1+\frac{2}{3}a_2)\sum_{s'=0}^{s}\frac{\eta^{s'}}{CM^2}k^*|\sum_{c'\in[C]}s_{c'}\tilde{\alpha}_{m,j,k^*,c'}\gamma_{k^*}|\max\{|\kappa^{s'}_{g,m,j}|^{k^*-1}, (\frac{1}{2}d^{-\frac{1}{2}})^{k^*-1}\}$$
$$+ \max_{c\neq c^*_m}\Big|\sum_{s'=0}^{s}\eta^{s'}w_g^{*\top}(I_d - w^{s'}_{m,j}w^{s'}_{m,j}{}^\top)\Xi^{s'}w_{m,j}\Big|\Big\}.$$

Since, $w_c^{*\top} w_{c'}^* = \tilde{O}(d^{-\frac{1}{2}})$ for all $c \neq c'$, $w_g^{*\top} w_c^* = \tilde{O}(d^{-\frac{1}{2}})$ for all $c$, $\kappa_{c,m,j}^s = \tilde{O}(d^{-\frac{1}{2}})$ for all $(c,j) \neq (c_m^*, j_m^*)$ and $\kappa_{g,m,j}^s = \tilde{O}(d^{-\frac{1}{2}})$ for all $j \in [J]$, the term $k^* \max_i |\sum_{c' \in [C]} s_{c'} \tilde{\alpha}_{m,j,k^*,c'} \gamma_{k^*}| |\kappa_{g,m,j}^s|^{k^*-1}$ subsumes the remaining terms within the expression $\frac{\eta^s}{CM^2}(\cdot)$ with wight $\frac{1}{3} a_2$ in the fourth inequality. $\frac{|\kappa_{g,m,j}^s|(\eta^s)^2 A_1^2 d}{2} + \frac{(\eta^s)^3 A_1^3 d^{\frac{3}{2}}}{2}$ are also subsumed with weight $\frac{1}{3} a_2$ by $\eta^s \leq a_4 d^{-\frac{k^*}{2}}$ and $|\kappa_{c,m,j}^s| \leq A_2 A_3 d^{-\frac{1}{2}}$ in the fourth inequality.

By upper bounding the noise term in the same way, we have

$$\left| \sum_{s'=0}^s \eta^{s'} w_g^{*\top} (I_d - w_{m,j}^{s'} w_{m,j}^{s'}{}^\top) \Xi^{s'} w_{m,j} \right| \leq \begin{cases} \frac{1}{2} a_2 d^{-\frac{1}{2}}, & \text{if } s \leq A_2 d^{k^*-1}, \\ \frac{a_2 \eta_e s}{3CM^2} k^* |\sum_{c' \in [C]} s_{c'} \tilde{\alpha}_{m,j,k^*,c'} \gamma_{k^*}| (\frac{1}{2} d^{-\frac{1}{2}})^{k^*-1}, & \text{if } s > A_2 d^{k^*-1}, \end{cases}$$

with high probability.

Thus,

$$|\kappa_{g,m,j}^s| \leq (1+a_2) \max \left\{ |\kappa_{g,m,j}^0|, \frac{1}{2} d^{-\frac{1}{2}} \right\}$$
$$+ (1+a_2) \sum_{s'=0}^s \frac{\eta^{s'}}{CM^2} k^* |\sum_{c' \in [C]} s_{c'} \tilde{\alpha}_{m,j,k^*,c'} \gamma_{k^*}| \max \left\{ |\kappa_{g,m,j}^{s'}|, R_{\mathrm{I},g}^{s'} \right\}^{k^*-1}$$

In contrast, the following inequality holds for $R_{\mathrm{I},c}^s$:

$$R_{\mathrm{I},g}^{s+1} \leq (1+a_2) \max\{ \max_{c \neq c_m^*} |\kappa_{g,m,j}^s|, \frac{1}{2} d^{-\frac{1}{2}} \} + (1+a_2) \sum_{s'=0}^s \frac{\eta^{s'}}{CM^2} k^* |\sum_{c' \in [C]} s_{c'} \tilde{\alpha}_{m,j,k^*,c'} \gamma_{k^*}| (R_{\mathrm{I},g}^{s'})^{k^*-1}.$$

Therefore, by induction, we establish that $|\kappa_{g,m,j}^s|$ is upper bounded by $R_{\mathrm{I},g}^s$ for all $j \notin \mathcal{J}_m^*$ and $s = 0, 1, \ldots, t+1$ with high probability. Finally, we consolidate the auxiliary sequences $Q_{\mathrm{I},c}^s$, $Q_{\mathrm{I},g}^s$, $R_{\mathrm{I},c}^s$, and $R_{\mathrm{I},g}^s$ into a unified auxiliary sequence $Q_{\mathrm{I}}^s$, which serves as an upper bound for $|\kappa_{c,m,j}^{s+1}|$ and $|\kappa_{g,m,j}^{s+1}|$ for all $(c,j) \neq (c_m^*, j_m^*)$, where $m \in \mathcal{M}_c$.

Clearly due to $A_3 \sum_{s'=0}^s \frac{\eta^{s'}}{CM^2} k^* \tilde{\alpha}_{m,j,k^*,c_m^*} \beta_{c_m^*,k^*} (\kappa_{c_m^*,m,j_m^*}^{s'})^{k^*-1} d^{-\frac{1}{2}} > 0$, the upper bound provided by $R_{\mathrm{I},c}^s$ is subsumed by the upper bound provided by $Q_{\mathrm{I},c}^s$ and the upper bound provided by $R_{\mathrm{I},g}^s$ is subsumed by the upper bound provided by $Q_{\mathrm{I},g}^s$. Consequently, for all $(c,j) \neq (c_m^*, j_m^*)$, $\kappa_{c,m,j}^s$ and $\kappa_{g,m,j}^s$ is upper bounded by $Q_{\mathrm{I}}^s$ for all $s = 0, 1, \ldots, t+1$ with high probability.

$(Q_{\mathrm{I},g}^s)_{s=0}^{t+1}$ is expressed as

$$Q_{\mathrm{I}}^{s+1} = Q_{\mathrm{I}}^s + (1+a_2) \frac{\eta^s}{CM^2} k^* \max\{ \max_{c'} |\tilde{\alpha}_{m,j,k^*,c'} \beta_{c',k^*}|, |\sum_{c' \in [C]} s_{c'} \tilde{\alpha}_{m,j,k^*,c'} \gamma_{k^*}| \} (Q_{\mathrm{I}}^s)^{k^*-1}$$
$$+ A_3 \frac{\eta^s}{CM^2} k^* \tilde{\alpha}_{m,j,k^*,c_m^*} \beta_{c_m^*,k^*} (\kappa_{c_m^*,m,j_m^*}^{s+1})^{k^*-1} d^{-\frac{1}{2}}$$

with $Q_{\mathrm{I},g}^0 = (1+a_2) \max\{ \max_c |\kappa_{c,m,j}^s|, |\kappa_{g,m,j}^s|, \frac{1}{2} d^{-\frac{1}{2}} \}$. $\qquad\square$

Based on Lemma C.12, we prove that $|\kappa_{c,m,j}^s|$ for $c \neq c_m^*$ or $j \notin \mathcal{J}_m^*$, and $|\kappa_{g,m,j}^s|$ remains upper bounded throughout the trajectory by induction.

**Lemma C.13.** *Consider the expert $m \in \mathcal{M}_c$. Let $w_c^{*\top} w_{c'}^* \leq A_4 d^{-\frac{1}{2}} = \tilde{O}(d^{-1/2})$ for all $c \neq c'$, $w_c^{*\top} w_g^* \leq A_4 d^{-\frac{1}{2}} = \tilde{O}(d^{-1/2})$ for all $c$, and $\eta^s = \eta_e \leq a_4 d^{-\frac{k^*}{2}}$. For all $s = 0, 1, \ldots, t$, suppose that*

- $\kappa_{c_m^*,m,j_m^*}^s \leq a_2$,

- $\max\{ |\kappa_{c,m,j}^s|, |\kappa_{g,m,j}^s| \} \leq \kappa_{c_m^*,m,j_m^*}^s$ *for all $(c,j) \neq (c_m^*, j_m^*)$,*

- $\max\{ |\kappa_{c,m,j}^s|, |\kappa_{g,m,j}^s| \} \leq 4 A_3 d^{-\frac{1}{2}}$ *for all $(c,j) \neq (c_m^*, j_m^*)$.*

*Then, if we have $\kappa_{c_m^*, m, j_m^*}^{t+1} \leq a_2$,*

- $\max\{|\kappa_{c,m,j}^{t+1}|, |\kappa_{g,m,j}^{t+1}|\} \leq \kappa_{c_m^*, m, j_m^*}^s, \textit{for all } (c,j) \neq (c_m^*, j_m^*),$

- $\max\{|\kappa_{c,m,j}^{t+1}|, |\kappa_{g,m,j}^{t+1}|\} \leq 4A_3 d^{-\frac{1}{2}}, \textit{for all } (c,j) \neq (c_m^*, j_m^*),$

*hold with high probability.*

*Proof.* To begin, consider the the case when

$$(1 + a_2)\frac{\eta^s}{CM^2} k^* \max\{\max_{c'} |\tilde{\alpha}_{m,j,k^*,c'}\beta_{c',k^*}|, |\sum_{c' \in [C]} s_{c'}\tilde{\alpha}_{m,j,k^*,c'}\gamma_{k^*}|\}(Q_\mathrm{I}^s)^{k^*-1}$$

$$> A_3 \frac{\eta^s}{CM^2} k^* \tilde{\alpha}_{m,j,k^*,c_m^*}\beta_{c_m^*,k^*}(P_\mathrm{I}^s)^{k^*-1}d^{-\frac{1}{2}}$$

holds for all $s = 0, 1, \ldots, t$.

Due to Lemma C.12,

$$Q_\mathrm{I}^{s+1} \leq Q_\mathrm{I}^s + (1 + 2a_2)\frac{\eta^s}{CM^2} k^* \max\{\max_{c'} |\tilde{\alpha}_{m,j,k^*,c'}\beta_{c',k^*}|, |\sum_{c' \in [C]} s_{c'}\tilde{\alpha}_{m,j,k^*,c'}\gamma_{k^*}|\}(Q_\mathrm{I}^s)^{k^*-1}$$

holds for all $s = 0, 1, \ldots, t$.

By applying Lemma A.4 to $Q_\mathrm{I}^s$,

$$Q_\mathrm{I}^s \leq \frac{Q_\mathrm{I}^0}{\left(1 - \eta_e k^*(k^* - 2)(1 + 3a_2)\max\{\max_{c'} |\tilde{\alpha}_{m,j,k^*,c'}\beta_{c',k^*}|, |\sum_{c' \in [C]} s_{c'}\tilde{\alpha}_{m,j,k^*,c'}\gamma_{k^*}|\}C^{-1}M^{-2}(Q_\mathrm{I}^0)^{k^*-2}s\right)^{\frac{1}{k^*-2}}}$$

holds for all $s = 0, 1, \ldots, t$,

where we used $\left(1 + (1 + 2a_2)\frac{\eta^s}{CM^2} k^* \max\{\max_{c'} |\tilde{\alpha}_{m,j,k^*,c'}\beta_{c',k^*}|, |\sum_{c' \in [C]} s_{c'}\tilde{\alpha}_{m,j,k^*,c'}\gamma_{k^*}|\}\right)^{k^*-1} - 1 \leq \frac{a_2}{1+2a_2}$.

By applying Lemma A.4 to $P_\mathrm{I}^s$,

$$P_\mathrm{I}^s \geq \frac{P_\mathrm{I}^0}{\left(1 - \eta_e k^*(k^* - 2)(1 - a_2), \tilde{\alpha}_{m,j,k^*,c_m^*}\beta_{c_m^*,k^*}C^{-1}M^{-2}(P_\mathrm{I}^0)^{k^*-2}s\right)^{-\frac{1}{k^*-2}}}$$

holds for all $s = 0, 1, \ldots, t$.

From Corollary C.3, Lemma C.8, and Lemma C.9, it hold that $Q_\mathrm{I}^0 < P_\mathrm{I}^0$ and $(1 + 3a_2)\max\{\max_{c'} |\tilde{\alpha}_{m,j,k^*,c'}\beta_{c',k^*}|, |\sum_{c' \in [C]} s_{c'}\tilde{\alpha}_{m,j,k^*,c'}\gamma_{k^*}|\}$
$(\max\{\max_c |\kappa_{c,m,j}^s|, |\kappa_{g,m,j}^s|, \frac{1}{2}d^{-\frac{1}{2}}\})^{k^*-2} \leq (1 - a_2)\tilde{\alpha}_{m,j,k^*,c_m^*}\beta_{c_m^*,k^*}(\kappa_{c_m^*,m,j_m^*}^s)^{k^*-2}$, which establish that $Q_\mathrm{I}^s < P_\mathrm{I}^s$.

Thus, $Q_\mathrm{I}^{t+1} \leq P_\mathrm{I}^{t+1}$, which indicates that $\max\{\max_c |\kappa_{c,m,j}^{t+1}|, |\kappa_{g,m,j}^{t+1}|\} \leq \kappa_{c_m^*,m,j_m^*}^{t+1}$ for all $(c,j) \neq (c_m^*, j_m^*)$.

Since $P_\mathrm{I}^s \leq \kappa_{c_m^*, m, j_m^*}^s \leq a_2$,

$$t \leq \frac{\eta_e^{-1}CM^2((P_\mathrm{I}^0)^{-k^*+2} - (a_2)^{-k^*+2})}{k^*(k^* - 2)(1 - a_2)\tilde{\alpha}_{m,j,k^*,c_m^*}\beta_{c_m^*,k^*}}$$

$$\leq \frac{\eta_e^{-1}CM^2(1 + 5a_2)\frac{\max\{\max_{c'} |\tilde{\alpha}_{m,j,k^*,c'}\beta_{c',k^*}|, |\sum_{c' \in [C]} s_{c'}\tilde{\alpha}_{m,j,k^*,c'}\gamma_{k^*}|\}}{\tilde{\alpha}_{m,j,k^*,c_m^*}\beta_{c_m^*,k^*}}(P_\mathrm{I}^0)^{-k^*+2}}{k^*(k^* - 2)(1 + 3a_2)\max\{\max_{c'} |\tilde{\alpha}_{m,j,k^*,c'}\beta_{c',k^*}|, |\sum_{c' \in [C]} s_{c'}\tilde{\alpha}_{m,j,k^*,c'}\gamma_{k^*}|\}}.$$

In contrast, $Q_\mathrm{I}^{t+1} > A_3 d^{-\frac{1}{2}}$ holds only if

$$t \geq \frac{\eta_e^{-1}CM^2\left((Q_\mathrm{I}^0)^{-k^*+2} - (A_3 d^{-\frac{1}{2}})^{-k^*+2}\right)}{k^*(k^* - 2)(1 + 3a_2)\max\{\max_{c'} |\tilde{\alpha}_{m,j,k^*,c'}\beta_{c',k^*}|, |\sum_{c' \in [C]} s_{c'}\tilde{\alpha}_{m,j,k^*,c'}\gamma_{k^*}|\}} - 1$$

$$\geq \frac{\eta_e^{-1}CM^2(1-2a_2)(Q_{\mathrm{I}}^0)^{-k^*+2}}{k^*(k^*-2)(1+3a_2)\max\{\max_{c'}|\tilde{\alpha}_{m,j,k^*,c'}\beta_{c',k^*}|,|\sum_{c'\in[C]}s_{c'}\tilde{\alpha}_{m,j,k^*,c'}\gamma_{k^*}|\}}$$

where we used $(A_3d^{-\frac{1}{2}})^{-k^*+2} \leq a_2(\kappa_{c_m^*,m,j_m^*}^s)^{-k^*+2} \leq a_2(P_{\mathrm{I}}^0)^{-k^*+2} \leq a_2(Q_{\mathrm{I}}^0)^{-k^*+2}$.

From Corollary C.3, Lemma C.8, and Lemma C.9, we have

$$(1+8a_2)\max\{\max_{c'}|\tilde{\alpha}_{m,j,k^*,c'}\beta_{c',k^*}|,|\sum_{c'\in[C]}s_{c'}\tilde{\alpha}_{m,j,k^*,c'}\gamma_{k^*}|\}\big(\max\{\max_c|\kappa_{c,m,j}^s|,|\kappa_{g,m,j}^s|,\frac{1}{2}d^{-\frac{1}{2}}\}\big)^{k^*-2}$$

$$< \tilde{\alpha}_{m,j,k^*,c_m^*}\beta_{c_m^*,k^*}(\kappa_{c_m^*,m,j_m^*}^0)^{k^*-2}.$$

However, this leads to the contradiction that

$$\eta_e^{-1}CM^2(1-2a_2)(Q_{\mathrm{I}}^0)^{-k^*+2} > \eta_e^{-1}CM^2(1+5a_2)\frac{\max\{\max_{c'}|\tilde{\alpha}_{m,j,k^*,c'}\beta_{c',k^*}|,|\sum_{c'\in[C]}s_{c'}\tilde{\alpha}_{m,j,k^*,c'}\gamma_{k^*}|\}}{\tilde{\alpha}_{m,j,k^*,c_m^*}\beta_{c_m^*,k^*}}(P_{\mathrm{I}}^0)^{-k^*+2}.$$

Thus, $Q_{\mathrm{I}}^{t+1} > A_3d^{-\frac{1}{2}}$ does not hold, which implies that $Q_{\mathrm{I}}^{t+1} \leq A_3d^{-\frac{1}{2}}$.

Next, consider the case where

$$(1+a_2)\frac{\eta^s}{CM^2}k^*\max\{\max_{c'}|\tilde{\alpha}_{m,j,k^*,c'}\beta_{c',k^*}|,|\sum_{c'\in[C]}s_{c'}\tilde{\alpha}_{m,j,k^*,c'}\gamma_{k^*}|\}(Q_{\mathrm{I}}^s)^{k^*-1}$$

$$> A_3\frac{\eta^s}{CM^2}k^*\tilde{\alpha}_{m,j,k^*,c_m^*}\beta_{c_m^*,k^*}(P_{\mathrm{I}}^s)^{k^*-1}d^{-\frac{1}{2}}$$

holds for all $s = 0, 1, \ldots, \tau_1 - 1$, but

$$(1+a_2)\frac{\eta^s}{CM^2}k^*\max\{\max_{c'}|\tilde{\alpha}_{m,j,k^*,c'}\beta_{c',k^*}|,|\sum_{c'\in[C]}s_{c'}\tilde{\alpha}_{m,j,k^*,c'}\gamma_{k^*}|\}(Q_{\mathrm{I}}^s)^{k^*-1}$$

$$\leq A_3\frac{\eta^s}{CM^2}k^*\tilde{\alpha}_{m,j,k^*,c_m^*}\beta_{c_m^*,k^*}(P_{\mathrm{I}}^s)^{k^*-1}d^{-\frac{1}{2}}$$

holds for $s = \tau_1 \leq t$.

Here, suppose that

$$Q_{\mathrm{I}}^{s+1} \leq Q_{\mathrm{I}}^s + 2a_2\frac{\eta_e}{CM^2}k^*\tilde{\alpha}_{m,j,k^*,c_m^*}\beta_{c_m^*,k^*}(P_{\mathrm{I}}^s)^{k^*-1}$$

holds for all $s = \tau_1, \tau + 1, \ldots, \tau_2 \leq t$.

Then,

$$Q_{\mathrm{I}}^{\tau_2+1} = Q_{\mathrm{I}}^{\tau_1} + \sum_{s=\tau_1}^{\tau_2}2A_3\frac{\eta_e}{CM^2}k^*\tilde{\alpha}_{m,j,k^*,c_m^*}\beta_{c_m^*,k^*}(P_{\mathrm{I}}^s)^{k^*-1}d^{-\frac{1}{2}}$$

$$\leq Q_{\mathrm{I}}^{\tau_1} + \frac{2A_3}{(1-a_2)d^{\frac{1}{2}}}(P_{\mathrm{I}}^{\tau_2+1} - P_{\mathrm{I}}^{\tau_1})$$

$$\leq \Big(\frac{\tilde{\alpha}_{m,j,k^*,c_m^*}\beta_{c_m^*,k^*}}{\max\{\max_{c'}|\tilde{\alpha}_{m,j,k^*,c'}\beta_{c',k^*}|,|\sum_{c'\in[C]}s_{c'}\tilde{\alpha}_{m,j,k^*,c'}\gamma_{k^*}|\}}\cdot\frac{A_3}{(1+a_2)d^{\frac{1}{2}}}\Big)^{\frac{1}{k^*-1}}P_{\mathrm{I}}^{\tau_1}$$

$$+ \frac{2A_3}{(1-a_2)d^{\frac{1}{2}}}(P_{\mathrm{I}}^{\tau_2+1} - P_{\mathrm{I}}^{\tau_1})$$

$$\leq \Big(\frac{\tilde{\alpha}_{m,j,k^*,c_m^*}\beta_{c_m^*,k^*}}{\max\{\max_{c'}|\tilde{\alpha}_{m,j,k^*,c'}\beta_{c',k^*}|,|\sum_{c'\in[C]}s_{c'}\tilde{\alpha}_{m,j,k^*,c'}\gamma_{k^*}|\}}\cdot\frac{A_3}{(1+a_2)d^{\frac{1}{2}}}\Big)^{\frac{1}{k^*-1}}P_{\mathrm{I}}^{\tau_2+1}.$$

Thus,

$$(1 + a_2)\frac{\eta_e}{CM^2}k^* \max\{\max_{c'} |\tilde{\alpha}_{m,j,k^*,c'}\beta_{c',k^*}|, |\sum_{c' \in [C]} s_{c'}\tilde{\alpha}_{m,j,k^*,c'}\gamma_{k^*}|\}(Q_{\mathrm{I}}^{\tau_2+1})^{k^*-1}$$

$$\leq A_3\frac{\eta_e}{CM^2}k^*\tilde{\alpha}_{m,j,k^*,c_m^*}\beta_{c_m^*,k^*}(P_{\mathrm{I}}^{\tau_2+1})^{k^*-1}d^{-\frac{1}{2}}.$$

We have $Q_{\mathrm{I}}^{t+1} \leq P_{\mathrm{I}}^{t+1}$ since

$$\left(\frac{(1 + a_2)\max\{\max_{c'} |\tilde{\alpha}_{m,j,k^*,c'}\beta_{c',k^*}|, |\sum_{c' \in [C]} s_{c'}\tilde{\alpha}_{m,j,k^*,c'}\gamma_{k^*}|\}}{A_3\tilde{\alpha}_{m,j,k^*,c_m^*}\beta_{c_m^*,k^*}}d^{\frac{1}{2}}\right)^{\frac{1}{k^*-1}} \geq 1.$$

Therefore, we obtain $\max\{\max_c |\kappa_{c,m,j}^{t+1}|, |\kappa_{g,m,j}^{t+1}|\} \leq \kappa_{c_m^*,m,j_m}^{t+1}$ for all $(c,j) \neq (c_m^*, j_m^*)$.

In the same way,

$$Q_{\mathrm{I}}^{t+1} = Q_{\mathrm{I}}^{\tau_1} + \sum_{s=\tau_1}^{t} 2A_3\frac{\eta_e}{CM^2}k^*\tilde{\alpha}_{m,j,k^*,c_m^*}\beta_{c_m^*,k^*}(P_{\mathrm{I}}^s)^{k^*-1}d^{-\frac{1}{2}}$$

$$\leq Q_{\mathrm{I}}^{\tau_1} + \frac{2A_3}{(1-a_2)d^{\frac{1}{2}}}(P_{\mathrm{I}}^{t+1} - P_{\mathrm{I}}^{\tau_1})$$

$$\leq Q_{\mathrm{I}}^{\tau_1} + \frac{2A_3}{(1-a_2)d^{\frac{1}{2}}}P_{\mathrm{I}}^{t+1}$$

$$\leq Q_{\mathrm{I}}^{\tau_1} + 3A_3 d^{-\frac{1}{2}}$$

$$\leq 4A_3 d^{-\frac{1}{2}}.$$

where the last inequality is by $Q_{\mathrm{I}}^{\tau_1} \leq A_3 d^{-\frac{1}{2}}$ from the first case. $\qquad\square$

Finally, we establish Lemma C.6.

*Proof of Lemma C.6.* Suppose that $T_1 = \lfloor(\eta_e k^*(k^*-2)(1-5a_2)(\tilde{\alpha}_{m,j,k^*,c_m^*}\beta_{c_m^*,k^*})C^{-1}M^{-2}(P_{\mathrm{I}}^0)^{k^*-2})^{-1}\rfloor$ and $\kappa_{c_m^*,m,j_m^*}^s \leq a_2$, $\max\{|\kappa_{c,m,j}^s|, |\kappa_{g,m,j}^s|\} \leq \kappa_{c_m^*,m,j_m^*}^s$, and $\max\{|\kappa_{c,m,j}^s|, |\kappa_{g,m,j}^s|\} \leq A_3 d^{-\frac{1}{2}}$, where $(c,j) \neq (c_m^*, j_m^*)$ for all $s = 0, 1, ..., T_1$. Then the bounds given by Lemma C.12 and Lemma C.13 hold for all $s = 0, 1, ..., T_1$ with high probability.

Thus, by Lemma A.4 and Lemma C.12,

$$\kappa_{c_m^*,m,j_m^*}^t \geq P_{\mathrm{I}}^t \geq \frac{P_{\mathrm{I}}^0}{(1 - \eta_e k^*(k^*-2)(1-a_2)\tilde{\alpha}_{m,j,k^*,c_m^*}\beta_{c_m^*,k^*}C^{-1}M^{-2}(P_{\mathrm{I}}^0)s)^{k^*-2}}$$

However, obviously when $t = T_1$,

$$\kappa_{c_m^*,m,j_m^*}^t \geq P_{\mathrm{I}}^t \geq \frac{P_{\mathrm{I}}^0}{(\eta_e k^*(k^*-2)(1-a_2)\tilde{\alpha}_{m,j,k^*,c_m^*}\beta_{c_m^*,k^*}C^{-1}M^{-2}(P_{\mathrm{I}}^0)s)^{\frac{1}{k^*-2}}}$$

$$\geq \frac{1}{(\eta_e k^*(k^*-2)(1-a_2)\tilde{\alpha}_{m,j,k^*,c_m^*}\beta_{c_m^*,k^*}C^{-1}M^{-2})^{\frac{1}{k^*-2}}} > 1.$$

This leads to a contradiction as $\kappa_{c_m^*,m,j_m^*}^{T_1} \leq 1$. Since $\max\{|\kappa_{c,m,j}^s|, |\kappa_{g,m,j}^s|\} \leq 4A_3 d^{-\frac{1}{2}}$ from Lemma C.13,

$$|\kappa_{c_m^*,m,j_m^*}^{t_1} - \kappa_{c_m^*,m,j_m^*}^{t_1-1}| \leq A_1\eta_e \leq A_1 a_4 d^{-\frac{k^*}{2}} \leq A_3 d^{-\frac{1}{2}}.$$

Thus,

$$|\kappa_{c,m,j}^{t_1}| \leq |\kappa_{c,m,j}^{t_1-1}| + |\kappa_{c_m^*,m,j_m^*}^{t_1} - \kappa_{c_m^*,m,j_m^*}^{t_1-1}| \leq 5A_3 d^{-\frac{1}{2}} = \tilde{O}(d^{-\frac{1}{2}}).$$

$\qquad\square$

### C.3. Router Learning Stage

This subsection is dedicated to proving that, after the exploration stage, the router successfully learns to dispatch the data $x_c$ to the appropriate experts $m \in \mathcal{M}_c$ with high probability. At this stage, we employ single-step training for the gating network, as in Damian et al. (2022); Ba et al. (2022); Dandi et al. (2024a).

**Lemma C.14.** *Take* $\eta^{T_1} = \eta_r \leq a_5$, $n = \tilde{\Theta}(d)$, *and* $T_2 = 1$. *Then, any* $m \notin \mathcal{M}_c$ *satisfies* $h_m(x_c; \Theta^{T_1+T_2}) - \max_{m' \in [M]} h_{m'}(x_c; \Theta^{T_1+T_2}) < 0$ *with high probability.*

To prove Lemma C.14, we show that, for experts not belonging to the set of professional experts, the alignment between the cluster signal $v_c$ and the weights of the gating network $\theta_m$, represented as $\iota^t_{c,m} := v_c^\top \theta^t_m$, is upper bounded.

**Lemma C.15.** *Take* $\eta^{T_1} = \eta_r \leq a_5$, $n = \tilde{\Theta}(d)$, *and* $T_2 = 1$. *Then, for all* $m \notin \mathcal{M}_c$, *we have* $\iota^{T_1+T_2}_{c,m} \leq -a_2^{k^*} a_5 A_6 \rho \Omega\left(\frac{1}{CJM^3}\right) \leq \max_{m' \in [M]} \iota^{T_1+T_2}_{c,m'} - a_2^{k^*} a_5 A_6 \rho \Omega\left(\frac{1}{CJM^3}\right)$.

Here, we describe an important property of the softmax router: by initializing the weights of the gating network $\theta_m$ to zero, we ensure that their sum over all experts is also zero.

**Lemma C.16.** $\sum_{m \in [M]} \theta^{T_1+T_2}_m = \sum_{m \in [M]} \theta^{T_1}_m = 0$ *where* $T_2 = 1$.

*Proof.* We define the gradient with respect to $\theta_m$ for an input pair $(x^i_c, y^i_c)$ as

$$\nabla_{\theta_m} \mathcal{L}_{c,i} := \nabla_{\theta_m}[\mathbf{1}\left(m(x^i_c) = m\right) \pi_m(x^i_c) y^i_c f_m(x^i_c)]$$
$$= (\mathbf{1}\left(m(x^i_c) = m\right)) - \pi_m(x^i_c)) \pi_{m(x^i_c)}(x^i_c) y^i_c f_{m(x^i_c)}(x^i_c) x^i_c.$$

The sum of gradients over the entire batch is

$$\frac{1}{n} \sum_{i=1}^n \sum_{m \in [M]} \nabla_{\theta_m} \mathcal{L}_{c,i} = \frac{1}{n} \sum_{i=1}^n \sum_{m \in [M]} (\mathbf{1}\left(m(x^i_c) = m\right) - \pi_m(x^i_c)) \pi_{m(x^i_c)}(x^i_c) y^i_c f_{m(x^i_c)}(x^i_c) x^i_c = 0.$$

This result, combined with the initialization $\theta^{T_1}_m = 0$ and the fact that $\sum_{m \in [M]} \pi_m(x^i_c) = 1$, completes the proof. $\qquad \square$

Based on the above properties of the gating network, we establish Lemma C.15.

*Proof of Lemma C.15.* The population gradient for the gating network of the router can be expressed as

$$-\nabla_{\theta_m} \mathbb{E}[\mathcal{L}_{c,i}] = \nabla_{\theta_m} \mathbb{E}_c \mathbb{E}_{x_c}[\mathbf{1}\left(m(x_c) = m\right) \pi_m(x_c) y_c f_m(x_c)]$$
$$= \mathbb{E}_c \mathbb{E}_{x_c}[(\mathbf{1}\left(m(x_c) = m\right) - \pi_m(x_c)) \pi_{m(x_c)}(x_c) y_c f_{m(x_c)}(x_c) x_c].$$

We decompose the key components of the population gradient.

$\mathbb{E}_c \mathbb{E}_{x_c}[y_c f_{m(x_c)} x_c]$

$$= \frac{1}{CJ} \sum_{c \in [C]} \sum_{j \in [J]} \mathbb{E}_{x_c}\left[\left(\sum_{i=k^*}^{p^*} \frac{\beta_{c,i}}{\sqrt{i!}} \mathrm{He}_i(w_c^{*\top} x_c) + s_c \sum_{i=k^*}^{p^*} \frac{\gamma_i}{\sqrt{i!}} \mathrm{He}_i(w_g^{*\top} x_c)\right)\left(\sum_{i=0}^\infty \frac{\alpha_{m,j,i}}{\sqrt{i!}} \mathrm{He}_i(w_{m,j}^\top x_c)\right) x_c\right]$$

$$= \frac{1}{CJ} \sum_{c \in [C]} \sum_{j \in [J]} \left\{\mathbb{E}_z\left[\left(\sum_{i=k^*}^{p^*} \frac{\beta_{c,i}}{\sqrt{i!}} \mathrm{He}_i(w_c^{*\top} z) + s_c \sum_{i=k^*}^{p^*} \frac{\gamma_i}{\sqrt{i!}} \mathrm{He}_i(w_g^{*\top} z)\right)\left(\sum_{i=0}^\infty \frac{\alpha_{m,j,i}}{\sqrt{i!}} \mathrm{He}_i(w_{m,j}^\top(z + \rho v_c))\right) z\right] \right.$$

$$\left. + \mathbb{E}_z\left[\left(\sum_{i=k^*}^{p^*} \frac{\beta_{c,i}}{\sqrt{i!}} \mathrm{He}_i(w_c^{*\top} z) + s_c \sum_{i=k^*}^{p^*} \frac{\gamma_i}{\sqrt{i!}} \mathrm{He}_i(w_g^{*\top} z)\right)\left(\sum_{i=0}^\infty \frac{\alpha_{m,j,i}}{\sqrt{i!}} \mathrm{He}_i(w_{m,j}^\top(z + \rho v_c))\right) \rho v_c\right]\right\}$$

$$= \frac{1}{CJ} \sum_{c \in [C]} \sum_{j=1}^J \left\{\underbrace{\mathbb{E}_z\left[\left(\sum_{i=k^*}^{p^*} \frac{i\beta_{c,i}}{\sqrt{i!}} \mathrm{He}_{i-1}(w_c^{*\top} z)\right)\left(\sum_{i=0}^\infty \frac{\alpha_{m,j,i}}{\sqrt{i!}}\left(\sum_{l=0}^i \binom{i}{l} \mathrm{He}_{i-l}(w_{m,j}^\top z)(\rho w_{m,j}^\top v_c)^l\right)\right) w_c^*\right]}_{(I)}\right.$$

$$+ \mathbb{E}_z \Big[ s_c \sum_{i=k^*}^{p^*} \frac{i\gamma_i}{\sqrt{i!}} \mathrm{He}_{i-1}(w_g^{*\top}z) \Big( \sum_{l=0}^{i} \binom{i}{l} \mathrm{He}_{i-l}(w_{m,j}^\top z)(\rho w_{m,j}^\top v_c)^l \Big) w_g^* \Big]$$

$$\underbrace{\phantom{+ \mathbb{E}_z \Big[ s_c \sum_{i=k^*}^{p^*} \frac{i\gamma_i}{\sqrt{i!}} \mathrm{He}_{i-1}(w_g^{*\top}z) \Big( \sum_{l=0}^{i} \binom{i}{l} \mathrm{He}_{i-l}(w_{m,j}^\top z)(\rho w_{m,j}^\top v_c)^l \Big) w_g^* \Big]}}_{\text{(II)}}$$

$$+ \mathbb{E}_z \Big[ \Big( \sum_{i=k^*}^{p^*} \frac{\beta_{c,i}}{\sqrt{i!}} \mathrm{He}_i(w_c^{*\top}z) + s_c \sum_{i=k^*}^{p^*} \frac{\gamma_i}{\sqrt{i!}} \mathrm{He}_i(w_g^{*\top}z) \Big) \Big( \sum_{i=0}^{\infty} \frac{\alpha_{m,j,i}}{\sqrt{i!}} \Big( \sum_{l=0}^{i-1} \binom{i}{l}(i-l)\mathrm{He}_{i-l-1}(w_{m,j}^\top z)(\rho w_{m,j}^\top v_c)^l \Big) \Big) w_{m,j} \Big]$$

$$\underbrace{\phantom{+ \mathbb{E}_z \Big[ \Big( \sum_{i=k^*}^{p^*} \frac{\beta_{c,i}}{\sqrt{i!}} \mathrm{He}_i(w_c^{*\top}z) + s_c \sum_{i=k^*}^{p^*} \frac{\gamma_i}{\sqrt{i!}} \mathrm{He}_i(w_g^{*\top}z) \Big) \Big( \sum_{i=0}^{\infty} \frac{\alpha_{m,j,i}}{\sqrt{i!}} \Big( \sum_{l=0}^{i-1} \binom{i}{l}(i-l)\mathrm{He}_{i-l-1}(w_{m,j}^\top z)(\rho w_{m,j}^\top v_c)^l \Big) \Big) w_{m,j} \Big]}}_{\text{(III)}}$$

$$+ \mathbb{E}_z \Big[ \Big( \sum_{i=k^*}^{p^*} \frac{\beta_{c,i}}{\sqrt{i!}} \mathrm{He}_i(w_c^{*\top}z) + s_c \sum_{i=k^*}^{p^*} \frac{\gamma_i}{\sqrt{i!}} \mathrm{He}_i(w_g^{*\top}z) \Big) \Big( \sum_{i=0}^{\infty} \frac{\alpha_{m,j,i}}{\sqrt{i!}} \Big( \sum_{l=0}^{i} \binom{i}{l} \mathrm{He}_{i-l}(w_{m,j}^\top z)(\rho w_{m,j}^\top v_c)^l \Big) \Big) \rho v_c \Big] \Big\}$$

$$\underbrace{\phantom{+ \mathbb{E}_z \Big[ \Big( \sum_{i=k^*}^{p^*} \frac{\beta_{c,i}}{\sqrt{i!}} \mathrm{He}_i(w_c^{*\top}z) + s_c \sum_{i=k^*}^{p^*} \frac{\gamma_i}{\sqrt{i!}} \mathrm{He}_i(w_g^{*\top}z) \Big) \Big( \sum_{i=0}^{\infty} \frac{\alpha_{m,j,i}}{\sqrt{i!}} \Big( \sum_{l=0}^{i} \binom{i}{l} \mathrm{He}_{i-l}(w_{m,j}^\top z)(\rho w_{m,j}^\top v_c)^l \Big) \Big) \rho v_c \Big]}}_{\text{(IV)}}$$

where the second equality follows from $x_c = z + \rho v_c$, where $z \sim \mathcal{N}(0, I_d)$, and $v_c^\top w_{c'}^*$ and $v_c^\top w_g^*$ for all $(c, c')$. Third equality follows from the binomial expansion, Stein's Lemma, and integration by parts.

From now, we look at the alignment $v_c^\top \theta_m$. Thus, (I) and (II) are negligible since $v_c^\top w_{c'}^* = 0$ and $v_c^\top w_g^* = 0$ for all $(c, c')$. We expand (III) and (IV).

$$(\text{III}) = \sum_{i=k^*}^{p^*} \sum_{l=i+1}^{\infty} \Big( \frac{\alpha_{m,j,l}}{\sqrt{l!}} \binom{l}{l-i-1} (\rho w_{m,j}^\top v_c)^{l-i-1} \Big) \frac{\beta_{c,i}}{\sqrt{i!}}(i+1)!(w_{m,j}^\top w_c^*)^i w_{m,j}$$

$$+ \sum_{i=k^*}^{p^*} \sum_{l=i+1}^{\infty} \Big( \frac{\alpha_{m,j,l}}{\sqrt{l!}} \binom{l}{l-i-1} (\rho w_{m,j}^\top v_c)^{l-i-1} \Big) \frac{s_c\gamma_i}{\sqrt{i!}}(i+1)!(w_{m,j}^\top w_g^*)^i w_{m,j}$$

$$= \sum_{i=k^*}^{p^*} \sqrt{i+1}\tilde{\alpha}_{m,j,i+1,c}\beta_{c,i}(\kappa_{c,m,j})^i w_{m,j} + \sum_{i=k^*}^{p^*} \sqrt{i+1}s_c\tilde{\alpha}_{m,j,i+1,c}\gamma_i(\kappa_{g,m,j})^i w_{m,j},$$

$$(\text{IV}) = \sum_{i=k^*}^{p^*} \sum_{l=i}^{\infty} \Big( \frac{\alpha_{m,j,l}}{\sqrt{l!}} \binom{l}{l-i} (\rho w_{m,j}^\top v_c)^{l-i} \Big) \frac{\beta_{c,i}}{\sqrt{i!}} i!(w_{m,j}^\top w_c^*)^i \rho v_c$$

$$+ \sum_{i=k^*}^{p^*} \sum_{l=i}^{\infty} \Big( \frac{\alpha_{m,j,l}}{\sqrt{l!}} \binom{l}{l-i} (\rho w_{m,j}^\top v_c)^{l-i} \Big) \frac{s_c\gamma_i}{\sqrt{i!}} i!(w_{m,j}^\top w_g^*)^i \rho v_c$$

$$= \sum_{i=k^*}^{p^*} \tilde{\alpha}_{m,j,i,c}\beta_{c,i}(\kappa_{c,m,j}^t)^i \rho v_c + \sum_{j=1}^{J} \sum_{i=k^*}^{p^*} s_c\tilde{\alpha}_{m,j,i,c}\gamma_i(\kappa_{g,m,j}^t)^i \rho v_c,$$

where we used the orthogonality property of Hermite polynomials in both (III) and (IV).

We introduce the discrepancy $\Xi_{\theta_m}$ between the population and the empirical gradient.

$$\Xi_{\theta_m} = -\nabla_{\theta_m} \frac{1}{n} \sum_{t=1}^{n} \mathcal{L}_{c_t, i_t} + \nabla_{\theta_m} \frac{1}{C} \sum_{c=1}^{C} \mathbf{E}_{i|c}[\mathcal{L}_{c,i}].$$

Next, we evaluate the empirical update of the alignment. Note that $\iota_{c,m}^t = v_c^\top \theta_m^t$ and $\pi_m(x_c) = \frac{\exp(\theta_m^\top x_c)}{\sum_{m'} \exp(\theta_{m'}^\top x_c)}$.

$$\iota_{c,m}^{T_1+T_2} = \iota_{c,m}^{T_1} - \eta^{T_1} v_c^\top \nabla_{\theta_m} \frac{1}{n} \sum_{t=1}^{n} \mathcal{L}_{c_t, i_t}$$

$$= \iota_{c,m}^{T_1} + \eta^{T_1} v_c^\top \nabla_{\theta_m} \mathbb{E}_c \mathbb{E}_{x_c}[\mathbf{1}(m(x_c) = m)\pi_m(x_c)y_c f_m(x_c)] + \eta^{T_1} v_c^\top \Xi_{\theta_m}$$

$$= \iota_{c,m}^{T_1} + \frac{\eta^{T_1} v_c^\top}{C} \sum_{c' \in [C]} \mathbb{E}_{x_{c'}}[\mathbf{1}(m(x_{c'}) = m)\pi_{m(x_{c'})}(x_{c'})y_{c'} f_{m(x_{c'})}(x_{c'})x_{c'}]$$

$$\underbrace{\phantom{= \iota_{c,m}^{T_1} + \frac{\eta^{T_1} v_c^\top}{C} \sum_{c' \in [C]} \mathbb{E}_{x_{c'}}[\mathbf{1}(m(x_{c'}) = m)\pi_{m(x_{c'})}(x_{c'})y_{c'} f_{m(x_{c'})}(x_{c'})x_{c'}]}}_{\text{(A)}}$$

$$- \frac{\eta^{T_1} v_c^\top}{C} \sum_{c' \in [C]} \underbrace{\mathbb{E}_{x_{c'}} [\mathbf{1}(m(x_{c'}) \in \mathcal{M}_c) \pi_{m(x_{c'})}(x_{c'}) \pi_m(x_c) y_{c'} f_{m(x_{c'})}(x_{x'}) x_{c'}]}_{(B)}$$

$$- \frac{\eta^{T_1} v_c^\top}{C} \sum_{c' \in [C]} \underbrace{\mathbb{E}_{x_{c'}} [\mathbf{1}(m(x_{c'}) \notin \mathcal{M}_c) \pi_{m(x_{c'})}(x_{c'}) \pi_m(x_c) y_{c'} f_{m(x_{c'})}(x_{c'}) x_{c'}]}_{(C)} + \eta^{T_1} v_c^\top \Xi_{\theta_m}.$$

We derive an upper bound for $|(A)|$ and $|(C)|$, and a lower bound for (B) for $m \notin \mathcal{M}_c$. We first derive a upper bound for $|(A)|$ and $|(C)|$.

$$|(A)| \leq \frac{\eta^{T_1}}{CJM} \sum_{c' \in [C]} \sum_{j=1}^{J} \sum_{m \notin \mathcal{M}_c} \sum_{i=k^*}^{p^*} \left| \sqrt{i+1} \tilde{\alpha}_{m,j,i,c'} \beta_{c',i} (\kappa_{c',m,j}^{T_1})^i (w_{m,j}^{T_1\top} v_c) \right.$$
$$\left. + \sqrt{i+1} \tilde{\alpha}_{m,j,i,c'} s_{c'} \gamma_i (\kappa_{g,m,j}^{T_1})^i (w_{m,j}^{T_1\top} v_c) \right|$$
$$+ \frac{\eta^{T_1}}{CJM} \sum_{j=1}^{J} \sum_{m \notin \mathcal{M}_c} \sum_{i=k^*}^{p^*} \left| \tilde{\alpha}_{m,j,i,c} \beta_{c,i} (\kappa_{c,m,j}^{T_1})^i \rho + \tilde{\alpha}_{m,j,i,c} s_c \gamma_i (\kappa_{g,m,j}^{T_1})^i \rho \right|$$
$$\leq \eta^{T_1} \left( p^* \sqrt{p^*+1} \max_{m \notin \mathcal{M}_c, c', j, i} |\tilde{\alpha}_{m,j,i,c'} \beta_{c',i}| \max_{m \notin \mathcal{M}_c, c', j} |\kappa_{c',m,j}^{T_1}|^{k^*} \max_{m \notin \mathcal{M}_c, j} |w_{m,j}^{T_1\top} v_c| \right.$$
$$\left. + p^* \sqrt{p^*+1} \max_{m \notin \mathcal{M}_c, c', j, i} |\tilde{\alpha}_{m,j,i,c'} s_{c'} \gamma_i| \max_{m \notin \mathcal{M}_c, j} |\kappa_{g,m,j}^{T_1}|^{k^*} \max_{m \notin \mathcal{M}_c, j} |w_{m,j}^{T_1\top} v_c| \right)$$
$$+ \frac{\eta_r^{T_1}}{C} \left( p^* \max_{m \notin \mathcal{M}_c, j, i, c} |\tilde{\alpha}_{m,j,i,c} \beta_{c,i}| \max_{m \notin \mathcal{M}_c, c, j} |\kappa_{c',m,j}^{T_1}|^{k^*} \rho + p^* \max_{m \notin \mathcal{M}_c, j, i, c} |\tilde{\alpha}_{m,j,i,c} s_c \gamma_i| \max_{m \notin \mathcal{M}_c, j} |\kappa_{g,m,j}^{T_1}|^{k^*} \rho \right)$$
$$= \tilde{O}(\eta_r d^{-\frac{1}{2}}),$$

$$|(C)| \leq \frac{\eta^{T_1}}{CJM} \sum_{c' \in [C]} \sum_{j=1}^{J} \sum_{m \notin \mathcal{M}_c} \sum_{i=k^*}^{p^*} \left| \sqrt{i+1} \tilde{\alpha}_{m,j,i,c'} \beta_{c',i} (\kappa_{c',m,j}^{T_1})^i (w_{m,j}^{T_1\top} v_c) \right.$$
$$\left. + \sqrt{i+1} \tilde{\alpha}_{m,j,i,c'} s_{c'} \gamma_i (\kappa_{g,m,j}^{T_1})^i (w_{m,j}^{T_1\top} v_c) \right|$$
$$+ \frac{\eta^{T_1}}{CJM} \sum_{j=1}^{J} \sum_{m \notin \mathcal{M}_c} \sum_{i=k^*}^{p^*} \left| \tilde{\alpha}_{m,j,i,c} \beta_{c,i} (\kappa_{c,m,j}^{T_1})^i \rho + \tilde{\alpha}_{m,j,i,c} s_c \gamma_i (\kappa_{g,m,j}^{T_1})^i \rho \right|$$
$$\leq \eta^{T_1} \left( p^* \sqrt{p^*+1} \max_{m \notin \mathcal{M}_c, c', j, i} |\tilde{\alpha}_{m,j,i,c'} \beta_{c',i}| \max_{m \notin \mathcal{M}_c, c', j} |\kappa_{c',m,j}^{T_1}|^{k^*} \max_{m \notin \mathcal{M}_c, j} |w_{m,j}^{T_1\top} v_c| \right.$$
$$\left. + p^* \sqrt{p^*+1} \max_{m \notin \mathcal{M}_c, c', j, i} |\tilde{\alpha}_{m,j,i,c'} s_{c'} \gamma_i| \max_{m \notin \mathcal{M}_c, j} |\kappa_{g,m,j}^{T_1}|^{k^*} \max_{m \notin \mathcal{M}_c, j} |w_{m,j}^{T_1\top} v_c| \right)$$
$$+ \frac{\eta^{T_1}}{C} \left( p^* \max_{m \notin \mathcal{M}_c, j, i, c} |\tilde{\alpha}_{m,j,i,c} \beta_{c,i}| \max_{m \notin \mathcal{M}_c, c, j} |\kappa_{c',m,j}^{T_1}|^{k^*} \rho + p^* \max_{m \notin \mathcal{M}_c, j, i, c} |\tilde{\alpha}_{m,j,i,c} s_c \gamma_i| \max_{m \notin \mathcal{M}_c, j} |\kappa_{g,m,j}^{T_1}|^{k^*} \rho \right)$$
$$= \tilde{O}(\eta_r d^{-\frac{1}{2}}),$$

where we used $\mathbb{P}(m(x_c) = m) = \frac{1}{M}$ and $\pi_m(x_{c'}), \pi_{m(x_{c'})}(x_{c'}) = \frac{1}{M}$ for all $m \in [M]$. Furthermore, by Lemma C.6, all terms in the RHS are upper bounded by $\tilde{O}(\eta_r d^{-\frac{1}{2}})$ since $\kappa_{c,m,j}^{T_1} = \tilde{O}(d^{-\frac{1}{2}})$ for all $(c,j) \neq (c_m^*, j_m^*)$ and $\kappa_{g,m,j}^{T_1} = \tilde{O}(d^{-\frac{1}{2}})$ for all $j \in [J]$. By Lemma C.10, $|\tilde{\alpha}_{m,j,i,c}|$ are by at most $\tilde{O}(1)$.

Next we derive a lower bound for (B).

Therefore,

$$(B) = \frac{\eta^{T_1}}{CJM^3} \sum_{c' \in [C]} \sum_{j=1}^{J} \sum_{m' \in \mathcal{M}_c} \sum_{i=k^*}^{p^*} \left[ \sqrt{i+1} \tilde{\alpha}_{m',j,i,c'} \beta_{c',i} (\kappa_{c',m',j}^{T_1})^i (w_{m',j}^{t\top} v_c) \right.$$

$$+ \sqrt{i+1} \tilde{\alpha}_{m',j,i,c'} s_{c'} \gamma_i (\kappa_{g,m',j}^{T_1})^i (w_{m',j}^{T_1 \top} v_c) \Big]$$

$$+ \frac{\eta^{T_1}}{CJM^3} \sum_{j=1}^{J} \sum_{m' \in \mathcal{M}_c} \sum_{i=k^*}^{p^*} \Big[ \tilde{\alpha}_{m',j,i,c} \beta_{c,i} (\kappa_{c,m',j}^{T_1})^i \rho + \tilde{\alpha}_{m',j,i,c} s_c \gamma_i (\kappa_{g,m',j}^{T_1})^i \rho \Big]$$

$$\geq \frac{\eta^{T_1}}{CJM^3} \Big( \sum_{m' \in \mathcal{M}_c} (\tilde{\alpha}_{m',j_m^*,k^*,c} \beta_{c,k^*} (\kappa_{c,m',j}^{T_1})^{k^*} \rho)$$

$$- J|\mathcal{M}_c|p^* \max_{m',j\neq j_m^*} |\tilde{\alpha}_{m',j,i,c}\beta_{c,i}| \max_{j\neq j_m^*,m'\in\mathcal{M}_c} (|\kappa_{c,m',j}^{T_1}|)^{k^*} \rho$$

$$- J|\mathcal{M}_c|p^* \max_{m',j,c'} |\tilde{\alpha}_{m',j,i,c'} s_{c'} \gamma_i| \max_{j,m'\in\mathcal{M}_c} (|\kappa_{g,m',j}^{T_1}|)^{k^*} \rho$$

$$- J|\mathcal{M}_c|p^* \sqrt{p^*+1} \max_{c',m',j,i} |\tilde{\alpha}_{m',j,i,c'}\beta_{c',i}| \max_{c',m',j} (|\kappa_{c',m',j}^{T_1}|)^{k^*} \max_{m',j} |w_{m',j}^{T_1 \top} v_c|$$

$$- J|\mathcal{M}_c|p^* \sqrt{p^*+1} \max_{c',m',j,i} |\tilde{\alpha}_{m',j,i,c'}s_c'\gamma_i| \max_{m'\in\mathcal{M}_c,j} (|\kappa_{g',m',j}^{T_1}|)^{k^*} \max_{m',j} |w_{m',j}^{T_1 \top} v_c| \Big)$$

$$= a_2{}^{k^*} A_6 \Omega \Big( \frac{\eta_r \rho}{CJM^3} \Big) > 0,$$

where we used that $\kappa_{c,m,j}^t \geq a_2$ for all $m \in \mathcal{M}_c$ and $j \in \mathcal{J}_m^*$, $\tilde{\alpha}_{m',j_m^*,k^*,c}\beta_{c,k^*} \geq A_6$, $\kappa_{c,m,j}^t = \tilde{O}(d^{-1/2})$ for all $(c,j) \neq (c_m^*, j_m^*)$, and $\kappa_{g,m,j}^t = \tilde{O}(d^{-1/2})$ for all $j \in [J]$, as shown in Lemma C.6. Among the terms on the RHS of $\frac{\eta^t}{CJM^3}(\cdot)$, all terms except for the first one are smaller than the first term by at least an order of $\tilde{O}(d^{-\frac{1}{2}})$.

Thus, in the case where $m \notin \mathcal{M}_c$, we have

$$\iota_{c,m}^{T_1+T_2} \leq \iota_{c,m}^{T_1} - a_2{}^{k^*} A_6 \Omega(\frac{\eta_r \rho}{CJM^3}) + \eta_r v_c^\top \Xi_{\theta_m}$$

$$\leq \iota_{c,m}^{T_1} - a_2{}^{k^*} A_6 \Theta \Big( \frac{\eta_r \rho}{CJM^3} \Big) + \eta_r |v_c^\top \Xi_{\theta_m}|$$

$$\leq -a_2{}^{k^*} A_6 \Theta \Big( \frac{\eta_r \rho}{CJM^3} \Big) + \eta_r |v_c^\top \Xi_{\theta_m}|.$$

which holds with high probability.

Since $|v_c^\top \nabla_{\theta_m} \mathcal{L}_{c,i}| = |v_c^\top [(\mathbf{1}\,(m(x_c^i) = m)) - \pi_m(x_c^i))\pi_{m(x_c^i)}(x_c^i)y_c^i f_{m(x_c^i)}(x_c^i)x_c^i]| = \tilde{O}(1)$, by applying Hoeffding's inequality with $n = \tilde{O}(d)$ sample and $C = O(1)$, we have

$$|v_c^\top \Xi_{\theta_m}| = \Big| -v_c^\top \nabla_{\theta_m} \frac{1}{n} \sum_{t=1}^{n} \mathcal{L}_{c_t,i_t} + v_c^\top \nabla_{\theta_m} \frac{1}{C} \sum_{c=1}^{C} \mathbf{E}_{i|c}[\mathcal{L}_{c,i}] \Big|$$

$$= \tilde{O}(d^{-\frac{1}{2}})$$

with high probability.

Therefore, by noting from Lemma C.16 that

$$\max_{m' \in [M]} \iota_{c,m'}^{T_2} \geq \frac{1}{M} \sum_{m' \in [M]} \iota_{c,m'}^{T_2} = 0,$$

we finally obtain that

$$\iota_{c,m}^{T_1+T_2} \leq -a_2{}^{k^*} A_6 \Omega \Big( \frac{\eta_r \rho}{CJM^3} \Big) \leq \max_{m' \in [M]} \iota_{c,m'}^{T_1+T_2} - a_2{}^{k^*} A_6 \Omega \Big( \frac{\eta_r \rho}{CJM^3} \Big)$$

$$= \max_{m' \in [M]} \iota_{c,m'}^{T_1+T_2} - a_2{}^{k^*} a_5 A_6 \Omega \Big( \frac{\rho}{CJM^3} \Big).$$

$\square$

We establish Lemma C.14.

*Proof of Lemma C.14.* In Lemma C.15, we demonstrated that the cluster signals of clusters not assigned to the router's gating network are aligned with high probability to be negative. Here, we aim to show that, upon observing new data $x_c = \rho v_c + z$, where $z \sim \mathcal{N}(0, I_d)$, the data is not dispatched to the experts not assigned to the corresponding cluster with high probability. With high probability, we have

$$\|\theta_m^{T_1+T_2}\| \leq \|\theta_m^{T_1+T_2} - \theta_m^{T_1}\| = \eta_r \left\|\nabla_{\theta_m} \frac{1}{n} \sum_{t=1}^{n} \mathcal{L}_{c_t, i_t}\right\|$$

$$= \eta_r \left\|\left(\nabla_{\theta_m} \frac{1}{C} \sum_{c=1}^{C} \mathbf{E}_{i|c}[\mathcal{L}_{c,i}] + \Xi_{\theta_m}\right)\right\|$$

$$\leq \eta_r \left\|\nabla_{\theta_m} \frac{1}{C} \sum_{c=1}^{C} \mathbf{E}_{i|c}[\mathcal{L}_{c,i}]\right\| + \eta_r \left\|\Xi_{\theta_m}\right\|$$

$$\leq a_5 A_5 O(\rho).$$

where we used the previously expanded (I), (II), (III), and (IV) along with $\|w_c^*\| = \|w_g^*\| = \|w_{m,j}\| = \|v_c\| = 1$. Since $x_c = z + \rho v_c$ and ${\theta_m^{T_1+T_2}}^\top z \sim \mathcal{N}(0, \|\theta_m^{T_1+T_2}\|^2)$, it follows that, with high probability,

$$|{\theta_m^{T_1+T_2}}^\top z| \leq a_5 A_5 O(\rho)\sqrt{\log d}.$$

For $m \notin \mathcal{M}_c$, we have, with high probability,

$$h_m(x_c; \Theta^{T_1+T_2}) = {\theta_m^{T_1+T_2}}^\top x_c = \rho \iota_{c,m}^{T_1+T_2} + {\theta_m^{T_1+T_2}}^\top z \leq a_5 \rho \underbrace{\left(-a_2^{k^*} A_6 \rho \Omega\left(\frac{1}{CJM^3}\right) + A_5 O(\sqrt{\log d})\right)}_{<0} < 0$$

where we used $\rho \gtrsim \frac{A_5 O(\sqrt{\log d})}{a_2^{k^*} A_6 \Omega(\frac{1}{CJM^3})}$.

By combining this result with $\sum_{m \in [M]} \theta_m^{T_1+T_2} = 0$, we have

$$\max_{m'} h_{m'}(x_c; \Theta^{T_1+T_2}) \geq \frac{1}{|\mathcal{M}_c|} \sum_{m' \in \mathcal{M}_c} h_{m'}(x_c; \Theta^{T_1+T_2})$$

$$= -\frac{1}{|\mathcal{M}_c|} \sum_{m' \notin \mathcal{M}_c} \underbrace{h_{m'}(x_c; \Theta^{T_1+T_2})}_{\leq -a_2^{k^*} a_5 A_6 \Omega(\frac{\rho^2}{CJM^3})}$$

$$\geq a_2^{k^*} a_5 A_6 \Omega\left(\frac{M - |\mathcal{M}_c|}{|\mathcal{M}_c|} \frac{\rho^2}{CJM^3}\right)$$

$$> 0$$

$$> h_m(x_c; \Theta^{T_1+T_2})$$

for $m \notin \mathcal{M}_c$ as desired. $\square$

## C.4. Expert Learning Stage

In this subsection, we discuss the individual learning of experts in the context of receiving data from their assigned clusters with high probability. As in Appendix C.2, we introduce $A_i$ and $a_i$ with the following order of strength, but they do not necessarily have to be the same.

$$A_1 \lesssim a_2^{-1} \lesssim A_2 \lesssim A_4 \lesssim a_4^{-1} \lesssim A_3 = \tilde{O}(1).$$

The proof in this subsection follows the same structure as that in Appendix C.2 and is based on the proof by Oko et al. (2024a). We define $t$ as $t - T_1 - T_2 + 1$.

**Re-initialization.** Before entering the expert learning stage, the expert weights $w_{mj}$ are reinitialized. Although this initialization is not strictly necessary, it is performed to ensure a decent path for the alignment so that the product of $g^*$ and the activation Hermite coefficients is positive. If the Hermite coefficients of $f^*$ and $g^*$ are identical, aligning with $w_g^*$ may become challenging, as $w_c^*$ is already aligned.

The re-initialization satisfies the following condition:

**Lemma C.17** (Following Lemma2 of Oko et al. (2024a)). *When $J \gtrsim \frac{1}{C} \log \frac{M}{\delta}$ for each $m$ in $\mathcal{M}_c$, we have at least $J_{\min}$ neurons $w_{mj}$ such that*

$$w_{m,j}^0{}^\top w_c^* \geq \frac{1}{\sqrt{d}}, \quad |\beta_{c,k^*}|(w_{m,j}^0{}^\top w_c^*)^{k^*-2} \geq |s_c \gamma_{k^*}||w_{m,j}^0{}^\top w_g^*|^{k^*-2} + a_c(w_{m,j}^0{}^\top w_c^*)^{k^*-2}$$

*with probability at least $0.999$ with sufficiently large $d$, where $a_c$ is a small constant, where $a_c \lesssim (\log d)^{k^*-2}$.*

*Likewise, when $J \gtrsim J_{\min} \operatorname{polylog}(d)$ for each $m$ in $\mathcal{M}_c$, we have at least $J_{\min}$ neurons $w_{mj}$ such that*

$$w_{m,j}^0{}^\top w_g^* \geq \frac{1}{\sqrt{d}}, \quad |s_c \gamma_{k^*}|(w_{m,j}^0{}^\top w_g^*)^{k^*-2} \geq |\beta_{c,k^*}||w_{m,j}^0{}^\top w_c^*|^{k^*-2} + a_g(w_{m,j}^0{}^\top w_g^*)^{k^*-2}$$

*with probability at least $0.999$ with sufficiently large $d$, where $a_g$ is a small constant, where $a_g \lesssim (\log d)^{k^*-2}$.*

Based on re-initialization, we define the set of neurons that comparatively align with the indexed features $w_c^*$ and $w_g^*$.

**Definition C.18.** We define the set $\mathcal{J}_c$ as the set of indices $j$ that satisfy the given conditions:

$$\mathcal{J}_c := \left\{ j \in [J] \mid w_{m,j}^0{}^\top w_c^* \geq \frac{1}{\sqrt{d}}, \quad |\beta_{c,k^*}|(w_{m,j}^0{}^\top w_c^*)^{k^*-2} \geq |s_c \gamma_{k^*}||w_{m,j}^0{}^\top w_g^*|^{k^*-2} + a_c(w_{m,j}^0{}^\top w_c^*)^{k^*-2} \right\}.$$

Similarly, we define the set $\mathcal{J}_g$ as the set of indices $j$ that satisfy the corresponding conditions:

$$\mathcal{J}_g := \left\{ j \in [J] \mid w_{m,j}^0{}^\top w_g^* \geq \frac{1}{\sqrt{d}}, \quad |s_c \gamma_{k^*}|(w_{m,j}^0{}^\top w_g^*)^{k^*-2} \geq |\beta_{c,k^*}||w_{m,j}^0{}^\top w_c^*|^{k^*-2} + a_g(w_{m,j}^0{}^\top w_g^*)^{k^*-2} \right\}.$$

*Remark* C.19. In the rest of this section, we will discuss Phase III and IV on the event that the re-initialization was successful.

**Adaptive top-$k$ routing.** Importantly, in this subsection, we conduct an analysis similar to that in Appendix C.2, but employ a different routing strategy. As demonstrated in the previous subsection, the router does not route data to experts where $m \notin \mathcal{M}_c$; however, it cannot definitively determine which expert among those where $m \in \mathcal{M}_c$ should receive the data. Therefore, resolving conflicts within $m \in \mathcal{M}_c$ without knowing $M_c$ or $v_c$ requires an alternative approach. One solution is to choose $k$ experts for each $x_c$ by the following strategy:

*Expert $m$ is in top-$k$ if and only if $h_m(x) \geq 0$.*

The complete MoE model, incorporating the adaptive top-$k$ routing, can be expressed as $\hat{F}_M(x_c; \{\hat{a}_m\}_{m=1}^M) := \sum_{m=1}^M \mathbb{1}[h_m(x_c) \geq 0] f_m(x_c)$. This routing strategy mitigates load imbalance, which would otherwise disrupt data routing among experts $m \in \mathcal{M}_c$ under top-1 routing.

**Lemma C.20.** *There is at least one $m \in \mathcal{M}_c$ such that $f_m$ is correctly routed with high probability over the randomness of $x_c$. In other words, for some fixed $m \in \mathcal{M}_c$, on the randomness of $x_c$,*

$$\mathbb{P}[x_c \text{ is routed to the set of experts including } m] \geq 1 - d^{-A}.$$

*In addition, $f_{m'}$ is never chosen for all $m' \notin \mathcal{M}_c$ when given $x_c$ with high probability.*

*Proof.* By Lemma C.14, with high probability, we have

$$\max_{m'} h_{m'}(x_c; \Theta^{T_2}) \geq a_2{}^{k^*} a_5 A_6 \Omega \left( \frac{M - |\mathcal{M}_c|}{|\mathcal{M}_c|} \frac{\rho^2}{CJM^3} \right) > 0$$

and

$$h_m(x_c; \Theta^{T_2}) < -a_2{}^{k^*} a_5 A_6 \Omega\left(\frac{\rho^2}{CJM^3}\right) < 0$$

for $m \notin \mathcal{M}_c$.

The proof is complete, since we employ adaptive top-$k$ routing, where expert $m$ is activated if and only if $h_m(x_c) \geq 0$.

$\square$

By Lemma C.20, the overall MoE model is, with high probability, equivalent to $\hat{F}_{\mathcal{M}_c}(x_c; \{\hat{a}_m\}_{m \in \mathcal{M}_c}) := \sum_{m \in \mathcal{M}_c} \mathbb{1}\left[h_m(x_c) \geq 0\right] f_m(x_c)$.

Following Oko et al. (2024a), we sequentially demonstrate the following:

- For $t_{3,1} \leq T_{3,1}$, $J_c$ neurons achieve an alignment of $\tilde{\Omega}(1)$ with $w_c^*$, and $J_g$ neurons achieve an alignment of $\tilde{\Omega}(1)$ with $w_g^*$, i.e., weak recovery.

- For $t_{3,2} \leq T_{3,2}$, $J_c$ neurons achieve an alignment of $1 - \tilde{O}(1)$ with $w_c^*$, and $J_g$ neurons achieve an alignment of $1 - \tilde{O}(1)$ with $w_g^*$.

- In a total time of $(T_{3,1} - t_{3,1}) + (T_{3,2} - t_{3,2}) + T_{3,3}$, $J_c$ neurons achieve an alignment of $1 - \epsilon$ with $w_c^*$, and $J_g$ neurons achieve an alignment of $1 - \epsilon$ with $w_g^*$, i.e., strong recovery.

**Weak recovery.** In the same manner as the exploration stage, we begin by evaluating the stochastic updates for the alignments, $\kappa_{c,m,j}^t$ and $\kappa_{g,m,j}^t$. Subsequently, we derive the lower bound of $\kappa_{c,m,j}^t$ for $j \in \mathcal{J}_c$ and $\kappa_{g,m,j}^t$ for $j \in \mathcal{J}_g$, as well as the upper bound of $|\kappa_{g,m,j}^t|$ for $j \in \mathcal{J}_c$ and $|\kappa_{c,m,j}^t|$ for $j \in \mathcal{J}_g$. Furthermore, we demonstrate that there exists a point in time when $\kappa_{c,m,j}^t$ for $j \in \mathcal{J}_c$ grows to a constant level, while $\kappa_{g,m,j}^t$ for $j \in \mathcal{J}_c$ remains at the saddle point.

**Lemma C.21.** *Consider the expert $m \in \mathcal{M}_c$ and $j \in \mathcal{J}_c$. Let $w_c^{*\top} w_g^* \leq A_4 d^{-\frac{1}{2}} = \tilde{O}(d^{-1/2})$ and $\eta^t = \eta_e \leq a_4 d^{-\frac{k^*}{2}}$. Then, with high probability, there exists some time $t_{3,1} \leq T_{3,1} = \tilde{\Theta}(\eta_e^{-1} d^{\frac{k^*-2}{2}})$ such that the following conditions hold:*

- $\kappa_{c,m,j}^{t_{3,1}} \geq a_2$, *and*

- $|\kappa_{g,m,j}^{t_{3,1}}| \leq 5 A_3 d^{-\frac{1}{2}} = \tilde{O}(d^{-1/2})$.

*The same argument applies symmetrically when exchanging $c$ and $g$.*

Similar to Lemma C.8 and Lemma C.9 in Appendix C.2, we provide a bound on the Hermite coefficients influenced by the mean vector.

**Lemma C.22.** *Under Adaptive top-$k$ routing and re-initialization in Lemma C.17, suppose that $|v_c^\top w_{m,j}^s| = \tilde{O}(d^{-\frac{1}{2}})$, $|\kappa_{c,m,j}^s| = \tilde{O}(d^{-\frac{1}{2}})$ and $|\kappa_{g,m,j}^s| = \tilde{O}(d^{-\frac{1}{2}})$ for all $s = 0, 1, \ldots, t \leq \tau = \tilde{O}(d^{k^*-1})$. Then, by setting $\eta^t = \eta_e \leq a_4 d^{-\frac{k^*}{2}}$, we obtain that $|\tilde{\alpha}_{m,j,i,c}^{t+1} - \alpha_{m,j,i}| = \tilde{O}(d^{-\frac{1}{2}})$ with high probability.*

*Proof.* Since the gradient in Lemma C.8 changes only by an order of $\mathrm{polylog}\, d$, the proof follows in the same manner. $\square$

**Lemma C.23.** *Consider a neuron which satisfies $\alpha_{m,j,k^*} \beta_{c,k^*} > 0$ and $\alpha_{m,j,i} \beta_{c,i} > 0$ for $k^* < i \leq p^*$. Under Adaptive top-$k$ routing and re-initialization in Lemma C.17, suppose that $|v_c^\top w_{m,j}^s| = \tilde{\Omega}(d^{-\frac{1}{2}})$, $\kappa_{c,m,j}^s = \tilde{\Omega}(d^{-\frac{1}{2}})$, and $|\kappa_{g,m,j}^s| = \tilde{O}(d^{-\frac{1}{2}})$ for all $s = 0, 1, \ldots, t \leq \tau = \tilde{O}(d^{k^*-1})$. Then, by setting $\eta^t = \eta_e \leq a_4 d^{-\frac{k^*}{2}}$, we obtain that $|\tilde{\alpha}_{m,j,i,c}^{t+1}| - |\alpha_{m,j,i}| = \tilde{\Omega}(d^{-\frac{1}{2}})$ with high probability.*

*Proof.* Since the gradient in Lemma C.9 changes only by an order of $\mathrm{polylog}\, d$, the proof follows in the same manner. $\square$

*Remark* C.24. For the sake of conciseness in the exposition of the proof, we omit the superscript $t$ in $\tilde{\alpha}_{m,j,i,c}^t$. Based on Lemma C.22, Lemma C.23, and Lemma C.10, the bounds in the subsequent lemmas are properly justified, regardless of the variations in the coefficients $\tilde{\alpha}_{c,m,j,i}^t$.

*Proof of Lemma C.21* . We begin by evaluating the stochastic updates of the experts $m \in \mathcal{M}_c$. Since the router learns to dispatch the data $x_c$ to the experts $m \in \mathcal{M}_c$ with high probability by Lemma C.20, we have $\mathbb{P}(m(x_c) \in \mathcal{M}_c) = 1 - d^{-A}$, for some $A > 0$. For clarity, we will henceforth assume and write $\mathbb{P}(m(x_c) = m) = 1$ throughout the remainder of this proof. Thus, we consider the probability conditioned on the event that the data is routed with high probability.

By analyzing the gradient update, as in Lemma C.7, we obtain that

$$
\begin{aligned}
\kappa_{c,m,j}^{t+1} &\geq \kappa_{c,m,j}^t + \eta^t w_c^{*\top}(I_d - w_{m,j}^t w_{m,j}^{t\top}) \nabla_{w_{m,j}} \mathbb{E}[\mathbf{1}(m(x_c) = m) y_c a_{m,j} \sigma_m(w_{m,j}^{t\top} x_c + b_{m,j})] \\
&\quad - \frac{|\kappa_{c,m,j}^t|(\eta^t)^2}{2} \|\nabla_{w_{m,j}} \mathbf{1}(m(x_c) = m) y_c a_{m,j} \sigma_m(w_{m,j}^{t\top} x_c + b_{m,j})\|^2 \\
&\quad - \frac{(\eta^t)^3}{2} \|\nabla_{w_{m,j}} \mathbf{1}(m(x_c) = m) y_c a_{m,j} \sigma_m(w_{m,j}^{t\top} x_c + b_{m,j})\|^3 \\
&\quad + \eta^t w_c^{*\top}(I_d - w_{m,j}^t w_{m,j}^{t\top}) \Xi_{w_{m,j}}^t \\
&\geq \kappa_{c,m,j}^t + \eta^t w_c^{*\top} \sum_{i=k^*}^{p^*} \Big[ \sum_{l=i}^{\infty} \Big( \frac{l\alpha_{m,j,l}}{\sqrt{l!}} \binom{l-1}{l-i} (\rho w_{m,j}^{t\top} v_c)^{l-i} \Big) \frac{i\beta_{c,i}}{\sqrt{i!}} (i-1)! (w_c^{*\top} w_{m,j}^t)^{i-1} \Big] \\
&\quad - \frac{|\kappa_{c,m,j}^t|(\eta^t)^2 A_1^2 d}{2} - \frac{(\eta^t)^3 A_1^3 d^{\frac{3}{2}}}{2} + \eta^t w_c^{*\top}(I_d - w_{m,j}^t w_{m,j}^{t\top}) \Xi_{w_{m,j}}^t \\
&\geq \kappa_{c,m,j}^t + \eta^t \sum_{i=k^*}^{p^*} \Big[ i\tilde{\alpha}_{m,j,i,c} \beta_{c,i} (\kappa_{c,m,j}^t)^{i-1} (1 - (\kappa_{c,m,j}^t)^2) \\
&\quad + is_c \tilde{\alpha}_{m,j,i,c} \gamma_i (\kappa_{g,m,j}^t)^{i-1} (w_c^{*\top} w_g^* - \kappa_{c,m,j}^t \kappa_{g,m,j}^t) \Big] \\
&\quad - \frac{|\kappa_{c,m,j}^t|(\eta^t)^2 A_1^2 d}{2} - \frac{(\eta^t)^3 A_1^3 d^{\frac{3}{2}}}{2} + \eta^t w_c^{*\top}(I_d - w_{m,j}^t w_{m,j}^{t\top}) \Xi_{w_{m,j}}^t,
\end{aligned}
$$

where we introduced a mean-zero random variable $\Xi_{w_{m,j}}^t = -\nabla_{w_{m,j}} \mathbb{E}[\mathbf{1}(m(x_c) = m) y_c a_{m,j} \sigma_m(w_{m,j}^{t\top} x_c + b_{m,j})] + \nabla_{w_{m,j}} \mathbf{1}(m(x_c) = m) y_c a_{m,j} \sigma_m(w_{m,j}^{t\top} x_c + b_{m,j})$ in the first inequality. We used similar decomposition to Lemma C.7 in the second inequality and adopt the notation $\frac{i\alpha_{m,j,i}}{\sqrt{i!}} + \sum_{l=i+1}^{\infty} \Big( \frac{l\alpha_{m,j,l}}{\sqrt{l!}} \binom{l-1}{l-i} (\rho w_{m,j}^{t\top} v_{c'})^{l-i} \Big) = \frac{i\tilde{\alpha}_{m,j,i,c'}^t}{\sqrt{i!}}$ in the third inequality.

In the same way, we obtain an upper bound of $\kappa_{c,m,j}^t$.

$$
\begin{aligned}
\kappa_{c,m,j}^{t+1} &\leq \kappa_{c,m,j}^t + \eta^t w_c^{*\top}(I_d - w_{m,j}^t w_{m,j}^{t\top}) \nabla_{w_{m,j}} \mathbb{E}[\mathbf{1}(m(x_c) = m) \pi_{m(x_c)} y_c a_{m,j} \sigma_m(w_{m,j}^{t\top} x_c + b_{m,j})] \\
&\quad + \frac{|\kappa_{c,m,j}^t|(\eta^t)^2 A_1^2 d}{2} + \frac{(\eta^t)^3 A_1^3 d^{\frac{3}{2}}}{2} + \eta^t w_c^{*\top}(I_d - w_{m,j}^t w_{m,j}^{t\top}) \Xi_{w_{m,j}}^t \\
&\leq \kappa_{c,m,j}^t + \eta^t \sum_{i=k^*}^{p^*} \Big[ i\tilde{\alpha}_{m,j,i,c} \beta_{c,i} (\kappa_{c,m,j}^t)^{i-1} (1 - (\kappa_{c,m,j}^t)^2) + is_c \tilde{\alpha}_{m,j,i,c} \gamma_i (\kappa_{g,m,j}^t)^{i-1} (w_c^* w_g^* - \kappa_{c,m,j}^t \kappa_{g,m,j}^t) \Big] \\
&\quad + \frac{|\kappa_{c,m,j}^t|(\eta^t)^2 A_1^2 d}{2} + \frac{(\eta^t)^3 A_1^3 d^{\frac{3}{2}}}{2} + \eta^t w_c^{*\top}(I_d - w_{m,j}^t w_{m,j}^{t\top}) \Xi_{w_{m,j}}^t.
\end{aligned}
$$

Similarly, we carry out the corresponding calculations for $\kappa_{g,m,j}^t$.

$$
\begin{aligned}
\kappa_{g,m,j}^{t+1} &\geq \kappa_{c,m,j}^t + \eta^t \sum_{i=k^*}^{p^*} \Big[ i\tilde{\alpha}_{m,j,i,c} \beta_{c,i} (\kappa_{c,m,j}^t)^{i-1} (w_c^{*\top} w_g^* - \kappa_{c,m,j}^t \kappa_{g,m,j}^t) \\
&\quad + is_c \tilde{\alpha}_{m,j,i,c} \gamma_i (\kappa_{g,m,j}^t)^{i-1} (1 - (\kappa_{g,m,j}^t)^2) \Big]
\end{aligned}
$$

$$- \frac{|\kappa_{g,m,j}^t|\eta^{t^2}A_1{}^2 d}{2} - \frac{(\eta^t)^3 A_1{}^3 d^{\frac{3}{2}}}{2} + \eta^t w_g^{*\top}(I_d - w_{m,j}^t w_{m,j}^{t}{}^\top)\Xi_{w_{m,j}}^t$$

and

$$\kappa_{g,m,j}^{t+1} \leq \kappa_{c,m,j}^t + \eta^t \sum_{i=k^*}^{p^*} \left[ (i\tilde{\alpha}_{m,j,i,c}\beta_{c,i}(\kappa_{c,m,j}^t)^{i-1}(w_c^{*\top}w_g^* - \kappa_{c,m,j}^t\kappa_{g,m,j}^t) \right.$$
$$\left. + is_c\tilde{\alpha}_{m,j,i,c}\gamma_i(\kappa_{g,m,j}^t)^{i-1}(1 - (\kappa_{g,m,j}^t)^2) \right]$$
$$+ \frac{|\kappa_{g,m,j}^t|(\eta^t)^2 A_1{}^2 d}{2} + \frac{(\eta^t)^3 A_1{}^3 d^{\frac{3}{2}}}{2} + \eta^t w_g^{*\top}(I_d - w_{m,j}^t w_{m,j}^{t}{}^\top)\Xi_{w_{m,j}}^t.$$

Next, we introduce auxiliary sequences to establish the following bounds for $\kappa_{c,m,j}^t$ and $\kappa_{g,m,j}^t$. The derivation follows the same approach as the proof of Lemma C.12 .

- Consider one neuron $j \in \mathcal{J}_c$ and suppose that $\kappa_{c,m,j}^s \leq a_2$, $|\kappa_{g,m,j}^s| \leq \kappa_{c,m,j}^s$, and $\kappa_{g,m,j}^s \leq A_2 A_3 d^{-\frac{1}{2}}$ for all $s = 0, 1, \ldots, t$. Then, $\kappa_{c,m,j}^s$ is lower bounded by $P_{\text{III},c}^s$ for all $s = 0, 1, \ldots, t+1$, where the sequence $\left(P_{\text{III},c}^s\right)_{s=0}^{t+1}$ is defined recursively as follows:

$$P_{\text{III},c}^0 = (1 - a_2)\kappa_{c,m,j}^s, \text{ and}$$
$$P_{\text{III},c}^{s+1} = P_{\text{III},c}^s + \eta^s a_2 k^* \tilde{\alpha}_{m,j,k^*for,c}\beta_{c,k^*}(P_{\text{III},c}^s)^{k^*-1} \text{ for } s \geq 0,$$

  with high probability.

  While, $\kappa_{g,m,j}^s$ is upper bounded by $Q_{\text{III},g}^s$ for all $s = 0, 1, \ldots, t+1$, where the sequence $\left(Q_{\text{III},g}^s\right)_{s=0}^{t+1}$ is defined recursively as follows:

$$Q_{\text{III},g}^0 = (1 + a_2)\max\left\{|\kappa_{g,m,j}^s|, \frac{1}{2}d^{-\frac{1}{2}}\right\}, \text{ and}$$
$$Q_{\text{III},g}^{s+1} = Q_{\text{III},g}^s + (1 + a_2)\eta^s k^*|s_c\tilde{\alpha}_{m,j,k^*,c}\gamma_{k^*}|(Q_{\text{III},g}^s)^{k^*-1}$$
$$+ A_3\eta^s k^*\tilde{\alpha}_{m,j,k^*,c}\beta_{c,k^*}(\kappa_{c,m,j}^s)^{k^*-1}d^{-\frac{1}{2}} \text{ for } s \geq 0,$$

  with high probability.

- Consider one neuron $j \in \mathcal{J}_g$ and suppose that $\kappa_{g,m,j}^s \leq a_2$, $|\kappa_{c,m,j}^s| \leq \kappa_{g,m,j}^s$, and $\kappa_{c,m,j}^s \leq A_2 A_3 d^{-\frac{1}{2}}$ for all $s = 0, 1, \ldots, t$. Then, $\kappa_{g,m,j}^s$ is lower bounded by $P_{\text{III},g}^s$ for all $s = 0, 1, \ldots, t+1$, where the $\left(P_{\text{III},g}^s\right)_{s=0}^{t+1}$ is defined recursively as follows:

$$P_{\text{III},g}^0 = (1 - a_2)\kappa_{g,m,j}^s, \text{ and}$$
$$P_{\text{III},g}^{s+1} = P_{\text{III},g}^s + \eta^s a_2 k^* s_c\tilde{\alpha}_{m,j,k^*,c}\gamma_{k^*}(P_{\text{III},g}^s)^{k^*-1}, \text{ for } s \geq 0,$$

  with high probability.

  While, $\kappa_{c,m,j}^s$ is upper bounded by $Q_{\text{III},c}^s$ for all $s = 0, 1, \ldots, t+1$, where the $\left(Q_{\text{III},c}^s\right)_{s=0}^{t+1}$ is defined recursively as follows:

$$Q_{\text{III},c}^0 = (1 + a_2)\max\left\{|\kappa_{c,m,j}^s|, \frac{1}{2}d^{-\frac{1}{2}}\right\}, \text{ and}$$
$$Q_{\text{III},c}^{s+1} = Q_{\text{III},c}^s + (1 + a_2)\eta^s k^*|\tilde{\alpha}_{m,j,k^*,c}\beta_{c,k^*}|(Q_{\text{III},c}^s)^{k^*-1}$$
$$+ A_3\eta^s k^* s_c\tilde{\alpha}_{m,j,k^*,c}\gamma_{k^*}(\kappa_{g,m,j}^s)^{k^*-1}d^{-\frac{1}{2}} \text{ for } s \geq 0,$$

  with high probability.

Using these auxiliary sequences, we can deduce the following. The derivation is the same as that of Lemma C.13.

- Consider one neuron $j \in \mathcal{J}_c$ and take $\eta^t = \eta_e \leq a_4 d^{-\frac{k^*}{2}}$. Suppose that $\kappa_{c,m,j}^s \leq a_2$, $|\kappa_{g,m,j}^s| \leq \kappa_{c,m,j}^s$, and $|\kappa_{g,m,j}^s| \leq 4A_3 d^{-\frac{1}{2}}$ hold for $0, 1, \ldots, t$.. Then, if $\kappa_{c,m,j}^{t+1} \leq a_2$, $|\kappa_{g,m,j}^{t+1}| \leq \kappa_{c,m,j}^{t+1}$, and $|\kappa_{g,m,j}^{t+1}| \leq 4a_3 d^{-\frac{1}{2}}$ with high probability.

- Consider one neuron $j \in \mathcal{J}_g$ and take $\eta^t = \eta_e \leq a_4 d^{-\frac{k^*}{2}}$. Suppose that $\kappa_{g,m,j}^s \leq a_2$, $|\kappa_{c,m,j}^s| \leq \kappa_{g,m,j}^s$, and $|\kappa_{c,m,j}^s| \leq 4A_3 d^{-\frac{1}{2}}$ hold for $0, 1, \ldots, t$.. Then, if $\kappa_{g,m,j}^{t+1} \leq a_2$, $|\kappa_{c,m,j}^{t+1}| \leq \kappa_{g,m,j}^{t+1}$, and $|\kappa_{c,m,j}^{t+1}| \leq 4A_3 d^{-\frac{1}{2}}$ with high probability.

Suppose that $T_{3,1} = \lfloor (\eta_e k^*(k^* - 2)(1 - 5k^*)(\tilde{\alpha}_{m,j,k^*,c}\beta_{c,k^*})(P_{\text{III},c}^0)^{k^*-2})^{-1} \rfloor$, $\kappa_{c,m,j}^s \leq a_2$ with $j \in \mathcal{J}_c$, and $|\kappa_{g,m,j}^s| \leq \kappa_{c,m,j}^s$ and $|\kappa_{g,m,j}^s| \leq A_3 d^{-\frac{1}{2}}$ with $j \notin \mathcal{J}_c$ for all $s = 0, 1, ..., T_{3,1}$. Then, the above bounds holds for all $s = 0, 1, ..., T_{3,1}$ with high probability.

Thus, by Lemma A.4,

$$\kappa_{c,m,j}^t \geq P_{\text{III},c}^t \geq \frac{P_{\text{III},c}^0}{(1 - \eta_e k^*(k^* - 2)(1 - a_2)\tilde{\alpha}_{m,j,k^*,c}\beta_{c,k^*}(P_{\text{I}}^0)s)^{k^*-2}}$$

However, when $t = T_1$,

$$\kappa_{c,m,j}^t \geq P_{\text{III},c}^t \geq \frac{P_{\text{III},c}^0}{(1 - \eta_e k^*(k^* - 2)(1 - a_2)\tilde{\alpha}_{m,j,k^*,c}\beta_{c,k^*}(P_{\text{III},c}^0)s)^{k^*-2}}$$

$$\geq \frac{1}{(\eta_e k^*(k^* - 2)(1 - a_2)(\tilde{\alpha}_{m,j,k^*,c}\beta_{c,k^*}))^{\frac{1}{k^*-2}}} > 1.$$

This leads to a contradiction as $\kappa_{c,m,j}^{T_1} \leq 1$. Since $|\kappa_{g,m,j}^s| \leq 4A_3 d^{-\frac{1}{2}}$,

$$|\kappa_{c,m,j}^{t_{3,1}} - \kappa_{c,m,j}^{t_{3,1}-1}| \leq A_1 \eta_e \leq A_1 a_4 d^{-\frac{k^*}{2}} \leq A_3 d^{-\frac{1}{2}}.$$

Thus,

$$|\kappa_{c,m,j}^{t_{3,1}}| \leq |\kappa_{c,m,j}^{t_{3,1}-1}| + |\kappa_{c,m,j}^{t_{3,1}} - \kappa_{c,m,j}^{t_{3,1}-1}| \leq 5A_3 d^{-\frac{1}{2}}.$$

This concludes that there exists some time $t_{3,1} \leq T_{3,1} = \Theta(\eta_e^{-1} d^{\frac{k^*-2}{2}})$ such that $\kappa_{c,m,j}^{t_{3,1}} > a_2$ and $|\kappa_{g,m,j}^{t_{3,1}}| \leq 5A_3 d^{-\frac{1}{2}}$ for $j \in \mathcal{J}_c$ with high probability.

In the same way, suppose that $T_{3,1} = \lfloor (\eta_e k^*(k^* - 2)(1 - 5k^*)(s_c\tilde{\alpha}_{m,j,k^*,c}\gamma_{k^*})(P_{\text{III},g}^0)^{k^*-2})^{-1} \rfloor$, $\kappa_{c,m,j}^s \leq a_2$ with $j \in \mathcal{J}_g$, and $|\kappa_{c,m,j}^s| \leq \kappa_{g,m,j}^s$ and $|\kappa_{c,m,j}^s| \leq A_3 d^{-\frac{1}{2}}$ with $j \notin \mathcal{J}_g$ for all $s = 0, 1, ..., T_{3,1}$, and then, there exists some time $t_{3,1} \leq T_{3,1} = \Theta(\eta_e^{-1} d^{\frac{k^*-2}{2}})$ such that $\kappa_{g,m,j}^{t_{3,1}} > a_2$ and $|\kappa_{c,m,j}^{t_{3,1}}| \leq 5A_3 d^{-\frac{1}{2}}$ for $j \in \mathcal{J}_g$ with high probability.

$\square$

**Transition from weak to strong recovery.** Next, we show that the neuron for $j \in \mathcal{J}_c$ aligns with $w_c^*$ up to a large constant $1 - c_2$. We denote $t \leftarrow t - t_{3,1}$.

**Lemma C.25.** *Consider the expert $m \in \mathcal{M}_c$ and $j \in \mathcal{J}_c$, where $j$ satisfies Lemma C.21. Let $w_c^{*\top} w_g^* \leq A_4 d^{-\frac{1}{2}} = \tilde{O}(d^{-1/2})$ and $\eta^t = \eta_e \leq a_4 d^{-\frac{k^*}{2}}$. Then, with high probability, there exists some time $t_{3,2} \leq T_{3,2} = \tilde{\Theta}(\eta_e^{-1})$ such that $\kappa_{c,m,j}^{t_{3,2}} \geq 1 - a_2$. The same argument applies symmetrically when exchanging $c$ and $g$.*

*Proof.* Once the alignment reaches a constant level, the projection onto the spherical constraint via $(I_d - w_{m,j}w_{m,j}^\top)$ weakens the signal, requiring the reconstruction of the auxiliary sequences discussed in Lemma C.21.

Consider experts $m \in \mathcal{M}_c$.

Suppose that, for $j \in \mathcal{J}_c$, $\kappa_{c,m,j}^s \leq 1 - a_2$, $|\kappa_{g,m,j}^s| \leq \kappa_{c,m,j}^s$, and $\kappa_{g,m,j}^s \leq A_2 A_3 d^{-\frac{1}{2}}$ for all $s = 0, 1, \ldots, t$. Then, by using a similar inequality evaluation as in Lemma C.7 and introducing a mean-zero random variable $\Xi_{w_{m,j}}^t = -\nabla_{w_{m,j}} \mathbb{E}[\mathbf{1}(m(x_c) = m) y_c a_{m,j} \sigma_m({w_{m,j}^t}^\top x_c + b_{m,j})] + \nabla_{w_{m,j}} \mathbf{1}(m(x_c) = m) y_c a_{m,j} \sigma_m({w_{m,j}^t}^\top x_c + b_{m,j})$,

$$
\begin{aligned}
\kappa_{c,m,j}^{s+1} &\geq \kappa_{c,m,j}^s + \eta^s \sum_{i=k^*}^{p^*} \Big[ i \tilde{\alpha}_{m,j,i,c} \beta_{c,i} (\kappa_{c,m,j}^s)^{i-1} (1 - (\kappa_{c,m,j}^s)^2) \\
&\quad + i s_c \tilde{\alpha}_{m,j,i,c} \gamma_i (\kappa_{g,m,j}^s)^{i-1} ({w_c^*}^\top w_g^* - \kappa_{c,m,j}^s \kappa_{g,m,j}^s) \Big] \\
&\quad - \frac{|\kappa_{c,m,j}^s| (\eta^s)^2 A_1^2 d}{2} - \frac{(\eta^s)^3 A_1^3 d^{\frac{3}{2}}}{2} + \eta^s {w_c^*}^\top (I_d - w_{m,j}^s {w_{m,j}^s}^\top) \Xi_{w_{m,j}}^s \\
&\geq \kappa_{c,m,j}^s + \eta^s k^* \tilde{\alpha}_{m,j,k^*,c} \beta_{c,k^*} (\kappa_{c,m,j}^s)^{k^*-1} (1 - (1 - \kappa_{c,m,j}^s)^2) - p^{*2} \eta^s \max_i |s_c \tilde{\alpha}_{m,j,i,c} \gamma_i| (\kappa_{g,m,j}^s)^{k^*-1} |{w_c^*}^\top w_g^*| \\
&\quad - p^{*2} \eta^s \max_i |s_c \tilde{\alpha}_{m,j,i,c} \gamma_i| (\kappa_{g,m,j}^s)^{k^*} - (\eta^s)^2 \kappa_{c,m,j}^s A_1^2 d + \eta^s {w_c^*}^\top (I_d - w_{m,j}^s {w_{m,j}^s}^\top) \Xi_{w_{m,j}}^s.
\end{aligned}
$$

Since ${w_c^*}^\top w_g^* \leq A_4 d^{-\frac{1}{2}}$, $|\kappa_{g,m,j}^s| \leq A_2 A_3 d^{-\frac{1}{2}}$ and $\kappa_{c,m,j}^s \geq \frac{1}{2} d^{-\frac{1}{2}}$, $p^{*2} \eta^s \max_i |s_c \tilde{\alpha}_{m,j,i,c} \gamma_i| (\kappa_{g,m,j}^s)^{k^*-1} |{w_c^*}^\top w_g^*| \leq \frac{1}{4} a_2 \eta^s k^* \tilde{\alpha}_{m,j,k^*,c} \beta_{c,k^*} (\kappa_{c,m,j}^s)^{k^*-1}$, $p^{*2} \eta^s \max_i |s_c \tilde{\alpha}_{m,j,i,c} \gamma_i| (\kappa_{g,m,j}^s)^{k^*} \leq \frac{1}{4} a_2 \eta^s k^* \tilde{\alpha}_{m,j,k^*,c} \beta_{c,k^*} (\kappa_{c,m,j}^s)^{k^*-1}$ and $(\eta^s)^2 \kappa_{c,m,j}^s A_1^2 d \leq \frac{1}{4} a_2 \eta^s k^* \tilde{\alpha}_{m,j,k^*,c} \beta_{c,k^*} (\kappa_{c,m,j}^s)^{k^*-1}$.

For the noise term,

$$
\left| \sum_{s'=0}^s \eta^{s'} {w_c^*}^\top (I_d - w_{m,j}^{s'} {w_{m,j}^{s'}}^\top) \Xi_{w_{m,j}}^{s'} \right| \leq \begin{cases} a_2 \kappa_{c,m,j}^0, & \text{if } s \leq A_2 d^{-k^*+1}, \\ \frac{1}{4} a_2 \eta^s k^* \tilde{\alpha}_{m,j,k^*,c} \beta_{c,k^*} (\kappa_{c,m,j}^s)^{k^*-1}, & \text{if } s > A_2 d^{k^*-1} \end{cases}
$$

with high probability.

Therefore, by noting that $(1 - (1 - \kappa_{c,m,j}^s)^2) \leq \frac{7}{4} a_2$, $\kappa_{c,m,j}^s$ can be lower bounded as

$$
\kappa_{c,m,j}^s \geq (1 - a_2) \kappa_{c,m,j}^0 + a_2 \sum_{s'=0}^s \eta^{s'} k^* \tilde{\alpha}_{m,j,k^*,c} \beta_{c,k^*} (\kappa_{c,m,j}^{s'})^{k^*-1}.
$$

By introducing an auxiliary sequence $(P_{\mathrm{III},c}'^s)_{s=0}^{t+1}$, where

$$
\begin{aligned}
P_{\mathrm{III},c}'^0 &= (1 - a_2) \kappa_{c,m,j}^s, \text{ and} \\
P_{\mathrm{III},c}'^{s+1} &= P_{\mathrm{III},c}'^s + a_2 \eta^s k^* \tilde{\alpha}_{m,j,k^*,c} \beta_{c,k^*} (P_{\mathrm{III},c}'^s)^{k^*-1} \text{ for } s \geq 0,
\end{aligned}
$$

then $\kappa_{c,m,j}^s$ is lower bounded by $P_{\mathrm{III},c}'^s$ for all $s = 0, 1, \ldots, t+1$ with high probability.

In addition, with the same proof as Lemma C.21, by introducing an auxiliary sequence $(Q_{\mathrm{III},g}'^s)_{s=0}^{t+1}$, where

$$
\begin{aligned}
Q_{\mathrm{III},g}'^0 &= 6 A_3 d^{-\frac{1}{2}}, \text{ and} \\
Q_{\mathrm{III},g}'^{s+1} &= Q_{\mathrm{III},g}'^s + (1 + a_2) \eta^s k^* |s_c \tilde{\alpha}_{m,j,k^*,c} \gamma_{k^*}| (Q_{\mathrm{III},g}'^s)^{k^*-1} \\
&\quad + A_3 \eta^s k^* \tilde{\alpha}_{m,j,k^*,c} \beta_{c,k^*} (\kappa_{c,m,j}^s)^{k^*-1} d^{-\frac{1}{2}} \text{ for } s \geq 0,
\end{aligned}
$$

then $\kappa_{g,m,j}^s$ is upper bounded by $Q_{\mathrm{III},g}'^s$ for all $s = 0, 1, \ldots, t+1$ with high probability.

Similarly, for a neuron $j \in \mathcal{J}_g$, $\kappa_{g,m,j}^s$ is lower bounded by $P_{\mathrm{III},g}'^s$ for all $s = 0, 1, \ldots, t+1$ and $\kappa_{c,m,j}^s$ is upper bounded by $Q_{\mathrm{III},c}'^s$ for all $s = 0, 1, \ldots, t+1$ with high probability. $(P_{\mathrm{III},g}'^s)_{s=0}^{t+1}$ and $(Q_{\mathrm{III},c}'^s)_{s=0}^{t+1}$ are defined as follows:

$$
P_{\mathrm{III},g}'^0 = (1 - a_2) \kappa_{g,m,j}^s, \text{and}
$$

$$P_{\text{III},g}'^{s+1} = P_{\text{III},g}'^{s} + a_2\eta^s k^* s_c\tilde{\alpha}_{m,j,k^*,c}\gamma_{k^*}(P_{\text{III},g}'^{s})^{k^*-1} \text{ for } s \geq 0.$$

$$Q_{\text{III},c}'^{0} = 6A_3 d^{-\frac{1}{2}}, \text{ and}$$
$$Q_{\text{III},c}'^{s+1} = Q_{\text{III},c}'^{s} + (1+a_2)\eta^s k^*|\tilde{\alpha}_{m,j,k^*,c}\beta_{c,k^*}|(Q_{\text{III},c}'^{s})^{k^*-1}$$
$$+ A_3\eta^s k^* s_c\tilde{\alpha}_{m,j,k^*,c}\gamma_{k^*}(\kappa_{g,m,j}^s)^{k^*-1}d^{-\frac{1}{2}} \text{ for } s \geq 0.$$

Note that the periods in the notation of the auxiliary sequences $\left(P_{\text{III},c}'^{s}\right)_{s=0}^{t+1}$, $\left(P_{\text{III},g}'^{s}\right)_{s=0}^{t+1}$, $\left(Q_{\text{III},g}'^{s}\right)_{s=0}^{t+1}$, and $\left(Q_{\text{III},c}'^{s}\right)_{s=0}^{t+1}$ are intentionally used to distinguish them from the auxiliary sequences in Lemma C.21.

We prove the following arguments by induction using this auxiliary sequence, in the same manner as Lemma C.6.

Consider one neuron $j \in \mathcal{J}_c$ and take $\eta^t = \eta_e \leq a_4 d^{-\frac{k^*}{2}}$. Suppose that $\kappa_{c,m,j}^s \leq 1 - a_2$, $|\kappa_{g,m,j}^s| \leq \kappa_{c,m,j}^s$, and $|\kappa_{g,m,j}^s| \leq A_2 A_3 d^{-\frac{1}{2}}$ hold for all $0, 1, \ldots, t$. Then, if $\kappa_{c,m,j}^{t+1} \leq 1 - a_2$, $|\kappa_{g,m,j}^{t+1}| \leq \kappa_{c,m,j}^{t+1}$, and $|\kappa_{g,m,j}^{t+1}| \leq A_2 A_3 d^{-\frac{1}{2}}$ with high probability.

In the same way, consider one neuron $j \in \mathcal{J}_g$ and take $\eta^t = \eta_e \leq a_4 d^{-\frac{k^*}{2}}$. Suppose that $\kappa_{g,m,j}^s \leq 1-a_2$, $|\kappa_{c,m,j}^s| \leq \kappa_{g,m,j}^s$, and $|\kappa_{c,m,j}^s| \leq A_2 A_3 d^{-\frac{1}{2}}$ hold for all $0, 1, \ldots, t$. Then, if $\kappa_{g,m,j}^{t+1} \leq 1 - a_2$, $|\kappa_{c,m,j}^{t+1}| \leq \kappa_{g,m,j}^{t+1}$, and $|\kappa_{c,m,j}^{t+1}| \leq A_2 A_3 d^{-\frac{1}{2}}$ with high probability.

Consider one neuron $j \in \mathcal{J}_c$. Suppose that $\kappa_{c,m,j}^s \leq 1 - a_2$ hold for all $s = 0, 1, \ldots, T_{3,2}$, where

$$T_{3,2} = \lfloor (\eta_e k^*(k^* - 2)a_2\tilde{\alpha}_{m,j,k^*,c}\beta_{c,k^*}(P_{\text{III},c}'^{0})^{k^*-2})^{-1} \rfloor.$$

However, at $t = T_{3,2}$,

$$\kappa_{c,m,j}^s \geq P_{\text{III},c}'^{s} \geq \frac{P_{\text{III},c}'^{0}}{(1 - \eta_e k^*(k^* - 2)a_2\tilde{\alpha}_{m,j,k^*,c}\beta_{c,k^*}(P_{\text{III},c}'^{0})^{k^*-2})s)^{\frac{1}{k^*-2}}}$$
$$\geq \frac{P_{\text{III},c}'^{0}}{(\eta_e k^*(k^* - 2)a_2\tilde{\alpha}_{m,j,k^*,c}\beta_{c,k^*}(P_{\text{III},c}'^{0})^{k^*-2})^{\frac{1}{k^*-2}}} > 1.$$

This leads to contradiction. Thus, there exists some time $t_{3,2} \leq T_{3,2}$ such that $\kappa_{c,m,j}^{t_{3,2}} > 1 - a_2$. The same proof applies to $\kappa_{g,m,j}^{t_{3,2}}$ for $j \in \mathcal{J}_g$ by taking $T_{3,2} = \lfloor (\eta_e k^*(k^* - 2)a_2 s_c\tilde{\alpha}_{m,j,k^*,c}\gamma_{k^*}(P_{\text{III},c}'^{0})^{k^*-2})^{-1} \rfloor$.

$\square$

**Strong recovery.** Finally, we show that the neuron for $j \in \mathcal{J}_c$ amplifies the alignment with $w_c^*$, and the neuron for $j \in \mathcal{J}_g$ amplifies the alignment with $w_g^*$, and we establish strong recovery ($\kappa_{c,m,j}^t \geq 1 - \epsilon > 1 - a_2$ for $j \in \mathcal{J}_c$, and $\kappa_{g,m,j}^t \geq 1 - \epsilon > 1 - a_2$ for $j \in \mathcal{J}_g$).

**Lemma C.26.** *Consider the expert $m \in \mathcal{M}_c$ and $j \in \mathcal{J}_c$, where $j$ satisfies Lemma C.25. Let $w_c^{*\top} w_g^* \leq A_4 d^{-\frac{1}{2}} = \tilde{O}(d^{-1/2})$, $\eta^t = \eta_{e'} \leq a_4 d^{-\frac{k^*}{2}}$ for $0 \leq t \leq (T_{3,1} - t_{3,1}) + (T_{3,2} - t_{3,2}) - 1$ and $\eta^t = \eta_{e'} \leq \min\{\frac{a_4}{3}\epsilon d^{-1}, \frac{a_4}{9}\epsilon^2\}$ for $(T_{3,1} - t_{3,1}) + (T_{3,2} - t_{3,2}) \leq t \leq (T_{3,1} - t_{3,1}) + (T_{3,2} - t_{3,2}) + T_{3,3} - 1$. Then, $\kappa_{c_m^*,m,j_m^*}^{(T_{3,1}-t_{3,1})+(T_{3,2}-t_{3,2})+T_{3,3}} > 1 - \epsilon$ where $T_{3,3} = \tilde{\Theta}(\epsilon^{-1}\eta_{e'}^{-1})$ holds with high probability.*

*The same argument applies symmetrically when exchanging $c$ and $g$.*

*Proof.* Consider experts $m \in \mathcal{M}_c$. Suppose that, for $j \in \mathcal{J}_c$, we have $1 - 2a_2 \leq \kappa_{c,m,j}^s \leq 1 - \frac{\epsilon}{3}$ for all $s = 0, 1, \ldots, t$. Then, we have

$$\kappa_{c,m,j}^{s+1} \geq \kappa_{c,m,j}^s + \eta^s k^*\tilde{\alpha}_{m,j,k^*,c}\beta_{c,k^*}(1 - (\kappa_{c,m,j}^s)^2)(\kappa_{c,m,j}^s)^{k^*-1}$$

$$+ \eta^s \sum_{i=k^*}^{p^*} i s_c\tilde{\alpha}_{m,j,i,c}\gamma_i(\kappa_{g,m,j}^s)^{i-1}(w_c^{*\top}w_g^* - \kappa_{c,m,j}^s\kappa_{g,m,j}^s) - \kappa_{c,m,j}^s\eta_{e'}^2 A_1^2 d + \eta_{e'}w_c^{*\top}(I_d - w_{m,j}^s w_{m,j}^{s\top})\Xi_{w_{m,j}}^s$$

$$\geq \kappa_{c,m,j}^s + \eta^s k^* \tilde{\alpha}_{m,j,k^*,c} \beta_{c,k^*} (1 - (\kappa_{c,m,j}^s)^2)(\kappa_{c,m,j}^s)^{k^*-1}$$

$$- \eta_{e'} p^{*2} \max_i |s_c \tilde{\alpha}_{m,j,i,c} \gamma_i| |w_c^{*\top} w_g^*| (1 - (\kappa_{c,m,j}^s)^2)$$

$$- \eta_{e'} p^{*2} \max_i |s_c \tilde{\alpha}_{m,j,i,c} \gamma_i| |\kappa_{g,m,j}^s|^{k^*-1} w_g^{*\top} (I_d - w_c^* w_c^{*\top}) w_{m,j}^s \kappa_{c,m,j}^s$$

$$- \kappa_{c,m,j}^s \eta_{e'}^2 A_1^2 d + \eta_{e'} w_c^{*\top} (I_d - w_{m,j}^s w_{m,j}^{s\top}) \Xi_{w_{m,j}}^s$$

$$\geq \kappa_{c,m,j}^s + \sum_{s'=0}^s \frac{6}{5} \eta^{s'} k^* \tilde{\alpha}_{m,j,k^*,c} \beta_{c,k^*} (1 - \kappa_{c,m,j}^{s'}) + \sum_{s'=0}^s \eta_{e'} w_c^{*\top} (I_d - w_{m,j}^{s'} w_{m,j}^{s'\top}) \Xi_{w_{m,j}}^{s'}.$$

Since $\kappa_{c,m,j}^s \geq 1 - 3a_2$, we have $w_g^{*\top}(I_d - w_c^* w_c^{*\top}) w_{m,j}^s \leq \sqrt{6a_2}$ and $\kappa_{g,m,j}^s = (w_g^{*\top} w_c^*)(w_c^{*\top} w_{m,j}^s) + w_g^{*\top}(I_d - w_c^* w_c^{*\top}) w_{m,j}^s \leq \sqrt{6a_2} + \tilde{O}(d^{-\frac{1}{2}})$. Hence, we obtain $\eta^s k^* \tilde{\alpha}_{m,j,k^*,c} \beta_{c,k^*} (1 - (\kappa_{c,m,j}^s)^2)(\kappa_{c,m,j}^s)^{k^*-1} \geq \frac{9}{5} \eta^s k^* \tilde{\alpha}_{m,j,k^*,c} \beta_{c,k^*} (1 - \kappa_{c,m,j}^s)$ by $\kappa_{c,m,j}^s \geq 1 - 3a_2$, $\eta_{e'} p^{*2} \max_i |s_c \tilde{\alpha}_{m,j,i,c} \gamma_i| |w_c^{*\top} w_g^*|(1 - (\kappa_{c,m,j}^s)^2) \leq \frac{1}{5} \eta^s k^* \tilde{\alpha}_{m,j,k^*,c} \beta_{c,k^*} (1 - \kappa_{c,m,j}^s)$ by $w_c^{*\top} w_g^* = \tilde{O}(d^{-\frac{1}{2}})$, $\eta_{e'} p^{*2} \max_i |s_c \tilde{\alpha}_{m,j,i,c} \gamma_i| |\kappa_{g,m,j}^s|^{k^*-1} w_g^{*\top} (I_d - w_c^* w_c^{*\top}) w_{m,j}^s \kappa_{c,m,j}^s \leq \frac{1}{5} \eta^s k^* \tilde{\alpha}_{m,j,k^*,c} \beta_{c,k^*} (1 - \kappa_{c,m,j}^s)$ by $\max_i |s_c \tilde{\alpha}_{m,j,i,c} \gamma_i| / \tilde{\alpha}_{m,j,k^*,c} \beta_{c,k^*} \lesssim \frac{1}{\sqrt{a_2}}$ when $\alpha_{m,j,i}, \rho w_{m,j}^\top v_c > 0$, and $\kappa_{c,m,j}^s \eta_{e'}^2 A_1^2 d \leq \frac{1}{5} \eta^s k^* \tilde{\alpha}_{m,j,k^*,c} \beta_{c,k^*} (1 - \kappa_{c,m,j}^s)$ by $\eta_{e'} \leq \frac{a_4}{3} \epsilon d^{-1}$. In addition, we have $|\sum_{s'=0}^s \eta_{e'} w_c^{*\top} (I_d - w_{m,j}^{s'} w_{m,j}^{s'\top}) \Xi_{w_{m,j}}^{s'}| \leq a_2 \epsilon + \frac{1}{5} \eta_{e'} k^* \tilde{\alpha}_{m,j,k^*,c} \beta_{c,k^*} (1 - \kappa_{c,m,j}^{s'})$ with high probability.

Therefore, if $1 - 3a_2 \leq \kappa_{c,m,j}^s \leq 1 - \frac{\epsilon}{3}$, for all $s = 0, 1, \ldots, t$,

$$\kappa_{c,m,j}^{s+1} \geq \kappa_{c,m,j}^0 - \frac{a_2}{3} \epsilon + \sum_{s'=0}^s \eta_{e'} k^* \tilde{\alpha}_{m,j,k^*,c} \beta_{c,k^*} (1 - \kappa_{c,m,j}^{s'})$$

$$\geq \kappa_{c,m,j}^0 - \frac{a_2}{3} \epsilon + \frac{1}{3} s \eta_{e'} k^* \tilde{\alpha}_{m,j,k^*,c} \beta_{c,k^*} \epsilon.$$

and $1 - 3a_2 \leq \kappa_{c,m,j}^{t+1}$ hold.

For $0 \leq t \leq (T_{3,1} - t_{3,1}) + (T_{3,2} - t_{3,2}) - 1$, it holds that $\kappa_{c,m,j}^{t+1} - \frac{\epsilon}{3} \geq 1 - 2a_2$ by taking $\eta^t = \eta_e \leq a_4 d^{-\frac{k^*}{2}}$. Take $\eta^t = \eta_{e'} \leq \min\{\frac{a_4}{3} \epsilon d^{-1}, \frac{a_4}{9} \epsilon^2\}$ and suppose that $\kappa_{c,m,j}^t \leq 1 - \frac{\epsilon}{3}$ and for all $(T_{3,1} - t_{3,1}) + (T_{3,2} - t_{3,2}) \leq t \leq (T_{3,1} - t_{3,1}) + (T_{3,2} - t_{3,2}) + T_{3,3} - 1$, where

$$T_{3,3} = \lfloor a_2 (\eta_{e'} \epsilon k^* \tilde{\alpha}_{m,j,k^*,c} \beta_{c,k^*})^{-1} \rfloor + 1.$$

Then, it holds that $\kappa_{c,m,j}^t \geq 1 - 3a_2 + \frac{1}{3} t \epsilon \eta_{e'} k^* \tilde{\alpha}_{m,j,k^*,c} \beta_{c,k^*}$ for all $(T_{3,1} - t_{3,1}) + (T_{3,2} - t_{3,2}) \leq t \leq (T_{3,1} - t_{3,1}) + (T_{3,2} - t_{3,2}) + T_{3,3} - 1$. However, at $t = (T_{3,1} - t_{3,1}) + (T_{3,2} - t_{3,2}) + T_{3,3}$, $1 - 3a_2 + \frac{1}{3} t \epsilon \eta_{e'} k^* \tilde{\alpha}_{m,j,k^*,c} \beta_{c,k^*} \geq 1$, which leads to contradiction. Thus, there exists some $t_{3,3} \leq (T_{3,1} - t_{3,1}) + (T_{3,2} - t_{3,2}) + T_{3,3}$ such that $\kappa_{c,m,j}^t \geq 1 - \frac{\epsilon}{3}$. When there exists some time $\kappa_{c,m,j}^t < 1 - \frac{\epsilon}{3}$ for $t > t_{3,3}$, $\kappa_{c,m,j}^t \geq 1 - 2a_2$ since $|\kappa_{c,m,j}^{t+1} - \kappa_{c,m,j}^t| \leq A_1 \eta_{e'}$. Thus, $\kappa_{c,m,j}^t > 1 - \epsilon$ holds for all $t_{3,3} \leq t \leq (T_{3,1} - t_{3,1}) + (T_{3,2} - t_{3,2}) + T_{3,3}$ until $\kappa_{c,m,j}^t > 1 - \frac{\epsilon}{3}$ holds. By recursively applying this step, we obtain the desired result. $\square$

## C.5. Second Layer Optimization Stage

We have i.i.d. test-time inputs $X^T = (x^1, \ldots, x^T)$ and we extract $T_c$ inputs $X_c^{T_c} = (x_c^1, \ldots, x_c^{T_c})$ in the cluster $c$ from $X^T$. Note that $\sum_c T_c = T$. We know that each $X_c^{T_c}$ is successfully routed to the expert $m \in \mathcal{M}_c$ with high probability over the randomness of $X^T$.

### C.5.1. APPROXIMATION OF SINGLE INDEX POLYNOMIALS

We suppose $\sigma_m$ are ReLU functions.

**Lemma C.27** (Following Damian et al. (2022); Oko et al. (2024a)). *Fix $c$ and the corresponding expert $m \in \mathcal{M}_c$ such that for all $t_c$, $h_m(x^{t_c}) \geq 0$ with high probability. Suppose that $b_j \sim \mathrm{Unif}([-C_b, C_b])$ with $C_b = \tilde{O}(1)$. Let $h_c(z)$ be a*

*polynomial with degree $q = O(1)$, $w \in \text{Unif}\,(\mathbb{S}^{d-1}(1))$, $w^- = -w$. Then there exists $a_1, \ldots, a_{2N} \in \mathbb{R}$ such that*

$$
\sup_{t_c = 1, \ldots, T_c} \left| \frac{1}{2N} \sum_{j=1}^{N} a_j \sigma_m(w^\top x_c^{t_c} + b_j) - \frac{1}{2N} \sum_{j=1}^{N} a_j \sigma_m(w^{-\top} x_c^{t_c} + b_j) - h_c(w^\top x_c^{t_c}) \right| = \tilde{O}(N^{-1}).
$$

*Moreover, we have $\sum_{j=1}^{2N} a_j^2 = \tilde{O}(N)$ and $\sum_{j=1}^{2N} |a_j| = \tilde{O}(N)$.*

We obtain similar approximation results for polynomial activations (see (Oko et al., 2024a) for details). Using Lemma C.27, we show that ,for all $c$, there exists some $m \in \mathcal{M}_c$ and $a_m^*$ such that $f_m$ can approximate $f_c^* + s_c g^*$.

**Lemma C.28** (Following Oko et al. (2024a)). *Let $\sigma_m$, $m = 1, \ldots, M$ be ReLU activations or polynomial activations. Fix $c$ and the corresponding expert $m \in \mathcal{M}_c$ such that for all $t_c$, $h_m(x^{t_c}) \geq 0$ with high probability. Assume $J \gtrsim J_{\min}\text{poly}\log d$. There exists some parameters $a_m^* = (a_{mj}^*)_j$ such that*

$$
\frac{1}{T_c} \sum_{t_c} \left( \sum_{m'=1}^{M} \frac{\mathbb{1}[h_{m'}(x_c^{t_c}) \geq 0]}{J} \sum_j a_{m'j}^* \sigma_{m'}(\hat{w}_{m'j}^\top x_c^{t_c} + b_j) - f_c^*(w_c^{*\top} x_c^{t_c}) - s_c g^*(w_g^{*\top} x_c^{t_c}) \right)^2 \lesssim \tilde{O}(|J_{\min}|^{-2} + \epsilon^2).
$$

*where $\|a_m^*\|_2^2 = \tilde{O}(J^2 |J_{\min}|^{-1})$, $\|a_m^*\|_1 = \tilde{O}(J\sqrt{C})$ and $a_{m'j}^* = 0$ for $m' \in \mathcal{M}_c \setminus \{m\}$.*

*Proof.* The main difference between the proof in (Oko et al., 2024a) is that we may have superfluous experts $m' \in \mathcal{M}_c \setminus \{m\}$. However, we only need to put $a_{m'j}^* = 0$ for all $m' \in \mathcal{M}_c \setminus \{m\}$ and $j = 1, \ldots, J$. $\qquad\square$

### C.5.2. OPTIMIZING THE SECOND LAYER

We present the result of optimization of the second layer:

**Lemma C.29.** *Suppose that $J = \Theta(J_{\min}\text{poly}\log d)$. There exists $\lambda > 0$ such that the ridge estimator $\hat{a}_m$ satisfies*

$$
\mathbb{E}_{x_c, c} \left[ \left| \sum_m \mathbb{1}[h_m(x_c^{t_c}) \geq 0] f_{m, \hat{a}_m}(x_c) - f_c^*(w_c^{*\top} x_c) - s_c g^*(w_g^{*\top} x_c) \right| \right] \lesssim \tilde{O}\left( (|J_{\min}|^{-1} + \epsilon) + \sqrt{\frac{\text{poly}\log d}{T}} \right).
$$

*with probability at least $1 - o_d(1)$. Therefore, by taking $|J_{\min}| = \tilde{O}(\epsilon^{-1})$ and $T = \tilde{O}(\epsilon^{-2})$, we have $\tilde{O}(\epsilon)$ loss.*

*Proof.* Let $\mathcal{A}_{\mathcal{M}}^* = \{\{\hat{a}_m\}_{m \in \mathcal{M}} \mid \sum_{m \in \mathcal{M}} \|\hat{a}_m\|_r \leq \sum_{m \in \mathcal{M}} \|a_m^*\|_r\}$. We know that the router $h$ exclusively route $X_c^{T_c}$ to the subset of experts $\subset \mathcal{M}_c$ with high probability. Therefore, the minimization problem of the empirical $L^2$ loss is decomposed as

$$
\min_{\{\hat{a}_m\}_m \in \mathcal{A}_{\{1,\ldots,M\}}^*} \frac{1}{T} \sum_t \left( \sum_{m=1}^{M} \mathbb{1}[h_m(x^t) \geq 0] f_{m, \hat{a}_m}(x^t) - y^t \right)^2
$$

$$
= \min_{\{\hat{a}_m\}_m \in \mathcal{A}_{\{1,\ldots,M\}}^*} \sum_c \frac{T_c}{T} \left[ \frac{1}{T_c} \sum_{t_c} \left( \sum_{m' \in \mathcal{M}_c} \mathbb{1}[h_{m'}(x_c^{t_c}) \geq 0] f_{m', \hat{a}_{m'}}(x_c^{t_c}) - y_c^{t_c} \right)^2 \right]
$$

$$
\leq \min_{\otimes_c \{\hat{a}_m\}_{m \in \mathcal{M}_c}, \text{ where } \{\hat{a}_m\}_{m \in \mathcal{M}_c} \in \mathcal{A}_{\mathcal{M}_c}^*} \sum_c \frac{T_c}{T} \left[ \frac{1}{T_c} \sum_{t_c} \left( \sum_{m' \in \mathcal{M}_c} \mathbb{1}[h_{m'}(x_c^{t_c}) \geq 0] f_{m', \hat{a}_{m'}}(x_c^{t_c}) - y_c^{t_c} \right)^2 \right]
$$

$$
= \sum_c \frac{T_c}{T} \min_{\{\hat{a}_{m'}\}_m \in \mathcal{A}_{\mathcal{M}_c}^*} \left[ \frac{1}{T_c} \sum_{t_c} \left( \underbrace{\sum_{m' \in \mathcal{M}_c} \mathbb{1}[h_{m'}(x_c^{t_c}) \geq 0] f_{m', \hat{a}_{m'}}(x_c^{t_c})}_{=: \hat{f}_c} - y_c^{t_c} \right)^2 \right] \, ,
$$

$$
\underbrace{\phantom{= \sum_c \frac{T_c}{T} \min_{\{\hat{a}_{m'}\}_m}}}_{=: \hat{\mathbb{E}}_{x_c}[\hat{f}_c - y_c]}
$$

where we used $h_{m'}(x_c^{t_c}) \geq 0 \Rightarrow m' \in \mathcal{M}_c$ with high probability, $\mathcal{M}_c \cap \mathcal{M}_{c'} = \emptyset$ for all $c \neq c'$, and $\forall c$, $\{\hat{a}_{m'}\}_m \in \mathcal{A}^*_{\mathcal{M}_c} \Rightarrow \{\hat{a}_m\}_m \in \mathcal{A}^*_{\{1,\ldots,M\}}$. Therefore, the optimization is performed in parallel for each subset of parameters $\{\hat{a}_m, m \in \mathcal{M}_c\}$ and we can bound the population loss as

$$
\mathbb{E}_c[\mathbb{E}_{x_c}[|\hat{f}_c - y_c|]]
$$
$$
\leq \sum_c \frac{1}{C} \left( \sup_{\hat{a}_{m'} \in \mathcal{A}^*_{\mathcal{M}_c}} \left\{ \mathbb{E}_{x_c}[|\hat{f}_c - y_c|] - \hat{\mathbb{E}}_{x_c}[|\hat{f}_c - y_c|] \right\} + \sqrt{\hat{\mathbb{E}}_{x_c}[(\hat{f}_c - y_c)^2]} \right).
$$

Applying Lemma 14 of (Oko et al., 2024a), we have

$$
\hat{\mathbb{E}}_{x_c}[(\hat{f}_c - y_c)^2] = \tilde{O}\left( \underbrace{|J_{\min}|^{-1} + \epsilon}_{\text{Approximation}} + \underbrace{\frac{1}{\sqrt{T_c}}}_{\text{Concentration}} \right)
$$

and the generalization error is bounded as

$$
\sup_{\hat{a}_{m'} \in \mathcal{A}^*_{\{m'\}}, \, m' \in \mathcal{M}_c} \left\{ \mathbb{E}_{x_c}[|\hat{f}_c - y_c|] - \hat{\mathbb{E}}_{x_c}[|\hat{f}_c - y_c|] \right\} \leq \tilde{O}\left( |J_{\min}|^{-1} + \epsilon + \frac{\text{poly} \log d}{\sqrt{T_c}} \right)
$$

with probability at least $1 - o_d(1)$ for the ridge estimator. $\qquad \square$

