# OpenReview forum: "Mixture of Experts Provably Detect and Learn the Latent Cluster Structure in Gradient-Based Learning"
_ICML.cc/2025/Conference — ICML 2025 poster_

### Official Review · Reviewer_wqTg · 2025-02-18

**Overall Recommendation:** 4

**Summary:**

The authors consider the problem of learning a mixture of $C$ tasks, plus one global task, with a mixture of two-layer network experts. They first establish that a single network is unable to recover the global signal, by constructing a special instance of this class of targets. They then turn to analyze the mixture of experts, trained with a stagewise algorithms in which different set of parameters are sequentially trained, and prove that this model is in contrast able to . A fine analysis of features learned at each stage is reached, with notably the first layer neurons specializing to different subtasks, and the routing weights then learning to point towards relevant experts, with the global task finally being learned.

**Claims And Evidence:**

The paper is primarily theoretical in nature. The authors bolster and illustrate their theoretical insights -- namely the specialization of experts and routing weights at different stages-- in convincing numerical experiments in Figs. 1,2 .

**Essential References Not Discussed:**

To the best of my knowledge of this line of works, I am not aware of an essential reference the authors fail to cite.

**Experimental Designs Or Analyses:**

The experiments presented in Fig. 1, 2 seem sound.

**Methods And Evaluation Criteria:**

The paper is primarily theoretical in nature.

**Other Comments Or Suggestions:**

I do not have major comments or suggestions.

**Other Strengths And Weaknesses:**

I am overall in favor of acceptance. The paper establishes a clear theoretical result for an interesting model, and connects to the body of works on information exponents, which I believe paves interesting bridges for future works. The theoretical results are clearly exposed, and sufficient discussion is provided whenever needed to build intuition (e.g. in subsection 3.3). I did not however carefully check the proofs, nor do I have the expertise to provide an educated reading of the latter.

**Questions For Authors:**

1. (Clarification question) Are there any conditions on the respective scaling of $M$ with $C$ for Theorem 4.7 to be applicable ? Is it the condition of Lemma 4.9?

**Relation To Broader Scientific Literature:**

The work is similar in spirit to the prior key work of (Chen et al., 2022) in the investigation of feature learning by mixtures of experts, although the task and setting differ. To the best of my understanding, the novelty of the present work lies in the fact that the complexity of the target function can be understood through its effective information exponent, thus connecting to a rich literature on this topic in the simpler setting of learning single or multi-index models with single networks (e.g. Arous et al., 2021). This allows for insightful explanations of the inability of single experts to learn the function, as discussed in 4.2.1.

**Theoretical Claims:**

I have not checked the proofs in detail. From my reading, the claims seem sound.

---

> ### Author Rebuttal · Authors · 2025-04-01
>
> # Reviewer wqTg,
> We sincerely thank the reviewer for their thoughtful evaluation and insightful questions.
>
> ## Q.1
> As the reviewer correctly noted, the conditions for the theorem rely on the inequality $M \gtrsim C \log (C / 0.001)$ in Lemma 4.9. In connection with this, while we initially described $M$ and $C$ as polylogarithmic in the input dimension $d$, we have since realized that ensuring reliable initialization requires both to be $O(1)$. We will revise the final version accordingly. This refinement is made for theoretical clarity and does not affect the main conclusions or practical relevance of our work. Aside from this adjustment, no other major changes to assumptions or results are necessary.
> As discussed in Oko et al. (2024a) in the context of feature learning, the Hermite coefficients of the student model can be perturbed by random initialization, a factor not considered in prior work on MoE classification (Chen et al., 2022). Our analysis builds on this observation to study robust specialization under such randomness. However, if the number of clusters $C$ is too large, the number of expert-to-router combinations grows rapidly, making the specialization problem significantly more challenging.
> We stress that this assumption is mainly to ensure **theoretical reliability**. In practice, MoE models often incorporate techniques such as auxiliary diversity losses in routing to promote **expert diversity** and avoid **premature collapse** (Cai et al., 2024). Our assumption reflects these practical strategies, and the insights into learning dynamics and specialization in MoE remain valid and relevant.
> ## Implications from our theoretical results
> While our study is theoretical in nature, it also carries practical implications. These implications are discussed in the **"Implications"** part of our response to Reviewer **YoCg**, which we kindly invite the reviewer to refer to.
>
> Due to character limits, we respectfully make use of this space to provide the references.
>
> ### References
> * Allen-Zhu & Li. (2023), Towards Understanding Ensemble, Knowledge Distillation and Self-Distillation in Deep Learning, ICLR.
> * Arous et al. (2021), Online stochastic gradient descent on non-convex losses from high-dimensional inference, JMLR.
> * Bietti et al. (2022), Learning single-index models with shallow neural networks, NIPS.
> * Cai et al. (2024). A survey on mixture of experts. arXiv preprint arXiv:2407.06204.
> Chen et al. (2022), Towards understanding the mixture-of-experts layer in deep learning, NIPS.
> * Chowdhury et al. (2023), Patch-level routing in mixture-of-experts is provably sample-efficient for convolutional neural networks, ICML.
> * Damian et al. (2022), Neural networks can learn representations with gradient descent, COLT.
> * Dayi & Chen. (2024), Gradient dynamics for low-rank fine-tuning beyond kernels, arXiv preprint arXiv:2411.15385.
> * Dudeja et al. (2018), Learning Single-Index Models in Gaussian Space, PMLR.
> * Ge et al. (2018), Learning One-hidden-layer Neural Networks with Landscape Design, ICLR.
> * Huang et al. (2024), Harder Tasks Need More Experts: Dynamic Routing in MoE Models, ACL.
> * Komatsuzaki et al. (2023), Sparse Upcycling: Training Mixture-of-Experts from Dense Checkpoints, ICLR.
> * Li et al. (2025), Theory on Mixture-of-Experts in Continual Learning, ICLR.
> * Liu et al. (2024), GRIN: GRadient-INformed MoE, arXiv preprint arXiv:2409.12136, 2024.
> * Mousavi-Hosseini et al. (2023), Gradient-Based Feature Learning under Structured Data, NIPS.
> * Oko et al. (2024a), Learning sum of diverse features: computational hardness and efficient gradient-based training for ridge combinations, COLT.
> * Oko et al. (2024b), Pretrained transformer efficiently learns low-dimensional target functions in-context, NIPS.
> * Pan et al. (2024), Dense Training, Sparse Inference: Rethinking Training of Mixture-of-Experts Language Models, arXiv preprint arXiv:2404.05567.
> * Roller et al. (2021), Hash Layers For Large Sparse Models, NIPS.
> * Shazeer et al. (2017), Outrageously Large Neural Networks: The Sparsely-Gated Mixture-of-Experts Layer, ICLR.
> * Simsek et al. (2025), Learning Gaussian Multi-Index Models with Gradient Flow: Time Complexity and Directional Convergence, CPAL.
> * Tang et al. (2025), Solving Token Gradient Conflict in Mixture-of-Experts for Large Vision-Language Model, ICLR.
> * Vural & Erdogdu. (2024), Pruning is Optimal for Learning Sparse Features in High-Dimensions, COLT.
> * Wei et al. (2024), Skywork-MoE: A Deep Dive into Training Techniques for Mixture-of-Experts Language Models, arXiv preprint arXiv:2406.06563.
> * Zeng et al. (2024), AdaMoE: Token-Adaptive Routing with Null Experts for Mixture-of-Experts Language Models, ACL.
> * Zhang et al. (2024), Diversifying the Expert Knowledge for Task-Agnostic Pruning in Sparse Mixture-of-Experts, arXiv preprint arXiv:2407.09590.
> * Zhou et al. (2022), Mixture-of-Experts with Expert Choice Routing, NIPS.

---

> > ### Comment · Reviewer_wqTg · 2025-04-02
> >
> > I thank the authors for the detailed clarifications. Regarding the revision of $M,C=O(1)$. If possible, could the authors provide clarifications on the following points:
> > - Does it change any of the statements of the technical results of the paper, or do they all carry through?
> > >  As discussed in Oko et al. (2024a) in the context of feature learning, the Hermite coefficients of the student model can be perturbed by random initialization
> > - Could the authors develop more on this point, or refer me to the relevant part of (Oko et al, 2024a) / their proofs?

---

> > > ### Author Response · Authors · 2025-04-04
> > >
> > > ## Reviewer wqTg,
> > >
> > > We greatly appreciate the reviewer’s thoughtful follow-up questions.
> > >
> > > Regarding the first point, the main results of the paper **remain unchanged and carry through**. However, we would like to revise the assumptions here to ensure the reliable initialization in Phase I below, which also addresses the second part of the question.
> > >
> > > In the original analysis, we did not fully account for the combinatorial growth in the number of expert-neuron and cluster assignments as $M$, $J$, and $C$ increase, which leads to a failure probability in initialization that becomes unacceptably large. Additionally, our original proof required collecting $O(\epsilon^{−1})$ specialized neurons per expert during initialization. In the revised version, we instead target **the specialization of a single neuron per expert** in Phase I, which simplifies the initialization argument and improves reliability under high-dimensional scaling.
> > > We reformulate the assumption as $M ,C = O(1)$, and $J$ satisfies $J \lesssim \sqrt{\log d}$. In addition, we introduce a mild technical assumption on the teacher model: the minimum order non-zero Hermite coefficients have approximately equal magnitudes across clusters.
> > >
> > > This constraint on $J$ in turn imposes a condition on the estimation error $\epsilon$, requiring $\epsilon^{-1}\lesssim \sqrt{\log d}$. Similar assumptions, in which $\epsilon\to 0$ as $d\to \infty$, appear in Definition 4 of Abbe et al. (2022), Corollary 9 of Abbe et al. (2023), and Theorem 1 of Braun et al. (2025).
> > >
> > > The revised argument in Phase I resembles the approach of Chen et al. (2022), rather than that of Oko et al. (2024a). Specifically, we use the early part of the proof of Lemma 2 of Oko et al. (2024a) to construct i.i.d. Gaussian variables, and then reduce the argument to the initialization framework of Chen et al. (2022). In this phase, weak recovery leads to the specialization of a single neuron per expert.
> > >
> > > Following the perspective of Oko et al. (2024a), we analyze this process through the Hermite expansion, which motivates the additional assumptions. The reason is as follows: To ensure robustness, we derive a condition for the specialization of a neuron during weak recovery based on the methods introduced in Chen et al. (2022), while accounting for interactions among neurons. As in Lemma 2 of Oko et al. (2024a), we must ensure that the margin $(\delta)^{(p-2)/2}$, where $\delta$ is defined in their lemma, is preserved. This quantity provides robustness under interactions and should remain greater than $(\log d)^{-1}$. This requires $M J^2 C^2\lesssim \log d$, which is satisfied with $M,C=O(1)$ and $J\lesssim \sqrt{\log d}$. As shown in Corollary 15 of Oko et al. (2024a), it is required that the factor $(C_p)^{(p-2)/2}$ satisfy
> > > $$
> > > (C_p)^{(p-2)/2} = 1 + O\left(\frac{1}{M J^2 C^2}\right) = 1 + O\left((\log d)^{-1}\right).
> > > $$
> > > Note that Lemma D.4 and Lemma F.1 in Chen et al. (2022) provide relevant discussion on this point. According to Corollary 15 of Oko et al. (2024a), $(C_p)^{(p-2)/2}$ corresponds to the ratio of absolute values of the lowest-order Hermite coefficients. Hence, the assumption that these coefficients are within $1 + O((\log d)^{-1})$ of each other is necessary.
> > > Furthermore, since the effective signal-to-noise ratio available to the router decreases with increasing network width $J$, we require $\epsilon^{-1}$ and $J \leq \sqrt{\log d}$ to ensure that the signal can be reliably detected during initialization.
> > >
> > > We will clarify these technical points and include more explicit references to the relevant parts of Chen et al. (2022) and Oko et al. (2024a) in the final version.
> > >
> > >
> > > ### Additional References
> > > * Abbe et al. (2022), The merged-staircase property: a necessary and nearly sufficient condition for SGD learning of sparse functions on two-layer neural networks, COLT.
> > > * Abbe et al. (2023), SGD learning on neural networks: leap complexity and saddle-to-saddle dynamics. COLT.
> > > * Braun et al. (2025), Learning a single index model from anisotropic data with vanilla stochastic gradient descent, arXiv preprint. arxiv:2503.23642.

---

### Official Review · Reviewer_bEDR · 2025-02-24

**Overall Recommendation:** 3

**Summary:**

This paper theoretically studies the learning dynamics of Mixture-of-Experts (MoE) models in nonlinear regression tasks with an underlying cluster structure. The main contribution of the paper is listed below

- Proves that a standard neural network fails to detect and exploit the latent cluster structure, while MoE successfully separates and learns simpler subproblems.
- Analyzes the SGD training dynamics of MoE, showing that experts weakly recover cluster-specific functions, and the router learns to assign data correctly.
- Establishes polynomial time and sample complexity, demonstrating MoE’s efficiency in learning clustered tasks.
- Proposes and theoretically validates a multi-phase training algorithm for MoE, alternating between expert learning and router learning to optimize specialization.

**Claims And Evidence:**

The paper provides strong theoretical proofs under specific assumptions like population gradient flow and orthogonal features. However, these conditions may be too restrictive for real-world scenarios, and the claimed failure of vanilla networks is shown under specific setups, so it may not be generalized.

**Essential References Not Discussed:**

The authors could add additional

**Experimental Designs Or Analyses:**

The experiments are limited in scope but provide valuable insights.

**Methods And Evaluation Criteria:**

The paper’s chosen regression setup (with latent clusters and a shared global component) generally aligns with its theoretical goals, making the methods and evaluation criteria reasonable for investigating MoE’s advantages in that specific setting. However, the multi-phase training procedure is not particularly relevant to state-of-the-art deep learning applications, though it provides valuable theoretical insights.

Moreover, the authors considered two routing mechanisms: top-1 routing and threshold-based routing, the latter of which is not a standard approach. These choices were likely made to simplify the theory, but they resulted in an oversimplification.

**Other Comments Or Suggestions:**

N/A

**Other Strengths And Weaknesses:**

- The application of theoretical ideas from feature learning to MoE is novel and interesting to me.

- The presentation of the theoretical results and problem setup is unnecessarily complex. I strongly encourage the authors to refine the notation for better clarity and readability.

**Questions For Authors:**

1. How do the theoretical results change with different routing mechanisms?

2. How might your analysis extend to scenarios where the cluster signals and global signal are only partially orthogonal or have overlapping directions, and do you expect the router to still reliably separate clusters under weaker assumptions?

**Relation To Broader Scientific Literature:**

This paper takes a step toward understanding MoE training. While the theoretical scope is overly simplified, it serves as a solid starting point.

**Theoretical Claims:**

I briefly reviewed the proofs, and the theoretical results appear overly complex and dense, making them harder to follow than necessary.

---

> ### Author Rebuttal · Authors · 2025-04-01
>
> # Reviewer bEDR,
> We sincerely thank the reviewer for the thoughtful and constructive feedback. We will incorporate the suggestions into the final version. Below, we address the reviewer’s comments. For references, please see our response to Reviewer **wqTg** due to character limits.
> ## Multi-phase training
> We agree that the multi-phase training procedure deviates from practical applications. However, multi-stage optimization is often employed to reveal the intrinsic nature of complex optimization problems and is commonly used in studies of feature learning (Arous et al., 2021; Ge et al., 2018; Gudeja et al., 2018; Bietti et al., 2022; Damian et al., 2022; Mousavi-Hosseini et al., 2024;Oko et al., 2024ab). Joint training of the router and experts is promising, but in nonlinear regression, it poses intractable non-convex challenges and is left for future work.
> ## Assumptions and practical relevance
> Since obtaining negative results for vanilla neural networks is mathematically challenging, we followed the approach of using gradient flow, as in Mousavi-Hosseini et al. (2024) and Simsek et al. (2025). In addition, while Chen et al. (2022) assume all signals are orthogonal, our analysis only requires orthogonality to the mean vector. Our theoretical results remain rich in implications. Please refer to the **"Implications"** section of our response to Reviewer **YoCg**.
> ## Router
> We would like to clarify that our theoretical routing mechanism is not an oversimplification, but rather a **principled approach** to addressing fundamental technical challenges in MoE without practical heuristics.
> In Phase II, training is performed using top-1 routing with added random noise $r_m$. After this phase, **a data $x_c$ is no longer routed to experts $m' \notin \mathcal{M}_c$** (where $\mathcal{M}\_c$ is the set of professional experts) due to the negative correlation between its router weight $\theta_{m'}$ and the cluster center vector $v_c$.(Lemma 4.11 and Lemma C.16). However, it remains possible that **competition among experts $m \in \mathcal{M}_c$** leads to load imbalance if we employ top-1 routing. This phenomenon differs from the classification setting in Chen et al. (2022) and Chowdhury et al. (2023), and stems from the need to estimate a continuous function in the regression setting.
> We analytically determine the cause and employ an **adaptive top-$k$** routing in Phases III and IV, where an expert $m \in \mathcal{M}_c$ is activated if it satisfies a sufficient condition $h_m(x_c) \geq 0$, ensuring that it covers the data distribution of cluster $c$. The value of $k$ is **adaptively** determined based on the data, a strategy that is also adopted in practice (Huang et al., 2024; Zeng et al., 2024). Importantly, while the adaptive top-$k$ arises from a theoretical and technical challenge, practical implementations often incorporate auxiliary losses or dense training to address load imbalance (see lines 126–131 in the right column), ensuring consistency between theory and practice.
> ## Q.1
> * **Expert choice routing (ECR)**:
> Since ECR (Zhou et al., 2022) uses the same gating network as our token choice routing (TCR) (Shazeer et al., 2017), both methods identify clusters through differences in expert recovery reflected in the gradients. The key difference is that TCR selects top-k experts per token, while ECR selects top-k tokens per expert, potentially dropping tokens unselected by any expert. We believe similar theoretical results are attainable if token drop is mitigated, e.g., via expert capacity constraints.
> * **Dense MoE**:
> Dense training (Pan et al., 2024) activates all experts without applying top-k selection to the softmax outputs. Since all experts are activated, the gradient for expert $m$ depends on a comparison with a weighted average over all experts, potentially causing gradient competition. Specifically, when weak recovery occurs across experts $m \in \mathcal{M}_c$, differences in gradient magnitudes can distort the update direction and complicate theoretical analysis. Therefore, similar guarantees in the dense setting demand extra assumptions or modified algorithms.
> * **Hash-based Routing**:
> Since hash-based routing (Roller et al., 2021) is non-learning, misaligned partitioning of the cluster structure may induce gradient interference and hinder theoretical guarantees.
> ## Q.2
> If all signals ($w^*_i$, $v_j$) have overlaps of order $\tilde{O}(d^{-1/2})$, then by carefully bounding the resulting overlap terms, we believe the MoE can still reliably learn the underlying cluster structure. Specifically, the gating network's gradient includes additional terms of magnitude $\tilde{O}(d^{-1/2})$ relative to the main signal, arising across task–neuron combinations.
> In contrast, if the overlap is of order $\tilde{\Omega}(1)$, the dynamics become profoundly more complex. Thus, we are unable to provide a definitive conclusion in this case.
>
> We are happy to clarify any remaining questions during the discussion.

---

> > ### Comment · Reviewer_bEDR · 2025-04-04
> >
> > Thank you to the authors for the thoughtful rebuttal and clarifications. I appreciate the effort you made to address the concerns raised. I will be maintaining my weak accept score. While I believe the theoretical setup considered in the paper is somewhat oversimplified, it provides a useful and focused starting point for understanding the training dynamics of MoE models.

---

### Official Review · Reviewer_VrRr · 2025-03-11

**Overall Recommendation:** 3

**Summary:**

This paper presents a comprehensive analysis of the sample and computational complexity of Mixture of Experts (MoEs) when optimized using stochastic gradient descent (SGD) for a regression problem.

**Claims And Evidence:**

Based on the data model proposed in Assumption 3.2, all the claims appear highly likely to be correct.

**Essential References Not Discussed:**

N/A

**Experimental Designs Or Analyses:**

Good.

**Methods And Evaluation Criteria:**

N/A

**Other Comments Or Suggestions:**

1. A numerical justification using real data is needed to support Assumption 3.2.

2. The theoretical results and experiments should be presented separately, or at least, a high-level summary of the theoretical findings should be provided. Currently, it is difficult to discern the major implications of your theoretical results. Additionally, it would be beneficial to highlight key differences from existing works, such as Theorem 4.3.

3. Real-data numerical experiments are necessary to further validate the theoretical findings.

----------------------------
Post-rebuttal comments:

I will keep my original score (3). I appreciate the new theoretical insights introduced in this paper; however, these insights need to be further validated through experiments on real data. Without such empirical justification, the impact remains limited.

**Other Strengths And Weaknesses:**

N/A

**Questions For Authors:**

Any thoughts on improving the structure or training process of MoE from your theoretical results?

**Relation To Broader Scientific Literature:**

N/A

**Theoretical Claims:**

The Appendix lacks a clear structure and primarily consists of a dense collection of theoretical results without sufficient explanatory text. As a result, I find it difficult to follow most of the proofs.

---

> ### Author Rebuttal · Authors · 2025-04-01
>
> # Reviewer VrRr,
> We sincerely thank the reviewer's insightful and constructive comments. In the final version, we will separate the presentation of theoretical results and experiments, and further elaborate on the writing in the supplementary material. For the theoretical results, we will include a high-level summary, detail implications, and highlight key differences from prior works. We clarify each point and question below.
> Owing to the character limits, we respectfully refer the reviewer to our author response to Reviewer **wqTg** for the references.
>
> ## Implication
> Please refer to the **"Implications"** part in our response to Reviewer **YoCg**.
>
> ## Key differences from existing works
> Prior MoE optimization studies (Chen et al., 2022; Chowdhury et al., 2023) focus on binary classification with noisy features. In contrast, we analyze regression and highlight how **gradient interference** (see lines 125-127 in the left column) across clusters weakens the signal and effectively increases the **information exponent**, revealing a clear separation between vanilla neural networks and MoE in Theorem 4.3. Although Li et al. (2025) also tackle regression, their linear setting does not account for the **non-convex** exploration phase we uncover.
> Regarding information exponents in feature learning, Oko et al. (2024a) assume additive task structure within single data points, whereas we study **additive structure across clusters**. Theorem 4.7 is the first to apply nonlinear sample complexity analysis to MoE, showing how the router’s gradients reflect expert specialization via **weak recovery** differences.
>
> ## Real-data experiments
> We certainly appreciate the importance of experiments on real-world data, as the reviewer has rightly pointed out. However, our study is theoretical in nature. Real-world datasets inevitably contain noise, which makes it challenging to isolate specific phenomena in a controlled and interpretable manner. Therefore, we conducted controlled experiments using synthetic data. Within the line of work on the complexity of learning low-dimensional nonlinear functions in specific architectures, Dayi & Chen (2024) and Oko et al. (2024b) also rely on synthetic experiments based on Gaussian data, while Vural & Erdogdu (2024) do not provide empirical results. In the final revision, we will include the more detailed learning process of MoE based on synthetic experiments.
>
> ## Q.
> ### Structure of MoE
> * **Gradient-aware routing:**
> While our analysis shows that the MoE mitigates gradient interference by explicitly partitioning the network, the number of experts is typically selected heuristically in practice. This raises the possibility that integrating gradient-aware routing mechanisms could lead to more efficient and adaptive MoE architectures. Recent works also explore the role of gradient interference in MoE (Liu et al., 2024; Tang et al., 2025).
>
> * **Adaptive routing:**
> To prevent competition among professional experts ($\in \mathcal{M}_c$), we adopted top-$k$ routing to avoid potential load imbalance. Thus, dynamically adjusting the number of activated experts $k$ is expected to stabilize training. This insight stems from our theoretical studies in nonlinear regression and aligns with recent NLP research (Huang et al., 2024; Zeng et al., 2024), where $k$ is adaptively varied based on tokens.
>
> * **Freezing (pruning) redundant experts:**
> To mitigate competition among professional experts, reducing redundant experts is also worth exploring, potentially lowering deployment cost. Recent work has investigated expert merging in this context (Zhang et al., 2024).
>
> ### Training process of MoE
> * **Upcycling from dense checkpoints:**
> We argued that meaningful router learning requires **differences** in weak recovery among experts, which in turn necessitates a **long exploration stage** due to the non-convex optimization. In this light, upcycling dense MLP checkpoints trained on distinct domains may offer a practical approach to accelerate convergence. This idea has seen several empirical successes in recent LLMs (Komatsuzaki et al., 2023; Wei et al., 2024).
>
> * **Stage-specific routing:**
> The noise $r_m$ introduced during Phase II ensures uniform gradient flow and sufficient signal for all experts. In contrast, the adaptive top-$k$ routing in Phase III and IV addresses competition among professional experts. This distinction—different challenges occur at each learning stage—suggests that stage-specific routing strategies may be helpful for effective MoE training.
>
> We welcome any further questions or concerns that may arise and would be pleased to provide clarification.

---

### Official Review · Reviewer_YoCg · 2025-03-13

**Overall Recommendation:** 2

**Summary:**

In this paper, the authors investigate the sample and runtime complexity of Mixture-of-Experts (MoE) optimized with the stochastic gradient descent when learning a regression task with an underlying cluster structure of single index models. In particular, they show that a single neural network cannot detect a latent cluster structure. On the other hand, MoE is capable of doing that as it aggregates the ability of each expert to weakly recover the simpler function associated with an individual cluster. All the results are proved rigorously. Finally, they conduct some numerical experiments to justify the theory.

**Claims And Evidence:**

Yes, the results are supported by both rigorous proofs and empirical evidences.

**Essential References Not Discussed:**

Most of the relevant references have been cited. However, I encourage the authors to cite at least one reference for the teacher models in Assumption 3.2. If this model is first introduced by this paper, then the authors should elaborate on the formulation of the data generation process.

Additionally, in lines 096-098 in the right column, the authors should add references for the claim "The complexity of
learning a nonlinear function via two-layer neural network optimized by SGD is closely associated with this value".

**Experimental Designs Or Analyses:**

Yes, I checked the experimental setup on page 6. The experiments are for justifying the theory only.

**Methods And Evaluation Criteria:**

Yes.

**Other Comments Or Suggestions:**

1. In the Assumption 3.2, I think $v_c$ should be $v^*_c$. Please feel free to correct me if I am wrong.

2. In line 111 in the left column, should $g^*_j$ be $g^*$?

**Other Strengths And Weaknesses:**

**Strengths:**

1. The problem considered in this paper is of interest due to the recent success of Mixture-of-Experts in large-scale models.

2. This theoretical results are corroborated with rigorous proofs.

**Weaknesses:**

1. The writing of Section 3.1 and Section 3.2 is not really good.

In Section 3.1, the writing in Assumptions 3.2 should be improved. In particular, the authors should provide an intuitive explanation why they have to involve the Hermite polynomials and Hermite expansions when before or after introducing them. The assumptions on model parameters, e.g., $\omega^*_c$, $\omega^*_g$ are not explained thoroughly. Additionally, the roles of parameters $s_c$, $\nu$ are not introduced, making it inaccessible for people who do not have background on the considered problem.

In Section 3.2, the introduction to the formulation of MoE is quite complicated as the authors discuss different gating functions and routing strategies alternately without clear separation, making it difficult to digest.

2. Although the paper considers a different problem than that in (Chen et al., 2022), the practical implications are quite similar, that is, an MoE is capable of performing something that a single expert cannot do. It would be more interesting if the theory can provide new insight into improving the MoE performance.

3. **Major issue:** The mixture-of-experts formulation considered in this paper is far from practice. In particular, in the formulation of $\hat{F}_M$ in line 119 in the right column, whether an expert $f_m$ is activated or not depends entirely on the positivity its respective gating value $h_m$ rather than the magnitude rank (TopK) of the gating value as in the literature (Shazeer et al., 2017). This point makes me confused about the relevance of the theory in this paper.

**References**

Chen, Z., Deng, Y., Wu, Y., Gu, Q., and Li, Y. Towards understanding the mixture-of-experts layer in deep learning. In Advances in Neural Information Processing Systems, volume 35, pp. 23049–23062. Curran Associates, Inc., 2022.

Shazeer, N., Mirhoseini, A., Maziarz, K., Davis, A., Le, Q., Hinton, G., and Dean, J. Outrageously large neural networks: The sparsely-gated mixture-of-experts layer. In International Conference on Learning Representations, 2017.

**Questions For Authors:**

1. Why do the parameters $\omega^*_c,\omega^*_g,v^*c$ belong to the set $\mathbb{S}^{d-1}$ rather than $\mathbb{R}^d$?

2. How do you guarantee that $k$ experts will be activated given the router $1[h_m(x)\geq 0]$ in the formulation of $\hat{F}_m$? Based on the router formulation, there could be more or less than $k$ experts activated per input.

3. In high-level, what are the reasons for involving the Hermite expansion in this case?

4. Can the current results be generalized to the sparse MoE setting in (Shazeer at al., 2017)? If not, what are the main challenges?

**References**

Shazeer, N., Mirhoseini, A., Maziarz, K., Davis, A., Le, Q., Hinton, G., and Dean, J. Outrageously large neural networks: The sparsely-gated mixture-of-experts layer. In International Conference on Learning Representations, 2017.

**Relation To Broader Scientific Literature:**

The theory in the paper is to understand the underlying mechanism of Mixture-of-Experts and how it works.

**Theoretical Claims:**

I did not check the theoretical proofs carefully. However, I read the proof sketches and the arguments made sense to me.

---

> ### Author Rebuttal · Authors · 2025-04-01
>
> # Reviewer YoCg,
> We sincerely appreciate the reviewer's thoughtful and constructive feedback. Below, we carefully address the raised concerns and questions.
> Due to character limits, please see our response to Reviewer **wqTg** for references.
> ## Model parameters
> Following the reviewer's precise feedback, we will revise the manuscript to provide clear explanation of $w^*_c$, $w^*_g$, $s_c$, and $\nu$.
> ## Router
> Please refer to the **"Router"** part in our response to Reviewer **bEDR**.
> ## Implications
> Chen et al. (2022) present a failure of a single expert in binary classification where noisy features confound the true label. In contrast, our work focuses on a **regression** setting and reveals a different failure mode where **gradient interference** across clusters attenuates the signal, increasing the information exponent and hindering optimization. Our findings highlight how MoE mitigates interference by explicitly partitioning the network. Such interference has been recognized as a major challenge in multi-task learning (see lines 125-127 in the left column).
>
> Furthermore, we demonstrate that, for the router to distinguish between experts, a **weak recovery** of the feature index is necessary. Our analysis reveals that this requires a sample complexity of $\tilde{\Theta}{(d^{k^*-1})}$ during the exploration stage, implying that a sufficiently **long exploration phase** is essential before the router can begin meaningful learning. This contrasts with the linear expert setting in Li et al. (2025), as our findings arise from non-convex optimization.
>
> Finally, unlike in classification (Chen et al., 2022; Chowdhury et al., 2023), in the regression setting, **competition among experts** within the set of professional experts $\mathcal{M}_c$ may lead to load imbalance after the router has learned to assign inputs. This observation suggests a practical motivation for several heuristic approaches (see lines 122–131 in the right column).
> ## References
> According to the reviewer’s suggestion, we will incorporate references Chen et al. (2022) and Allen-Zhu & Li (2023) as inspiration for our data structure, and Liu et al. (2024) and Tang et al. (2025) as practical works on gradient interference in MoE in Assumption 3.2. We will also include Arous et al. (2021), Ge et al. (2018), Gudeja et al. (2018), Bietti et al. (2022), Damian et al. (2022), and Oko et al. (2024a) in lines 096–098 in the right column.
> ## $v_c$ and $g_j$
> We appreciate the reviewer pointing that out. The notation $v_c$ is correct, but $g_j$ is a typo. We will correct the typo in the final version.
> ## Q1.
> In our setting, feature vectors are sampled from $\mathbb{S}^{d-1}$ rather than from $\mathbb{R}^d$, in order to normalize signal strength for the analysis of simultaneous learning of multiple signals. This setting is frequently employed in prior works (e.g., Arous et al., 2021; Dudeja et al., 2018; Bietti et al., 2022; Damian et al., 2022; Oko et al., 2024ab). We would also like to note, in high-dimension, a uniform vector on $\mathbb{S}^{d-1}$ converges in distribution to $\mathcal{N}(0,d^{-1}I_d)$, in each coordinate, by concentration of measure.
> ## Q2.
> As the reviewer correctly pointed out, in $\hat{F}_m$, the number of activated experts is not fixed and $k$ is a stochastic variable. There have indeed been prior attempts to vary the number of activated experts depending on each token (Huang et al., 2024; Zeng et al., 2024).
> ## Q3.
> When analyzing the gradient, the correlation between the **nonlinear** target and the **nonlinear** activation arises. To decompose this correlation, we employ the Hermite expansion, which forms an **orthogonal** basis in the $L^2$ space under the Gaussian measure, as first used in Ge et al. (2018) and Dudeja et al. (2018).
> Specifically, we have
> $$
> \mathbb{E}\_{z \sim \mathcal{N}(0, I_d)} [ f^*\_c({w^*\_c }^\top z) a_{m,j}  \sigma_m({w_{m,j}}^\top z + b_{m,j}) ] = \sum_{i = k^*}^{p^*} \alpha_{m,j,i} \beta_{c,i} {\langle w_{m,j}, w^*_c \rangle}^i.
> $$
> This expresses the interaction as an expansion in powers of **alignment**. Crucially, the first non-zero order $k^*$ in the expansion (i.e., information exponent) **governs** the signal strength.
> ## Q4.
> In our study, the top-1 routing used in Phase II constitutes a sparse MoE. Although the adaptive top-$k$ routing employed in Phases III and IV allows $k$ to vary, it remains a sparse MoE in the sense that not all experts are activated. While adaptive top-$k$ is motivated by technical challenges in the theoretical analysis of nonlinear regression, we believe that end-to-end training with top-1 routing is feasible with the incorporation of auxiliary losses to control load balancing, which presents an interesting direction for future work.
>
> We will incorporate the revisions identified through this review into the final version. We would be happy to address any further questions during the discussion.

---

> > ### Comment · Reviewer_YoCg · 2025-04-03
> >
> > Dear the Authors,
> >
> > Thank you for your response, I really appreciate it. After reading the rebuttal, I am still not convinced by your response to the Weakness #3 about the sparse router. Note that all the popular LLMs such as GPT and DeepSeek use sparse routers which determine activated experts based on the magnitude of their affinity scores rather than the positivity of the affinity scores. Furthermore, in the literature of MoE, the sparse routers with the stochastic number of activated experts as proposed in the paper have not been shown to have clear benefits over the traditional sparse router as in (Shazeer et al., 2017). For these reasons, I think that the theory of this paper is quite irrelevant to practice. Hence, I will keep the original rating.

---

> > > ### Author Response · Authors · 2025-04-04
> > >
> > > ## Reviewer YoCg,
> > >
> > > We sincerely thank the reviewer for their thoughtful response and for taking the time to review our paper.
> > > While we understand that the reviewer does not plan to raise their score, we would like to respectfully clarify a point for completeness and for the benefit of other reviewers and the area chair.
> > >
> > > Despite theoretical idealizations, we firmly believe that both the implications we have presented and the insights into the learning dynamics—specifically, how the router learns the underlying cluster structure via gradients of the gating network, which reflect signals related to the weak recovery of the experts—are novel and well supported. Accordingly, we believe our findings are practically relevant.
> > >
> > > While it is true that in practical LLMs, the top-k experts are often selected based on the largest values of ${\theta_m}^\top x_c$, adaptive top-$k$ routing based on ${\theta_m}^\top x_c \geq 0$ in Phase III and IV was introduced to maintain analytical tractability and to better handle competition among professional experts, as heuristic methods such as auxiliary losses for learning nonlinear regression with nonlinear experts (a direction largely avoided in prior works) would significantly complicate the mathematical analysis.

---

### Decision · Program_Chairs · 2025-05-01

**Decision:**

Accept (poster)

**Comment:**

The paper investigates the sample and runtime complexity of Mixture-of-Experts (MoE) optimized using SGD when learning a regression problem with an underlying cluster structure of single index models.

All reviewers thought that the paper made some contributions to the theory of understanding MoE. The reviewers commented that the theory was somewhat limited and may not have too much impact practical since it was simplified, but they all found it a nice starting point to provable and theoretically understanding MoEs. As such, I recommend it for acceptance.

I will advise the authors to consider the comments made by the reviewers especially regarding cleaning up some of the proofs in the appendix.